# Annexin A1 is a polarity cue that directs mitotic spindle orientation during mammalian epithelial morphogenesis

Maria Fankhaenel [1,2], Farahnaz S. Golestan Hashemi[1,2], Larissa Mourao[3], Emily Lucas[1,2], Manal M. Hosawi [1,2], Paul Skipp[1,2,4], Xavier Morin[5], Colinda L.G.J. Scheele [3] & Salah Elias [1,2] ✉

Oriented cell divisions are critical for the formation and maintenance of structured epithelia. Proper mitotic spindle orientation relies on polarised anchoring of force generators to the cell cortex by the evolutionarily conserved protein complex formed by the $G_{\alpha i}$ subunit of heterotrimeric G proteins, the Leucine-Glycine-Asparagine repeat protein (LGN) and the nuclear mitotic apparatus protein. However, the polarity cues that control cortical patterning of this ternary complex remain largely unknown in mammalian epithelia. Here we identify the membrane-associated protein Annexin A1 (ANXA1) as an interactor of LGN in mammary epithelial cells. Annexin A1 acts independently of $G_{\alpha i}$ to instruct the accumulation of LGN and nuclear mitotic apparatus protein at the lateral cortex to ensure cortical anchoring of Dynein-Dynactin and astral microtubules and thereby planar alignment of the mitotic spindle. Loss of Annexin A1 randomises mitotic spindle orientation, which in turn disrupts epithelial architecture and luminogenesis in three-dimensional cultures of primary mammary epithelial cells. Our findings establish Annexin A1 as an upstream cortical cue that regulates LGN to direct planar cell divisions during mammalian epithelial morphogenesis.

Oriented cell divisions (OCDs) represent a fundamental mechanism for epithelial tissue morphogenesis, repair and differentiation during development and homeostasis[1]. Disruption in OCDs has direct implications for developmental disorders and cancer[2]. OCDs are defined by the orientation of the mitotic spindle, which in turn determines the position and fate of daughter cells in epithelial tissues[1,3]. The evolutionarily conserved ternary complex formed by the $G_{\alpha i}$ subunit of heterotrimeric G proteins, the Leucine-Glycine-Asparagine repeat protein (LGN) (also known as GPSM2: G-protein-signalling modulator 2) and the nuclear mitotic apparatus protein (NuMA), are essential regulators that polarise force generators and capture astral microtubules at the cell cortex to orient the mitotic spindle[4]. During mitosis,

LGN is recruited to the cell cortex through the interaction of its carboxy-terminal GoLoco motifs with GDP-bound $G_{\alpha i}$, which is anchored at the plasma membrane through myristoylation. The N-terminal tetratricopeptide repeat (TPR) domain of LGN interacts with NuMA that is released in the cytoplasm upon nuclear-envelope breakdown[4]. Polarised cortical localisation of the $G_{\alpha i}$-LGN-NuMA complex is crucial for the interaction of NuMA with astral microtubules and recruitment of the Dynein-Dynactin motor complex through direct NuMA binding, which triggers minus-end directed movement of Dynein, generating pulling forces on astral microtubules that position and orient the mitotic spindle[5]. $G_{\alpha i}$ subunits have a default distribution throughout the whole cell surface and do not contribute to the

[1]School of Biological Sciences, University of Southampton, Southampton SO17 1BJ, UK. [2]Insitute for Life Sciences, University of Southampton, Southampton SO17 1BJ, UK. [3]VIB-KULeuven Center for Cancer Biology, Herestraat 49, 3000 Leuven, Belgium. [4]Centre for Proteomic Research, University of Southampton, Southampton SO17 1BJ, UK. [5]Ecole Normale Supérieure, CNRS, Inserm, Institut de Biologie de l'Ecole Normale Supérieure (IBENS), PSL Research University, Paris, France. ✉e-mail: s.k.elias@soton.ac.uk

polarisation of LGN at cortical crescents facing the spindle poles[6]. The mechanisms that polarise LGN at the cell cortex remain unfolding.

The orientation of the mitotic spindle in epithelia is instructed by polarity cues that translate external signals to pattern cortical LGN. The core members of the Par polarity complex (Par3, atypical protein kinase C (aPKC), and Par6), were shown to play a key role in nearly every instance of OCDs[3,4,7]. In *Drosophila* neuroblasts, establishment of an apico-basal polarity axis allows the adapter protein Inscuteable (Insc) to accumulate at the apical cortex, where it binds to Par3, facilitating apical recruitment of LGN to form a ternary complex that orients the mitotic spindle perpendicularly to the epithelial surface[8,9]. Through the same mechanism, mammalian Insc (mInsc) controls perpendicular divisions in the *mouse* embryonic epidermis to promote stratification[10]. Combined loss of $G_{\alpha i}$ and mInsc in the *mouse* epidermis results in a shift towards planar divisions, suggesting that these occur by default in the absence of apical polarity cues[10]. Nonetheless, studies in *canine* kidney MDCK cells showed that phosphorylation of LGN by apical aPKC mediates its exclusion to the lateral cortex, favouring the planar orientation of the mitotic spindle[11]. Other polarity cues instruct lateral patterning of cortical LGN in mammalian epithelial cells. While E-cadherin, Afadin, Disk large (Dlg) and Ric-8A direct the recruitment of LGN at the lateral cortex[12–16], ABL1 and SAPCD2 act on its cortical accumulation and restriction[17,18]. These diverse mechanisms that control LGN localisation at restricted cortical domains have also been shown to act in a context-dependent and tissue-specific manner to ensure adequate positioning of cell fate determinants and daughter cells during planar or perpendicular cell divisions.

The mammary gland is a unique organ that develops predominantly after birth and represents a powerful model system to study the mechanisms of adult mammalian epithelial morphogenesis. Postnatal morphogenesis of the mammary gland in response to hormonal stimuli drives dramatic tissue turnover and remodelling during successive pregnancies[19]. The contribution of OCDs to mammary epithelial morphogenesis remains ill-defined[3]. A few studies have shown that Aurora-A kinase (AurkA), Polo-like kinase 2 (Plk2), integrins, stathmin, huntingtin and Slit2/Robo1, control mitotic spindle orientation in mammary epithelial cells (MECs)[20–25]. So far, only huntingtin and Slit2/Robo1 were reported to regulate cortical localisation of LGN in the mammary epithelium[22,25]. Recent evidence strongly suggests that OCDs affect cell fate outcomes in mammary stem/progenitor cells[3,24,26], and their dysregulation is linked to breast cancer development and aggressiveness[3,24,26,27]. Thus, unravelling the molecular components and precise mechanisms of mitotic spindle orientation in the mammary epithelium remains an important priority.

Here we exploit affinity purification and mass spectrometry methods to characterise the cortical interactome of LGN in mitotic MECs. We identify Annexin A1 (ANXA1), a potent membrane-associated immunomodulatory protein[28], as an LGN interactor that has not been reported previously to play any function in cell division. We demonstrate that ANXA1 interacts with LGN to restrict its distribution to the lateral cell cortex, thereby regulating the dynamic and planar orientation of the mitotic spindle. In addition, we show that ANXA1-mediated regulation of planar division is crucial for epithelial architecture and luminogenesis in three-dimensional (3D) cultures of primary MECs. Collectively, the present experiments establish ANXA1 as a key polarity cue that instructs the accumulation of LGN at the lateral cell cortex, providing a mechanism that controls planar cell divisions during mammalian epithelial morphogenesis.

## Results
### LGN interactome in mitotic MECs
To identify LGN-binding proteins, we generated clonal MCF-10A MECs stably expressing GFP-LGN. GFP-expressing MCF-10A cells were used as controls. GFP-LGN display the same mitosis-dependent cortical localisation as compared to endogenous LGN (Supplementary Fig. 1a)

and mitosis progression is not affected by the expression of GFP-LGN (Supplementary Fig. 1b). We used affinity purification to isolate the GFP-LGN complex from cells synchronised in mitosis (arrested in metaphase), and in interphase (arrested in G2 phase) (Supplementary Fig. 1c). We validated synchronisation efficiency by assessing the accumulation of phospho-Histone H3, and cellular distribution of LGN and microtubules (Supplementary Fig. 1d, e). Finally, we confirmed that the protocol specifically purifies the exogenous LGN in the GFP-LGN-bead-bound fractions as compared to controls (Supplementary Fig. 1f).

The purified GFP-LGN protein complex was trypsin-digested and analysed using liquid chromatography-tandem mass spectrometry (LC-MS/MS). All mass spectrometry proteomic data obtained from interphase and metaphase cells in two independent experiments have been summarised in Supplementary Dataset 1 and deposited to the ProteomeXchange Consortium via the PRIDE partner repository[29]. Removal of proteins represented in the control beads, as well as common MS contaminants using the Contaminant Repository for Affinity Purification (CRAPome) database[30] identified an LGN/GPSM2 protein subnetwork of 18 proteins in metaphase cells (Fig. 1a, Supplementary Dataset 2). The LGN protein network includes known interactors such as NuMA, INSC and its partner PAR3 (Fig. 1a, Supplementary Dataset 2). Our data also suggest that NUMB, a key modulator of NOTCH and regulator of asymmetric cell divisions[3], may form a complex with INSC and PAR3. A Gene Ontology (GO) analysis revealed that "spindle organisation", "establishment of spindle localisation", "regulation of spindle assembly", "ciliary basal body-plasma membrane docking", "spindle assembly", "regulation of mitotic spindle organisation", "metaphase plate congression", "regulation of protein localisation to cell periphery", "regulation of chromosome segregation", and "regulation of mitotic sister chromatid segregation", are the most enriched biological processes, and that the identified proteins are involved in pathways regulating mitosis progression, centrosome formation and maturation, and microtubule organisation (Fig. 1b, Supplementary Dataset 2). Several proteins including PLK1, Dynein (DYNC1H1), p150$^{Glued}$ (DCTN2), γ-tubulin (TUBG1), importin-α1 (KPNA2) and RAN, known to form a complex with NuMA to regulate mitotic spindle assembly and orientation, are found in the LGN protein network and co-purify with additional mitotic spindle regulators, namely CLASP1, Cyclin B1 (CCNB1) and HAUS6[6,31–46], in the NuMA subnetwork where we also found the oncogene EIF3E[46,47]. Finally, we identified a ternary complex comprising the membrane-associated protein Annexin A1 (ANXA1)[48] and its partner S100A11 (S100 Ca$^{2+}$-binding protein A11)[49], as well as the Serum Amyloid A-1 protein (SAA1)[50], that co-purifies with LGN. We decided to gain more insight into this interaction. When we performed pull-down assays combined with western blotting analysis, we determined that GFP-LGN precipitates ANXA1 (Fig. 1c). Reciprocal immunoprecipitation shows that ANXA1-mCherry co-purifies with LGN (Fig. 1d). To further validate the association between ANXA1 and LGN, we used an in situ proximity ligation assay (PLA). We detected proximity between ANXA1 and LGN in the cytoplasm and the cell cortex, where the cortical signal was higher in metaphase as compared to interphase (Fig. 1e). Taken together these experiments uncover ANXA1 as a cortical interactor of LGN in mitotic mammalian epithelial cells.

### Cell-cycle-dependent regulation of ANXA1 association with LGN at the cell cortex
Previous studies in post-mitotic epithelial cells established ANXA1 as a cytosolic protein that is recruited to the phospholipids of the plasma membrane and intracellular vesicular organelles, in a Ca$^{2+}$-dependent manner[51], but no mitosis-specific function has been reported for the protein. We characterised the cell-cycle dependent localisation of ANXA1 in MCF-10A cells by immunofluorescence and confocal imaging. We observed that ANXA1 displays a consistent accumulation at

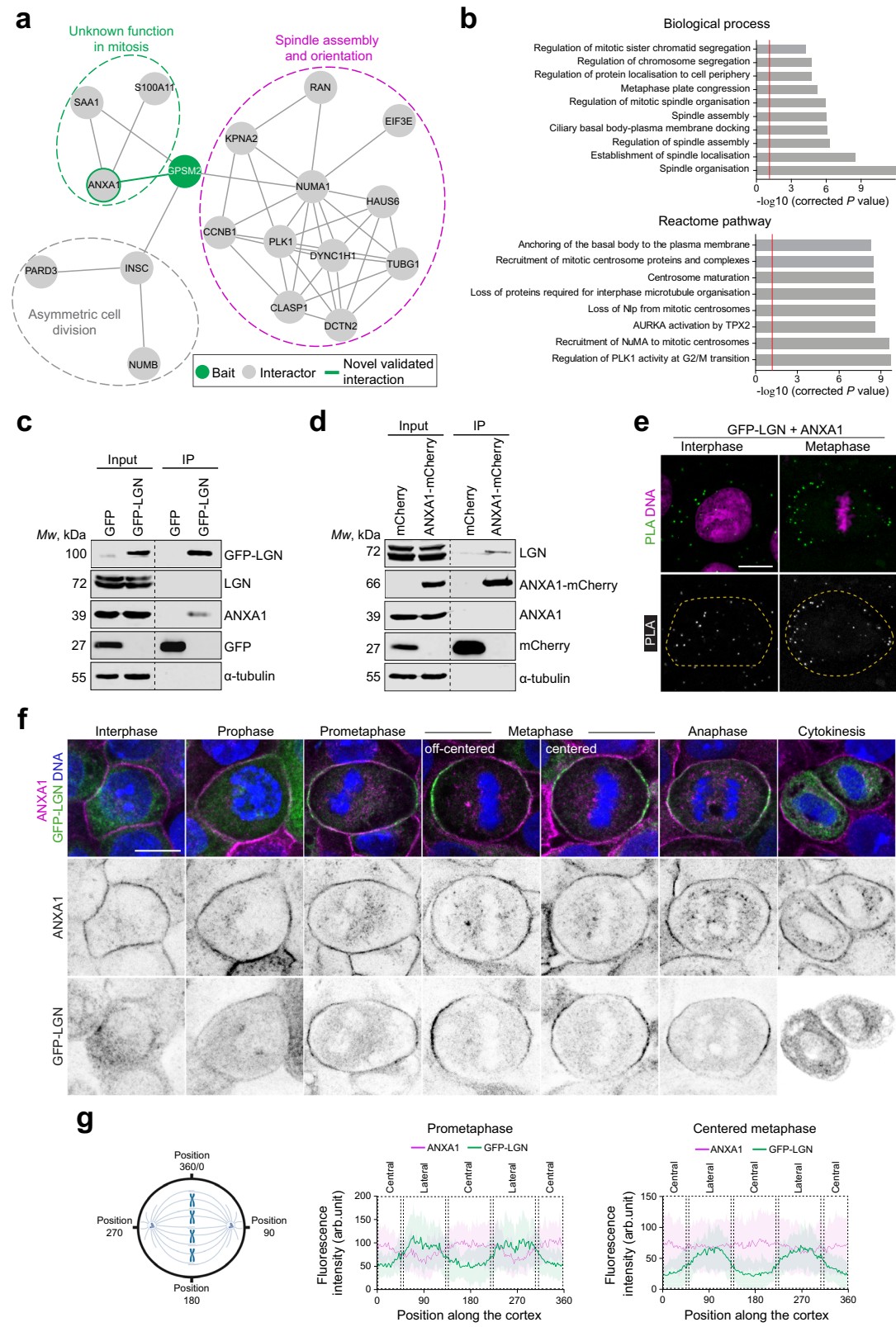

the cell cortex, where it co-localises with LGN, NuMA and S100A11 during mitosis (Fig. 1f, Supplementary Fig. 2a). A fraction of ANXA1 redistributes to the cytosol to localise around microtubules throughout mitosis (Supplementary Fig. 2b). We then further assessed the co-distribution of endogenous and ectopically expressed ANXA1 and LGN (Fig. 1f, Supplementary Fig. 2c, d). LGN displays a diffuse cytosolic localisation in interphase, whereas ANXA1 is uniformly distributed at

the cell cortex (Fig. 1g, Supplementary Fig. 2c, d). At mitosis entry, LGN relocates to the cell cortex where it colocalises with ANXA1. Consistent with previous studies[6,52,53], LGN is excluded from the central cortex during prometaphase, to polarise at the lateral cortex at metaphase. In sharp contrast, ANXA1 accumulates at the central cortex during prometaphase, while its amounts at the lateral cortex are lower as compared to LGN (Fig. 1f, g, Supplementary Fig. 2c, d). Moreover, before

**Fig. 1 | ANXA1 interacts with LGN at the cell cortex during mitosis. a** Protein-protein interaction network of LGN (GPSM2, green) showing several interactors (grey) of known and unknown mitotic functions. The green line highlights a novel interaction between LGN and ANXA1, which is further validated in this study. **b** Gene Ontology analysis of the LGN network presenting terms from Biological process (top) and Reactome pathway analysis (bottom) significantly enriched. *P*-values were derived from the hypergeometric test (two-sided Fisher's Exact Test) with Bonferroni step-down correction from multiple tests. **c** ANXA1 and GFP-LGN co-immunoprecipitate from MCF-10A stably expressing GFP-LGN, arrested in metaphase. Lysates were subjected to affinity purification with GFP-Trap beads. The immunoprecipitates (IP) were analysed by western blotting (3 independent experiments). **d** LGN and ANXA1-mCherry co-immunoprecipitate from clonal MCF-10A cells stably expressing ANXA1-mCherry, arrested in metaphase. Lysates were subjected to affinity purification with RFP-Trap beads. The immunoprecipitates were analysed by western blotting (3 independent experiments). **e** Proximity ligation assay (PLA) performed using GFP and ANXA1 antibodies in GFP-LGN-expressing MCF-10A cells. Cells were counterstained with DAPI (DNA, magenta) (3 independent experiments). Yellow dotted lines indicate the cell contour. **f** Confocal images of representative MCF-10A cells stably expressing GFP-LGN (green) stained for ANXA1 (magenta) and counterstained with DAPI (DNA, blue) (3 independent experiments). **g** Average cortical fluorescence intensity profiles of GFP-LGN and ANXA1 in prometaphase cells (left) and metaphase cells (right) (3 independent experiments, prometaphase *n* = 33 cells; metaphase *n* = 36 cells). Data are presented as mean ± s.e.m. Positions along the cortex in the graphs correspond to coordinates as illustrated in the schematic. arb. units (arbitrary units). All scale bars, 10 μm. Source data are provided as a Source Data file.

the metaphase plate finds a central position, ANXA1 accumulates more at the closest lateral cortex to the chromosomes whereas LGN is more abundant at the opposing distant cortex (Fig. 1f). Once chromosomes are aligned centrally in metaphase, LGN distributes equally at the lateral cortex, consistent with previous studies[6], while ANXA1 displays a more uniform, circumferential cortical distribution, a pattern observed until cytokinesis (Fig. 1f, g, Supplementary Fig. 2c, d). Collectively these experiments indicate that the distribution of ANXA1 along with LGN is spatiotemporally regulated during mitosis.

## ANXA1 restricts LGN localisation to the lateral cortex during mitosis

LGN orients the mitotic spindle by localising NuMA and Dynein-Dynactin to the cell cortex[3,4]. To investigate a possible function for ANXA1 in controlling LGN cortical localisation, we performed live imaging and assessed the effect of ANXA1 knockdown on the dynamic distribution of GFP-LGN during mitosis in siRNA-transfected MCF-10A cells. Twenty-four hours (24 h) post-transfection, cells were left for 48 h in culture to polarise and establish cell-cell adhesions, which are essential to ensure controlled orientation of the mitotic spindle in epithelial cells[14,15]. In cells transfected with a control siRNA (si-Control), we observed that GFP-LGN is recruited to the cell cortex during prometaphase to accumulate bilaterally at the cortex opposite to the spindle poles during metaphase (Fig. 2a, Supplementary Movie 1), consistent with previous studies in HeLa cells[6]. The amounts of cortical GFP-LGN decrease progressively during anaphase, until telophase and cytokinesis where the protein redistributes to the cytosol (Fig. 2a). In contrast, siRNA-mediated knockdown of ANXA1 impairs the dynamic distribution of GFP-LGN at the cortex in ~78% of ANXA1-depleted cells (Fig. 2a–c, Supplementary Movie 2, Supplementary Movie 3, Supplementary Movie 4). We found that GFP-LGN remains at the central cortex in 50.37 ± 5.19% of ANXA1-depleted cells, while in 17.78 ± 3.40% the protein shows a circumferential distribution. Additionally, 10.37 ± 5.79% of ANXA1-depleted cells display a unilateral cortical accumulation of GFP-LGN. We did not find cells negative for cortical GFP-LGN in the absence of ANXA1, allowing us to conclude that ANXA1 is required for LGN restricted distribution to the lateral cortex, rather than for its cytoplasm-to-cortex recruitment.

Next, we performed immunofluorescence and confocal imaging to evaluate the extent to which ANXA1 affects the ability of LGN to localise NuMA and the p150$^{Glued}$ Dynactin subunit to the lateral cortex during metaphase. In agreement with our live imaging results, ANXA1 knockdown prevents the bilateral cortical accumulation of LGN, with the protein displaying central (si-ANXA1#1: 53.33 ± 7.70%; si-ANXA1#2: 43.02 ± 8.76%), circumferential (si-ANXA1#1: 19.68 ± 3.86%; si-ANXA1#2: 25.7 ± 1.50 %), or unilateral (si-ANXA1#1: 16.51 ± 5.20%; si-ANXA1#2: 14.26 ± 7.20%) distributions in ANXA1-depleted cells (Fig. 2d, e). Similarly, ANXA1 knockdown impairs NuMA cortical-lateral restriction, which displays a central distribution in a large proportion of ANXA1-depleted cells (si-ANXA1#1: 35.71 ± 7.14%; si-ANXA1#2: 27.34 ± 8.77%) (Fig. 2d, e). By contrast, we found that p150$^{Glued}$ was

absent at the cell cortex in the majority (si-ANXA1#1: 48.60 ± 3.01%; si-ANXA1#2: 48.46 ± 0.79%) of ANXA1-depleted cells (Fig. 2d, e), indicating an impairment in its cortical recruitment. Ectopic expression of ANXA1-mCherry that is si-ANXA1#2-resistant rescues the cortical localisation defects of LGN, NuMA and p150$^{Glued}$, induced upon depletion of endogenous ANXA1 (Fig. 2d–f). Interestingly, ANXA1 knockdown did not affect G$_{\alpha i}$ cortical distribution (Fig. 2g, h). These results further indicate that ANXA1 is required for the restricted accumulation of the LGN-NuMA complex at the lateral cortex during mitosis, independently of G$_{\alpha i}$.

ANXA1 recruitment to the membrane phospholipids is facilitated by its binding to S100A11[54–56]. This was suggested to be modulated by phosphorylation of ANXA1 on its NH$_2$-terminal serine 5 (Ser5) by the TRPM7 (transient receptor potential melastatin 7) channel-kinase[57]. To assess the requirement of ANXA1 localisation to the plasma membrane for the polarised cortical accumulation of LGN and NuMA, we treated MCF-10A cells for 2 h with 20 μM of TG100-115, a potent TRPM7 kinase inhibitor[58,59]. This concentration and duration of treatment with TG100-115 leaves cell proliferation unaffected. Upon TG100-115 treatment, ANXA1-S100A11 accumulation at the plasma membrane is impaired with ANXA1 redistributing uniformly to the cytoplasm during metaphase (Fig. 3a, b). Similarly to ANXA1 knockdown, this treatment results in an aberrant accumulation of LGN and NuMA at the cell cortex and impairs p150$^{Glued}$ cortical recruitment (Fig. 3c, d), further indicating that ANXA1 serves as a membrane-associated molecular landmark to ensure proper cortical distribution of LGN and its effectors, NuMA and Dynein-Dynactin, during metaphase.

## ANXA1 acts upstream of LGN to regulate mitotic progression and spindle orientation

Proper cortical localisation of the LGN-NuMA-Dynein-Dynactin complex is required to generate polarised pulling forces on astral microtubules that orient the mitotic spindle[4]. The results above strongly suggest that ANXA1 regulates mitotic spindle positioning. To test this possibility, we performed live imaging in control and ANXA1-depleted MCF-10A cells, in which we labelled microtubules with SiR-tubulin[60] and DNA with Hoechst 33342. First, we observed significant defects in the dynamics of cell division in a vast majority of ANXA1-depleted cells (90.95 ± 2.15%) as compared to controls (12.04 ± 2.45%) (Fig. 4a, b, Supplementary Movie 5, Supplementary Movie 6, Supplementary Movie 7, Supplementary Movie 8). The proportion of cells that completed mitosis decreases significantly upon ANXA1 knockdown (si-ANXA1#1: 57.14 ± 9.18%, versus si-Control: 100 ± 0.00%) (Fig. 4c). In ANXA1-depleted cells that completed mitosis, the duration of the mitotic process is significantly extended as revealed by increased transition time from nuclear envelope breakdown (NEBD) to anaphase onset (si-ANXA1#1: 54.11 ± 1.08 min versus si-Control: 38.56 ± 1.52 min) (Fig. 4d). The size and morphology of the mitotic spindle are affected as well, with ~32% and ~14% of ANXA1-depleted cells displaying abnormal bipolar or multipolar spindles, respectively (Fig. 4e). Our analyses also reveal that ANXA1 knockdown results in chromosome

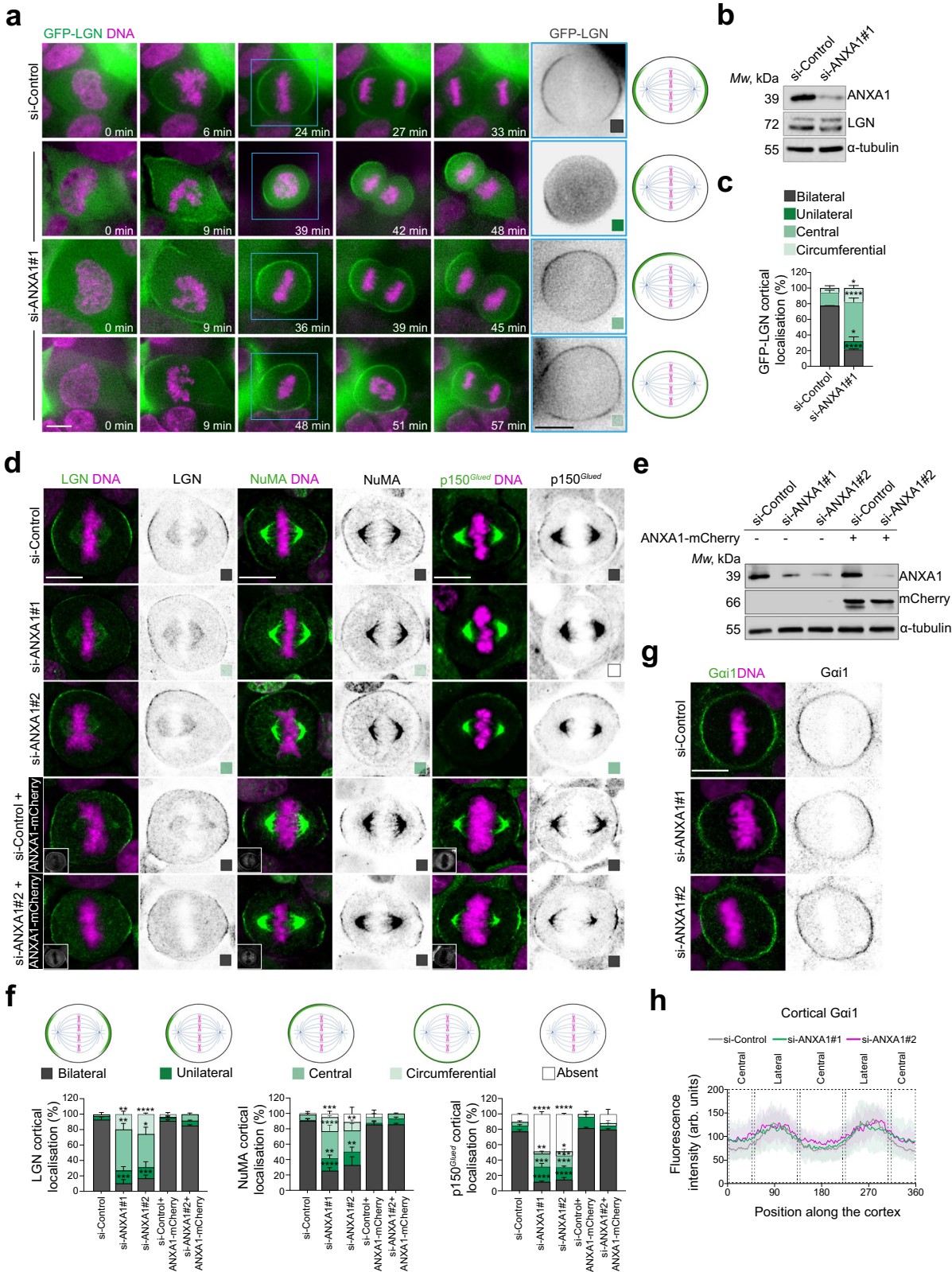

segment defects in ~32% of cells (Fig. 4e). During metaphase, ~82% of ANXA1-depleted cells display excessive oscillations of the mitotic spindle relative to the z axis between successive time frames, whereas in control cells the spindle does not display notable oscillatory z rotations and remains in a planar position (Fig. 4f–h). We further confirmed these observations using immunofluorescence and confocal imaging, which shows that virtually all mitotic spindles in the

controls orient parallel to the substratum plane (3.96° ± 0.31°), whereas ANXA1-depleted cells display spindle orientation defects (si-ANXA1#1: 15.37° ± 0.77°; si-ANXA1#2: 18.73° ± 1.73°), which are rescued upon ectopic expression of the si-ANXA1#2-resistant ANXA1-mCherry (si-ANXA1#2 + ANXA1-mCherry: 4.6° ± 0.5°) (Supplementary Fig. 3a, b). Live imaging experiments also show a significant increase in mitotic spindle oscillatory rotations within the xy plane upon ANXA1

**Fig. 2 | ANXA1 is required for correct localisation of the LGN-NuMA-Dynein-Dynactin complex at the cell cortex during mitosis. a** Left: time-lapse images of representative MCF-10A cells stably expressing GFP-LGN (green), treated with si-Control or si-ANXA1#1. DNA is stained with Hoechst 33342 (magenta) 30 min before imaging. Right: illustration showing the observed cortical localisation of GFP-LGN. **b** Western blotting of extracts from siRNA-transfected cells. **c** Percentage of cortical localisation of GFP-LGN in siRNA-transfected cells (si-Control: $n = 53$ cells; si-ANXA1#1: $n = 42$ cells). Two-sided $t$-test, bilateral: ****$P < 0.000001$; unilateral: *$P = 0.0475$; central: ****$P = 0.000004$; circumferential: *$P = 0.0233$. **d** Confocal images of representative si-Control-, si-ANXA1#1- and si-ANXA1#2-transfected metaphase cells expressing or not ANXA1-mCherry and stained for LGN, NuMA or p150$^{Glued}$ (green) and counterstained with DAPI (DNA, magenta). **e** Western blotting of extracts from siRNA-transfected cells expressing or not ANXA1-mCherry. **f** Percentage of cortical localisation of LGN, NuMA or p150$^{Glued}$ in siRNA-transfected

metaphase cells expressing or not ANXA1-mCherry: LGN (si-Control: $n = 70$ cells; si-ANXA1#1: $n = 51$ cells; si-ANXA1#2: $n = 43$ cells; si-Control + ANXA1-mCherry: $n = 69$ cells; si-ANXA1#2 + ANXA1-mCherry: $n = 47$ cells), NuMA (si-Control: $n = 73$; si-ANXA1#1: $n = 62$; si-ANXA1#2: $n = 45$; si-Control + ANXA1-mCherry: $n = 41$ cells; si-ANXA1#2 + ANXA1-mCherry: $n = 50$ cells), p150$^{Glued}$ (si-Control $n = 79$; si-ANXA1#1 $n = 101$; si-ANXA1#2 $n = 99$; si-Control + ANXA1-mCherry: $n = 33$ cells; si-ANXA1#2 + ANXA1-mCherry: $n = 34$ cells). Two-way ANOVA with Dunnett's multiple comparisons test, *$P \le 0.05$, **$P \le 0.01$, ***$P \le 0.001$, ****$P \le 0.0001$. **g** Confocal images of representative si-Control-, si-ANXA1#1- and si-ANXA1#2-treated metaphase cells stained for G$_{\alpha i1}$ (green) and counterstained with DAPI (DNA, magenta). **h** Average cortical fluorescence intensity profiles of G$_{\alpha i1}$ in siRNA-transfected metaphase cells (si-Control: $n = 32$; si-ANXA1#1: $n = 30$; si-ANXA1#2: $n = 30$). All data are presented as mean ± s.e.m. from 3 independent experiments. arb. units (arbitrary units). All scale bars, 10 μm. Source data are provided as a Source Data file.

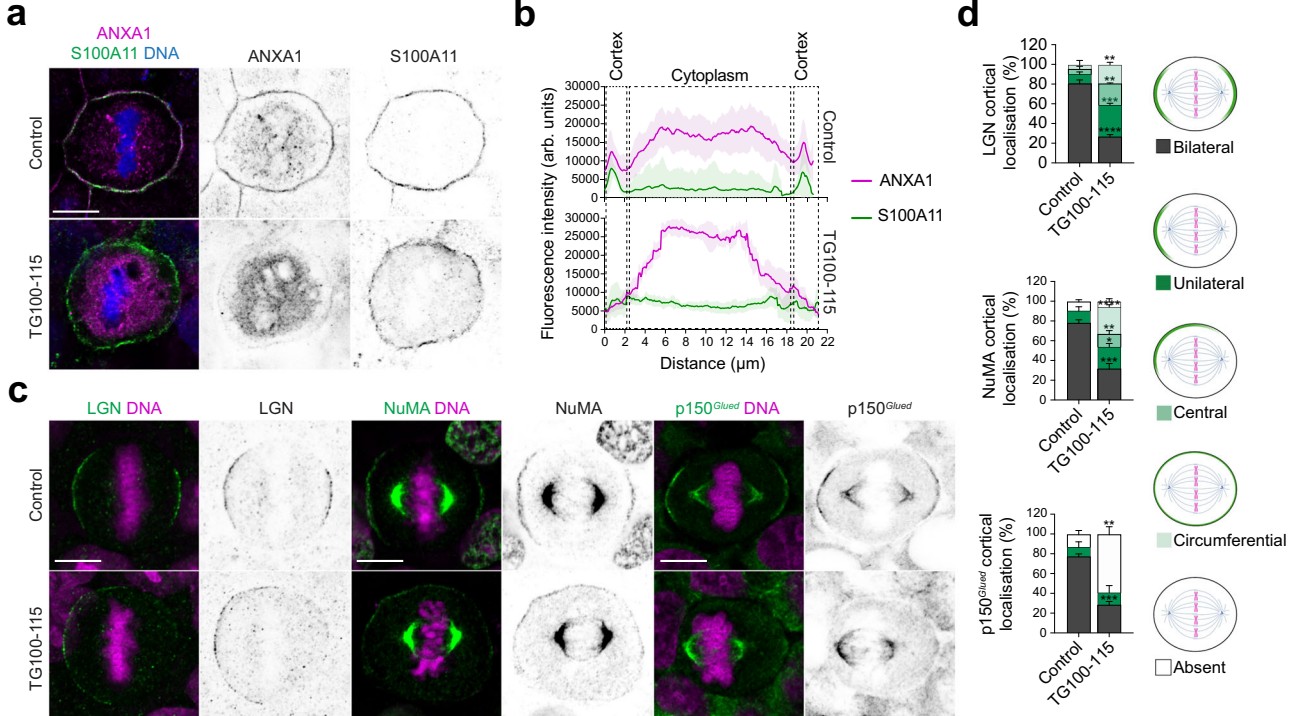

**Fig. 3 | ANXA1 localisation to the plasma membrane is required for the patterning of LGN-NuMA-Dynein-Dynactin at the lateral cortex during metaphase. a** Confocal images of representative MCF-10A cells treated with DMSO (Control) or 20 μM TG100-115 for 2 h, stained for ANXA1 (magenta) and S100A11 (green), and counterstained with DAPI (DNA, blue). **b** Average cortical and cytoplasmic fluorescence intensity profiles of ANXA1 and S100A11 from metaphase cells (Control: $n = 30$; TG100-115: $n = 30$). **c** Confocal images of representative MCF-10A cells treated with DMSO (Control) or 20 μM TG100-115 for 2 h, stained for LGN, NuMA or p150$^{Glued}$ (green), and counterstained with DAPI (DNA, magenta). **d** Percentage of

cortical localisation of LGN, NuMA or p150$^{Glued}$ in Control and TG100-115-treated metaphase cells: LGN (Control: $n = 39$ cells; TG100-115: $n = 37$), NuMA (Control: $n = 43$ cells; TG100-115: $n = 40$), p150$^{Glued}$ (Control: $n = 31$ cells; TG100-115: $n = 32$). Two-sided $t$ test, LGN (bilateral: ****$P = 0.00005$; unilateral: ***$P = 0.0008$; central: **$P = 0.006$; circumferential: **$P = 0.01$); NuMA (bilateral: ***$P = 0.001$; unilateral: *$P = 0.044$; central: **$P = 0.005$; circumferential: ****$P = 0.00009$; absent: $P = 0.405$); p150$^{Glued}$ (bilateral: ***$P = 0.0002$; unilateral: $P = 0.728$; absent: **$P = 0.005$). All data are presented as mean ± s.e.m. from 3 independent experiments. arb. units (arbitrary units). All scale bar, 10 μm. Source data are provided as a Source Data file.

knockdown, with spindle oscillation frequencies significantly higher as compared to controls (Fig. 4i–k). In control cells, mitotic spindles mostly oscillate around their initial position at metaphase onset, whereas ANXA1-depleted cells display random spindle displacements (Fig. 4l), where average movements away from the original position are significantly higher as compared to controls (si-ANXA1#1: $2.64 \pm 0.4$ μm versus si-Control: $1.74 \pm 0.16$ μm) (Fig. 4m). Spindle displacement speed was not affected upon ANXA1 depletion (si-ANXA1#1: $0.32 \pm 0.05$ μm/min versus si-Control: $0.27 \pm 0.02$ μm/min) (Fig. 4n). Together, these data indicate that ANXA1 is required for proper mitotic spindle positioning and assembly, faithful chromosome segregation to daughter cells, as well as mitotic progression.

We next tested whether ANXA1 lies upstream of LGN, using immunofluorescence and confocal imaging. Consistent with previous studies[15,61–63], we found that siRNA mediated LGN knockdown results in mitotic spindle misorientation, with spindle angles αz increasing from 3.14° in controls to 13.80° and 17.03° in cells treated with si-LGN#1 and si-LGN#2, respectively (Supplementary Fig. 3c, d). However, LGN depletion does not affect the cortical distribution of ANXA1, whereas it impairs the recruitment of NuMA to the lateral cortex (Supplementary Fig. 3e, f). These data suggest that ANXA1 acts upstream of LGN to control mitotic spindle orientation.

The anchoring of astral microtubules to the cell cortex, and the regulation of their number and stability by the LGN-NuMA-Dynein-

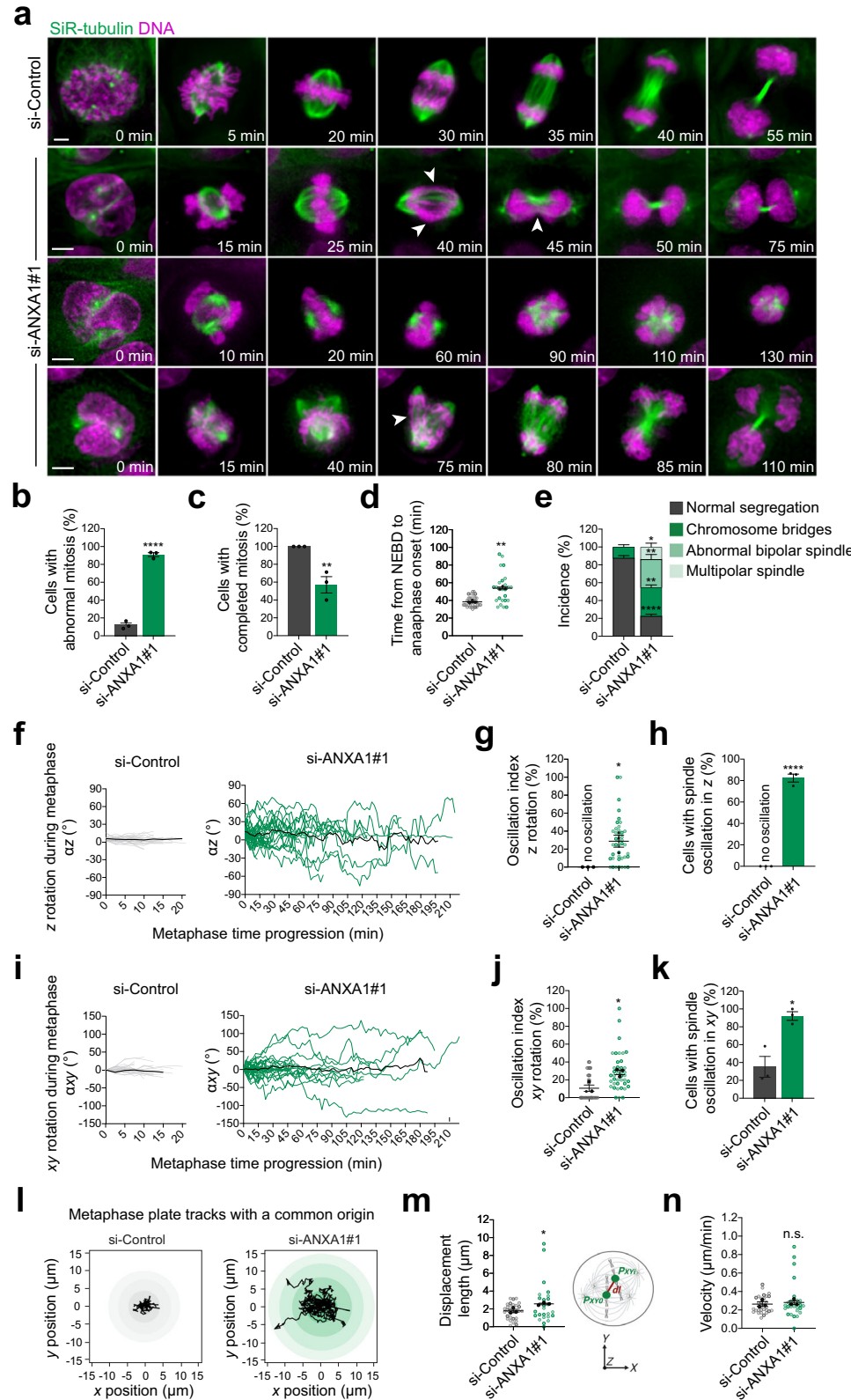

Dynactin complex are key to ensure correct mitotic spindle orientation and assembly[4]. We therefore examined microtubule stability and dynamics upon ANXA1 knockdown in mitotic cells. Using immunofluorescence and confocal imaging in MCF-10A cells labelled for α-tubulin or stably expressing the microtubule plus-end-binding protein EB3-GFP, we show that ANXA1 knockdown results in elongation and buckling of astral microtubules, while their overall intensity and number remain unaffected, suggesting an increased stability (Fig. 5a–d). Astral microtubule elongation and stabilisation correlate with spindle misplacement close to one side of ANXA1-depleted cells, suggesting imbalance in the forces exerted on the mitotic spindle (Fig. 5e). These results are in line with our earlier observations showing an increase in mitosis duration and proportion of cells displaying abnormal mitotic spindles in ANXA1-depleted cells. Actin cytoskeleton

**Fig. 4 | ANXA1 is required for correct mitotic progression and spindle dynamics. a** Time-lapse images of representative MCF-10A cells treated with si-Control or si-ANXA1#1. Microtubules are labelled with SiR-tubulin (green) and DNA with Hoechst 33342 (magenta), 2 h and 30 min before acquisition, respectively. Arrowheads indicate the phenotypes listed in (**e**). Scale bars, 5 μm. **b** Percentage of cells with abnormal mitosis (si-Control: $n = 33$; si-ANXA1#1: $n = 44$). Two-sided $t$-test, ****$P < 0.0001$. **c** Percentage of cells that completed mitosis (si-Control: $n = 33$; si-ANXA1#1: $n = 44$). Two-sided $t$-test, **$P = 0.0095$. **d** Time from nuclear envelope breakdown (NEBD) to anaphase onset in siRNA-transfected cells (si-Control: $n = 33$; si-ANXA1#1: $n = 25$). Two-sided $t$-test, **$P = 0.006$. **e** Percentage of chromosomes segregation and spindle assembly defects in siRNA-transfected cells (si-Control: $n = 33$; si-ANXA1#1: $n = 44$). Two-sided $t$ test, normal segregation: ****$P = 0.000033$; chromosome bridges: **$P = 0.0055$; abnormal bipolar spindle: **$P = 0.0043$; multi-polar spindle: *$P = 0.032$. **f** Dynamics of $z$ orientation (α$z$) during metaphase in siRNA-transfected cells (si-Control: $n = 33$; si-ANXA1#1: $n = 39$). Black lines represent the average spindle angles. **g** Oscillation index in siRNA-transfected cells, in the $z$-axis (si-Control: $n = 33$; si-ANXA1#1: $n = 41$). Two-sided $t$-test, *$P = 0.0125$.

**h** Percentage of cells with spindle oscillations in the $z$ axis (si-Control: $n = 33$; si-ANXA1#1: $n = 44$). Two-sided $t$-test, ****$P < 0.0001$. **i** Dynamics of $xy$ orientation (α$xy$) during metaphase in siRNA-transfected cells (si-Control: $n = 33$; si-ANXA1#1: $n = 39$). Black lines represent the average spindle angles. **j** Oscillation index in siRNA-transfected cells, in the $xy$ axis (si-Control: $n = 33$; si-ANXA1#1: $n = 41$). Two-sided $t$-test, *$P = 0.0153$. **k** Percentage of cells with spindle oscillations in the $xy$ axis (si-Control: $n = 33$; si-ANXA1#1: $n = 44$). Two-sided $t$ test, *$P = 0.0105$. **l** Origin-aligned metaphase plate tracks in the $xy$ plane of si-Control- (left) and si-ANXA1#1-transfected (right) cells (si-Control: $n = 33$; si-ANXA1#1: $n = 43$). **m** Displacement length from the starting point to the position at the end of metaphase in siRNA-transfected cells (si-Control: $n = 31$; si-ANXA1#1: $n = 30$). Two-sided $t$-test, *$P = 0.037$. **n** Velocity of metaphase plate movements in si-RNA-transfected, calculated by dividing the track length by metaphase duration (si-Control: $n = 31$; si-ANXA1#1: $n = 44$). Two-sided $t$-test, $P = 0.384$. All data are presented as mean ± s.e.m. from 3 independent experiments. Source data are provided as a Source Data file.

reorganisation during mitosis into a uniform contractile meshwork at the cell cortex drives the generation of cortical tensions, which are required for accurate mitotic spindle positioning[13,64,65]. ANXA1 has been shown to bind and bundle F-actin in a Ca$^{2+}$-dependent manner, ensuring a dynamic crosstalk between the plasma membrane and the actin cytoskeleton[66]. This prompted us to analyse the consequences of ANXA1 knockdown on cortical actin integrity during metaphase. Phalloidin labelling shows that F-actin organisation is perturbed in ANXA1-depleted cells, leading to a reduction in the cortical actin signal (Fig. 5f, g). Interestingly, treatment of MCF-10A cells with 1 μM of the actin depolymerising drug latrunculin A (Fig. 5h) does not affect the localisation of ANXA1 to the plasma membrane, whereas it impairs the recruitment of LGN, NuMA and p150$^{Glued}$ to the cell cortex during metaphase (Fig. 5i, j). Together, these experiments further sustain the hypothesis that ANXA1 is an upstream cortical cue regulating proper microtubule-cortex crosstalk to ensure accurate spindle assembly and orientation.

## ANXA1 is required for planar cell divisions and epithelial morphogenesis

How ANXA1 regulates normal mammary epithelial homeostasis and morphogenesis remains unknown. To characterise ANXA1-expressing cells in vivo, we first performed immunostaining experiments in *mouse* adult virgin mammary glands. We reveal that ANXA1 expression is restricted to the luminal epithelial compartment, where ANXA1$^+$ cells selectively co-express the luminal marker E-cadherin and not the basal marker keratin (K)14 (Fig. 6a). Immunofluorescence experiments in *human* healthy breast tissue confirms that ANXA1 is expressed in the luminal compartment, but also reveal ANXA1 expression in a few α-smooth muscle actin (α-SMA)-positive basal cells (Supplementary Fig. 4a). Consistent with our observations in MCF-10A cells, ANXA1 is mostly detected at the cell cortex of both *mouse* and *human* mammary epithelial cells (Fig. 6a, Supplementary Fig. 4a). Thus, we conclude that ANXA1 is a surface marker enriched in luminal cells of the *murine* and *human* mammary epithelia.

Most studies of mitotic spindle orientation in 3D culture are performed in MDCK and Caco-2 cells, which when grown in reconstituted extracellular matrix (ECM) such as Matrigel, form acini characterised by a central lumen surrounded by a monolayer of polarised epithelial cells[67]. Luminogenesis in MDCK- or Caco-2-derived acini is driven by a tight interplay between planar cell division and apico-basal polarity[68]. Our experiments in MCF-10A-derived acini show that ~60% and ~70% of cells align the mitotic spindle planarly to the basement membrane at 96 h and 192 h of 3D culture, respectively (Supplementary Fig. 4b, c). ANXA1 localises to the cell cortex of mitotic MCF-10A cells (Supplementary Fig. 4d), further indicating that cortical ANXA1 is required for planar mitotic spindle orientation in MCF-10A 3D cultures.

However, planar cell divisions do not contribute to cavitation in MCF-10A cell-derived acini (Supplementary Fig. 4b, c), consistent with studies showing that a central cavity is formed by apoptosis of inner cells at the end of cystogenesis in this 3D culture system[69]. Remarkably, our experiments in primary *human* MECs (hMECs) reveal multi-layered acini displaying high architectural heterogeneity (Supplementary Fig. 4e). Out of 601 acini, only 15 (~2.5%) form a central lumen and establish an apico-basal polarity with ANXA1 distributing at the cell cortex and Par6 accumulating at the apical surface (Supplementary Fig. 4e, f), suggesting that hMEC are not suitable to study the contribution of OCDs to luminogenesis in 3D culture. *Mouse* MECs (mMECs) grown in 3D culture self-organise and polarise around a central lumen, which is formed in an apoptosis-independent manner[70,71]. Therefore, we explored the contribution of ANXA1-mediated regulation of mitotic spindle orientation to luminogenesis in mMEC-derived acini. We isolated MECs from *mouse* mammary glands at days 15.5-16.5 of pregnancy (P15.5-P16.5) and cultured them in Matrigel[70]. Using immunofluorescence and confocal imaging we first characterised mMECs division modes throughout epithelial morphogenesis (Fig. 6b–d). After 48 h of culture, ~64% of mMECs align their mitotic spindle planar to the basement membrane (angle α 0°–30°), whereas the proportion of mMECs displaying planar divisions increases to ~90% and ~85% at 72 h and 96 h, respectively (Fig. 6b, c). This increase in planar divisions is accompanied with a re-organisation of mMECs into a monolayer around an expanding single central lumen that matures at 120 h, where mMECs cease to divide and complete their apico-basal polarisation, as revealed by the accumulation of F-actin and Par6 at their apical surface (Fig. 6b). Consistent with our results in 2D cultures of MCF-10A cells, ANXA1 distributes at the cell cortex during mitosis in mMECs (Fig. 6d). Together these findings, allow us to establish mMEC-derived acini as a useful 3D culture model for the study of the mechanisms linking mitotic spindle orientation to luminogenesis in the mammary epithelium.

To investigate the effect of ANXA1 knockdown on mitotic spindle orientation and luminogenesis in 3D culture, we generated mMECs expressing control or ANXA1 short hairpin (sh) RNAs (sh-Control versus sh-ANXA1#1 or sh-ANXA1#2) (Fig. 6e). As expected, ~87% of control MECs display planar cell divisions in 72 h acini (Fig. 6f, g). In sharp contrast, ~69% and ~70% of ANXA1-depleted mMECs fail to align planar mitotic spindles, upon sh-ANXA1#1 or sh-ANXA1#2 treatment, respectively, with spindle angles α greater than 30° (Fig. 6f, g). Moreover, while LGN localises at the lateral cortex in control mitotic mMECs, ANXA1 depletion impairs its polarised cortical distribution (Fig. 6h). By 120 h of 3D culture, control acini are mono-layered with a single central lumen, whereas ANXA1-depleted mMECs form larger, disorganised, and multi-layered acini, with multiple small or collapsed lumen (Fig. 6i-k). While controls are circular, ANXA1-depleted acini

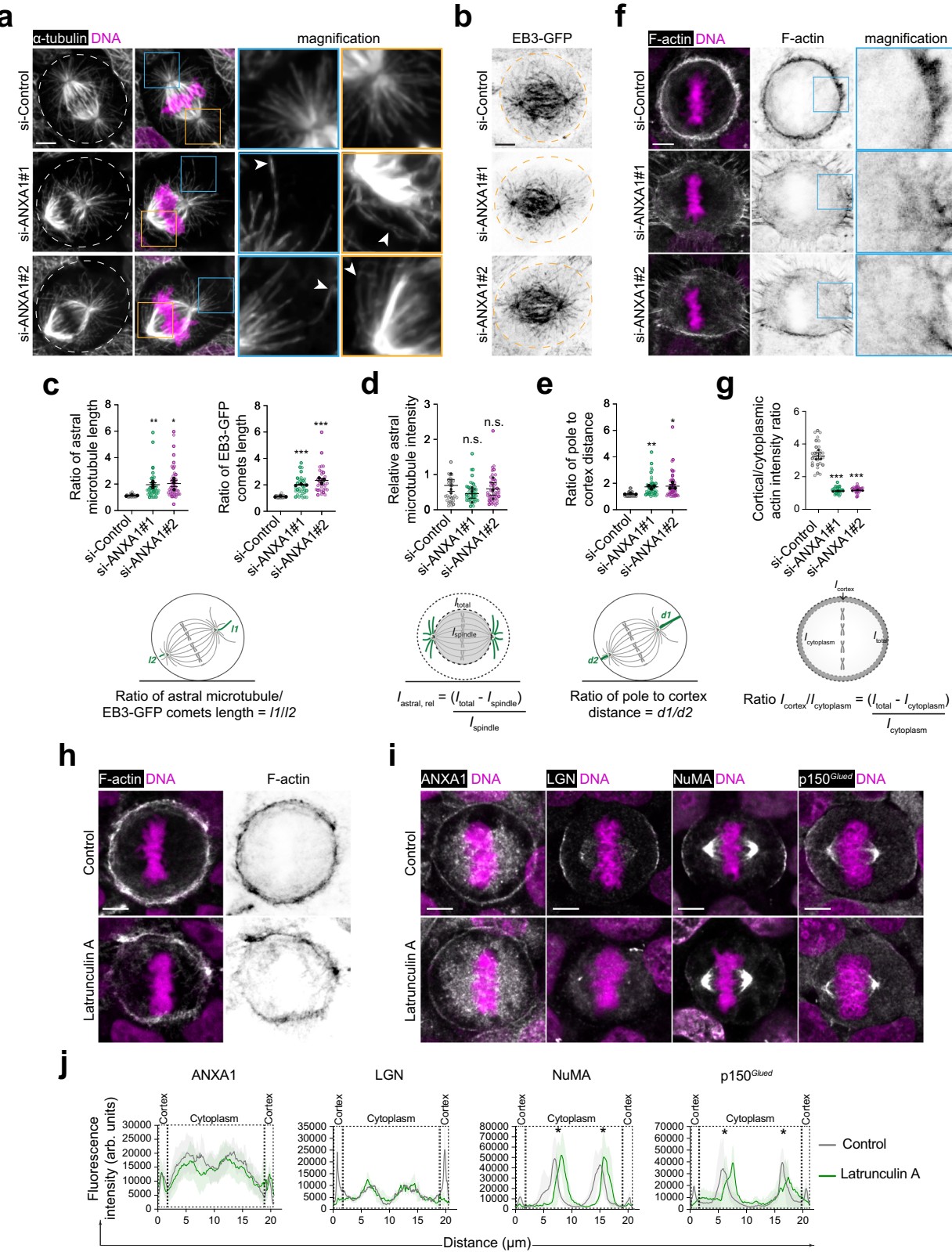

display irregular and elongated shapes, further indicating aberrant epithelial architecture (Fig. 6i, j). Our experiments suggest that mitotic spindle misorientation upon ANXA1 knockdown results in daughter cells generated perpendicularly to the epithelial plane, thereby leading to epithelial cell multi-layering and lumen filling. Consistent with this, ANXA1-depleted mMECs display dramatic cytoarchitectural defects and fail to acquire correct apico-basal polarity as revealed by misslocalised E-cadherin and F-actin (Fig. 6i, l). Thus, ANXA1-mediated regulation of planar mitotic spindle orientation in mMECs ensures single luminogenesis and normal epithelial morphogenesis.

## Discussion

OCDs are essential for establishing proper epithelial tissue architecture and function[3,4]. Our molecular knowledge of the mechanisms

**Fig. 5 | Interplay between ANXA1, LGN-NuMA, astral microtubules and cortical F-actin during metaphase. a** Confocal images of representative metaphase MCF-10A cells transfected with si-Control, si-ANXA1#1 or si-ANXA1#2 stained for α-tubulin (grey) and counterstained with DAPI (DNA, magenta). White arrowheads indicate elongated and buckled astral microtubules. Dashed lines outline the cell contour. **b** Confocal images of representative metaphase MCF-10A cells stably expressing EB3-GFP and transfected with si-Control, si-ANXA1#1 or si-ANXA1#2. Dashed lines outline the cell contour. **c** Ratio of astral microtubule and EB3-GFP comets length ($l$) in siRNA-transfected cells: astral microtubules (si-Control: $n = 31$; si-ANXA1#1: $n = 43$; si-ANXA1#2: $n = 45$); EB3-GFP comets (si-Control: $n = 30$; si-ANXA1#1: $n = 34$; si-ANXA1#2: $n = 34$). $l1$ and $l2$ of astral microtubules were measured as depicted on the illustration. One-way ANOVA with Tukey's test, astral microtubules: $**P = 0.007$ and $*P = 0.035$; EB3-GFP: $***P = 0.001$ and $***P = 0.0009$. **d** Relative fluorescence intensities of astral microtubules ($I_{astral, rel}$) in siRNA-transfected cells (si-Control: $n = 31$; si-ANXA1#1: $n = 47$; si-ANXA1#2: $n = 47$). Fluorescence intensities on the spindle ($I_{spindle}$) and total cell ($I_{total}$) were measured as depicted on the illustration. One-way ANOVA with Tukey's test, $P = 0.467$ and $P = 0.676$. **e** Ratio of pole-to-cortex distance ($d$) in siRNA-transfected cells (si-Control: $n = 40$; si-ANXA1#1: $n = 48$; si-ANXA1#2: $n = 52$). Pole-to-cortex distance $d1$ and $d2$ were measured as depicted on the illustration. One-way ANOVA with Tukey's test, $**P = 0.004$ and $*P = 0.032$. **f** Confocal images of representative metaphase MCF-10A cells transfected with si-Control, si-ANXA1#1 or si-ANXA1#2 stained for F-actin (grey) and counterstained with DAPI (DNA, magenta). **g** Cortical to cytoplasmic ratio of actin fluorescence intensities ($I_{cortex}/I_{cytoplasm}$) in siRNA-transfected cells (si-Control: $n = 30$; si-ANXA1#1: $n = 30$; si-ANXA1#2 $n = 30$). Fluorescence intensities in the cytoplasm ($I_{cytoplasm}$) and total cell ($I_{total}$) were measured as depicted on the illustration. One-way ANOVA with Tukey's test, $***P = 0.001$ and $***P = 0.0003$. **h** Confocal images of representative MCF-10A cells treated with DMSO (Control) or 1 μM Latrunculin A for 30 min, stained for F-actin (grey) and counterstained with DAPI (DNA, magenta). **i** Confocal images of representative MCF-10A cells treated with DMSO (Control) or 1 μM Latrunculin A for 30 min, stained for ANXA1, LGN, NuMA or p150$^{Glued}$ (grey), and counterstained with DAPI (DNA, magenta). **j** Average cortical and cytoplasmic fluorescence intensity profiles of ANXA1, LGN, NuMA and p150$^{Glued}$ from metaphase cells (Control: $n = 30$; Latrunculin A: $n = 30$). Asterisks indicate the spindle poles. All data are presented as mean ± s.e.m. from 3 independent experiments. n.s. (not significant). arb. units (arbitrary units). All scale bars, 5 μm. Source data are provided as a Source Data file.

that align the mitotic spindle in polarised epithelia has advanced, largely from extensive studies in model systems in the genetically tractable *Drosophila melanogaster* and *Caenorhabditis elegans*[3,4]. The $G_{αi}$-LGN-NuMA ternary complex is established as an evolutionarily conserved spindle orientation machinery that anchors the Dynein-Dynactin force generator complex and astral microtubules to the cell cortex, ensuring correct cell division orientation to achieve proper tissue morphogenesis and differentiation[3,4]. However, the molecular mechanisms governing mitotic spindle positioning in mammalian epithelia remain poorly defined. The present experiments uncover ANXA1, a membrane-associated protein, as an interactor of LGN, regulating planar mitotic spindle orientation for correct mammary epithelial morphogenesis. We further report that ANXA1 is an upstream cortical cue that is required for F-actin organisation and LGN and NuMA polarised accumulation at the lateral cortex, ensuring balanced Dynein-Dynactin-mediated pulling forces on astral microtubules and correct alignment of the mitotic spindle (Fig. 7).

Cumulative evidence points to additional molecular mechanisms in mammalian cells that control the cortical localisation of LGN upon its $G_{αi}$-mediated active recruitment. A chromosome-derived RANGTP gradient acts in concert with PLK1 to mediate the exclusion of LGN from the cortical regions above chromosomes towards the lateral cortex in HeLa cells[6,52,53], where the polarity proteins ABL1 and SAPCD2 also control the cortical abundance of LGN[17,18]. By contrast, Afadin and E-cadherin facilitate LGN and NuMA recruitment to the lateral cortex in HeLa cells[13] and to the cell-cell contacts in MDCK cells[15], respectively. Our studies in mitotic MCF-10A cells identify ANXA1 as a polarity cue that co-purifies with LGN and controls its dynamic cortical distribution by ensuring its exclusion from the central cortex during prometaphase and stable accumulation at the lateral cortex at metaphase, in a $G_{αi}$-independent manner, thereby directing planar spindle orientation. By comparison, SAPCD2 interacts with LGN to modulate its translocation between the lateral and apical cortical domains to control the balance between planar and perpendicular divisions, respectively[18]. Upon ANXA1 depletion or inhibition, both LGN and NuMA remain at the cell cortex where their restricted lateral accumulation is similarly impaired, suggesting that ANXA1 does not act on the assembly of the LGN-NuMA complex, but rather controls its polarised cortical distribution. While TG100-115 is an established potent inhibitor of the TRPM7 kinase activity affecting the translocation of ANXA1 to the plasma membrane, the inhibitor also acts on the Phosphatidylinositol-3-kinase (PI3K) signalling[72]. This raises the possibility that PI3K inhibition may also affect ANXA1 localisation. However, cumulative evidence shows that ANXA1 acts as an upstream regulator of the PI3K pathway in several normal and cancer cell models[73–75]. In contrast to most PI3K

inhibitors[76], our experiments using short-term treatments with TG100-115 did not affect cell proliferation. Thus, it is unlikely that PI3K affects the localisation or function of ANXA1. Together, our findings are in sharp contrast with depletion of SAPCD2, Afadin or E-cadherin that abrogates NuMA cortical localisation, where NuMA was shown to compete LGN from SAPCD2, Afadin or E-cadherin[13,15,18]. Moreover, our proteomic data show that NuMA co-purifies with the ANXA1-LGN complex, suggesting that ANXA1, LGN and NuMA may form a ternary complex. Structure-function studies will be key to dissect the mechanisms underlying the assembly and spatiotemporal dynamics of the ANXA1-LGN-NuMA complex at the cell cortex and understand how ANXA1 acts with NuMA and synergises with $G_{αi}$ to restrict the cortical accumulation of LGN.

We show that disruption of the polarised cortical localisation of the LGN-NuMA-Dynein-Dynactin complex upon ANXA1 depletion leads to elongated and buckled astral microtubules affecting the assembly and orientation of the mitotic spindle, pointing to a role of ANXA1 in the regulation of cortical pulling forces on astral microtubules that control the dynamics of the mitotic spindle. Consistent with this, recent studies have shown that correct cortical targeting of NuMA and its binding to LGN and astral microtubules are required for the dynamic crosstalk between microtubules and the cell cortex and for the stabilisation of Dynein on astral microtubules to generate balanced forces that orient the mitotic spindle[13,34,77]. Moreover, ANXA1 depletion impairs F-actin integrity, further suggesting that the protein may regulate the microtubule-actin crosstalk at the cell cortex, which is essential to maintain the balanced forces on astral microtubules that define the mitotic spindle orientation axis[78]. Remarkably, while ANXA1 knockdown affects F-actin organisation and the polarised cortical accumulation of LGN and NuMA, depolymerisation of F-actin using latrunculin A abrogates the recruitment of LGN-NuMA to the cell cortex, similar to what was shown in HeLa cells[13], but has no effect on the localisation of ANXA1 to the plasma membrane. This further reinforces a model whereby ANXA1 acts as an upstream polarity cue that controls the cortical distribution of LGN-NuMA. Our results also suggest that additional molecular links may exist between ANXA1 and LGN to mediate the function of ANXA1 in the regulation of LGN-NuMA polarised accumulation at the cell cortex. ANXA1 may also act on F-actin dynamics at the cell cortex, influencing the microtubule-actin crosstalk and thereby the cortical localisation of LGN-NuMA. In fact, ANXA1 has been shown to bind and bundle F-actin in a $Ca^{2+}$-dependent manner[79]. ANXA1 $Ca^{2+}$-dependent translocation to the plasma membrane regulates membrane and F-actin dynamics[54,55]. Furthermore, ANXA1 influences actin dynamics via a direct interaction with the actin polymerising profilin[80,81]. Additionally, ANXA1 interacts with vimentin,

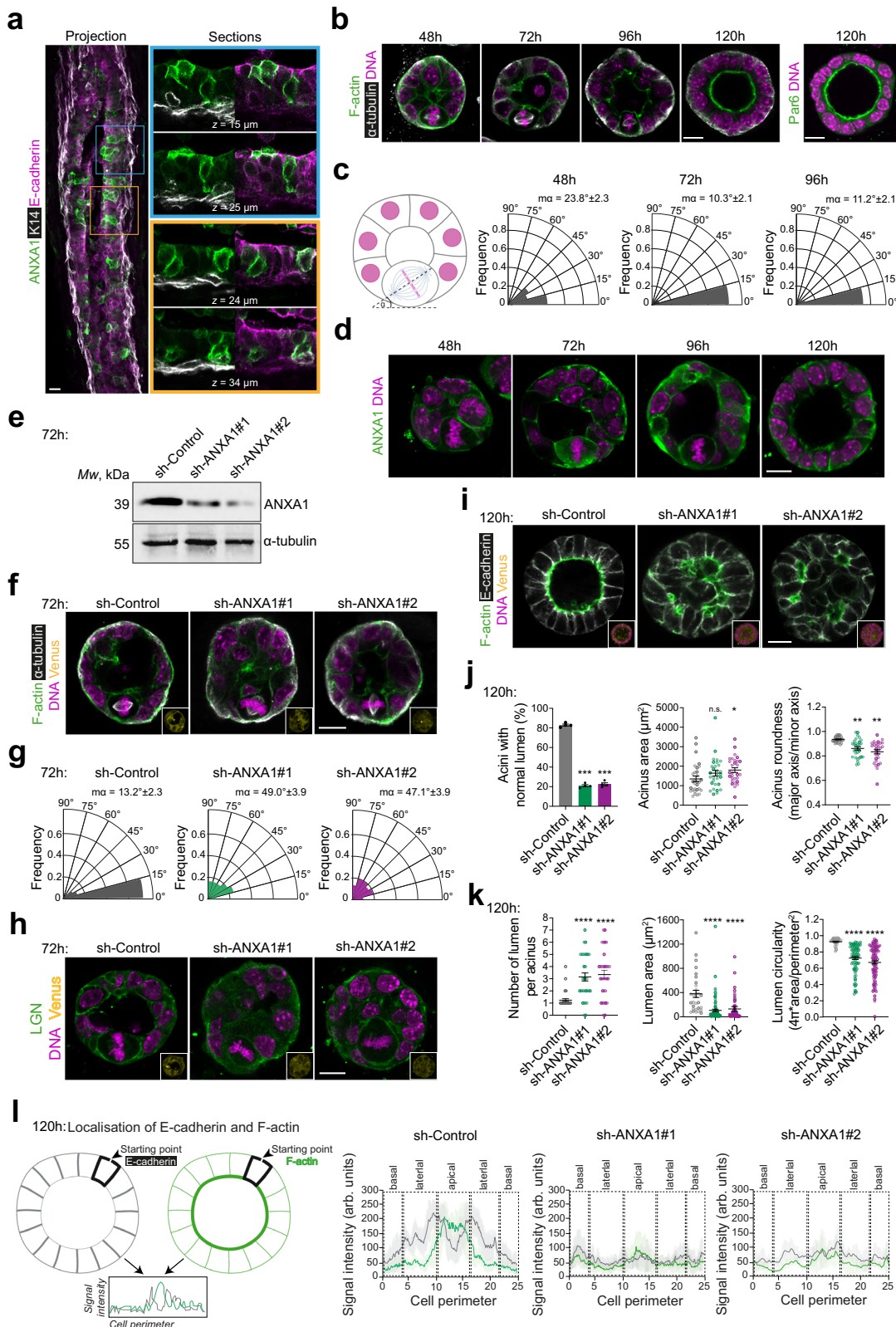

a key F-actin-binding protein that regulates cortical actin organisation and mechanics to ensure normal mitotic progression[64,82,83]. Afadin has been shown to bind directly and concomitantly to F-actin and LGN, providing a molecular link between the cell cortex and astral microtubules[13]. Future studies characterising the interactome of ANXA1 during mitosis will be key to uncover the molecular mechanisms linking ANXA1 to LGN and F-actin and understand how ANXA1

regulates the astral microtubule-cortical actin crosstalk to ensure balanced pulling forces that orient the mitotic spindle to its correct position.

Defects in LGN-NuMA cortical localisation and spindle orientation can be caused by incorrect metaphase plate formation and chromosome misalignment[52,53]. Our live imaging experiments reveal that ANXA1 depletion leads to incorrect chromosome segregation and

**Fig. 6 | ANXA1 is required for correct mitotic spindle orientation during 3D morphogenesis. a** Confocal images of representative *mouse* mammary gland cryosections (50 μm-thick) stained for ANXA1 (green), K14 (grey) and E-cadherin (magenta). **b** Confocal images of representative mMEC acini stained for F-actin (green) and α-tubulin (grey) or Par6 (green) and counterstained with DAPI (DNA, magenta). **c** Spindle angle frequencies and mean angles (*mα*) (3 independent experiments, 48 h: *n* = 42 acini; 72 h: *n* = 40 acini; 96 h: *n* = 39 acini). **d** Confocal images of representative acini stained for ANXA1 (green) and counterstained with DAPI (DNA, magenta). **e** Western blotting of extracts from shRNA-transduced acini. **f** Confocal images of representative shRNA-transduced acini (Venus, yellow) stained for F-actin (green) and α-tubulin (grey) and counterstained with DAPI (DNA, magenta). **g** Spindle angle frequencies and mean angles (mα) (3 independent experiments, sh-Control: *n* = 30 acini; sh-ANXA1#1: *n* = 30 acini; sh-ANXA1#2: *n* = 30 acini). Kolmogorov-Smirnov test, **P < 0.01. **h** Confocal images of representative shRNA-transduced acini (Venus, yellow) stained for LGN (green) and counterstained with DAPI (DNA, magenta). **i** Confocal images of representative shRNA-transduced acini (Venus, yellow) stained for F-actin (green) and E-cadherin (grey)

and counterstained with DAPI (DNA, magenta). **j** Percentage of acini with normal lumen (left) (4 independent experiments, sh-Control: *n* = 163 acini; sh-ANXA1#1: *n* = 142 acini; sh-ANXA1#2: *n* = 168 acini), acinus area (middle) and acinus roundness (right) (3 independent experiments, sh-Control: *n* = 30 acini; sh-ANXA1#1: *n* = 30 acini; sh-ANXA1#2: *n* = 30 acini). One-way ANOVA with Tukey's test, left: ***P = 0.0008 and ***P = 0.0005; middle: P = 0.201 and *P = 0.067; right: **P = 0.007 01 and **P = 0.002. **k** Number of lumen per acinus (left), lumen area (middle) and lumen circularity (right) (3 independent experiments, sh-Control: *n* = 36; sh-ANXA1#1: *n* = 32; sh-ANXA1#2: *n* = 31). One-way ANOVA with Tukey's test, left: ****P = 0.00009 and ****P = 0.00002; middle: ****P = 0.0001 and ****P = 0.00007; right: ****P = 0.0001 and ****P = 0.00006. **l** Left: illustration showing quantification of cortical fluorescence intensity of F-actin (green) and E-cadherin (grey) in acini. Right: average cortical fluorescence intensity profiles of E-cadherin and F-actin (3 independent experiments, sh-Control: *n* = 30; sh-ANXA1#1: *n* = 30; sh-ANXA1#2: *n* = 30). All time-points of 3D culture are indicated in hours (h). All data are presented as mean ± s.e.m. n.s. (not significant). arb. units (arbitrary units). All scale bars, 10 μm. Source data are provided as a Source Data file.

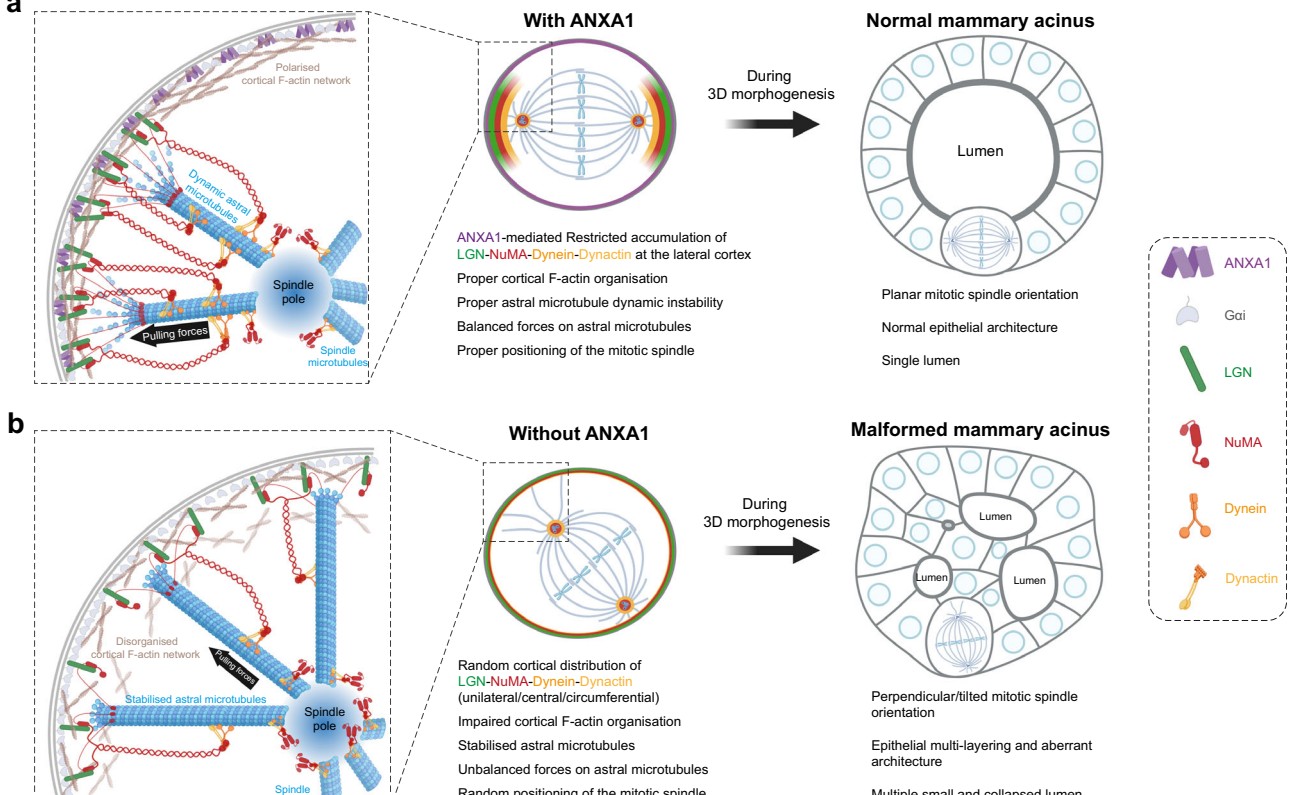

**Fig. 7 | Proposed model for ANXA1 regulation of planar mitotic spindle orientation and epithelial morphogenesis. a** Cortical ANXA1 interacts with LGN to instruct polarised accumulation of the LGN-NuMA complex at the lateral cortex. NuMA in turn, recruits Dynein-Dynactin generating pulling forces on the spindle poles that are balanced by ANXA1-dependent regulation of cortical F-actin and astral microtubule organisation, thereby ensuring planar alignment of the mitotic spindle. During 3D morphogenesis, ANXA1-dependent regulation of planar mitotic spindle orientation is crucial for single lumen formation and normal epithelial

architecture. **b** Loss of ANXA1 results in a diffuse cortical distribution of LGN and NuMA, which in turn impairs the recruitment of Dynein-Dynactin to the cell cortex. This affects the length and dynamic instability of astral microtubules as well as cortical F-actin integrity, generating unbalanced forces that result in randomised mitotic spindle orientation. During 3D morphogenesis, mitotic spindle misorientation upon ANXA1 knockdown results in epithelial multi-layering, leading to multi-lumen formation and aberrant epithelial architecture.

delay in mitotic progression, raising the question of whether the errors that we observe in chromosome segregation are independent of defects in mitotic spindle orientation. Increasing evidence supports that unbalanced astral microtubule-cortex crosstalk leads to defective chromosome dynamics[4,65]. Additionally, epithelial cell geometry and polarity have been shown to influence chromosome segregation fidelity[84]. Our cortical LGN interactome data identify several NuMA-interacting proteins that have been described to play pleotropic roles

in the regulation of astral microtubule stability, mitotic spindle positioning, and kinetochore-microtubule dynamics, including the DYNC1H1/Dynein-p150[Glued]/Dynactin complex[6,32–36,85], CLASP1[37–39], HAUS6[40,41], KPNA2/Importin-α[86], PLK1[6,31], TUBG1/γ-tubulin[41,42], EIF3E/Int-6[47], CCNB/CDK1[36,43], RAN[39,44,45]. These findings reinforce the idea that defects in astral microtubule organisation and spindle orientation upon ANXA1 depletion are likely to cause incorrect chromosome segregation. It will be interesting to explore further the molecular

mechanisms linking ANXA1-mediated regulation of astral microtubule dynamics to accurate chromosome segregation.

Regulation of the LGN-NuMA complex is essential to maintain the balance between planar and perpendicular divisions that ensures proper position and fate of daughter cells during epithelial morphogenesis and differentiation[3,4]. Our data in 3D cultures of mMECs demonstrate that ANXA1-mediated control of LGN cortico-lateral localisation and planar spindle orientation is required for normal luminogenesis and epithelial architecture. This is consistent with studies in MDCK, Caco-2 and hepatocytic HepG2 cells establishing LGN as a key regulator of planar cell division during cystogenesis[11,13,63,87,88]. We establish mMEC-derived acini as a useful 3D culture system to study the mechanisms linking planar spindle orientation to luminogenesis in the mammary epithelium. Our results showing that planar divisions do not contribute to cavitation in MCF-10A cells acini further corroborate previous studies establishing apoptosis as the mechanism that drives cavitation[69]. S1 mammary cells have been used to study the mechanisms of mitotic spindle orientation in 3D culture[89,90]. However, while S1 cells can form a mature apical membrane, their derived acini do not expand a lumen[91]. During MDCK 3D cystogenesis, early maturation of the apical membrane at the two-cell stage is required for the coupling of apical polarity to mitotic spindle orientation, and aPKC-dependent exclusion of LGN to the lateral cortex, ensuring planar cell division and luminogenesis[11,68]. Our experiments in mMEC acini suggest, however, that LGN lateral accumulation does not need a mature apical membrane, further sustaining our model establishing ANXA1 as a key polarity cue that instructs LGN restriction to the lateral cortex. Moreover, our data showing that apical polarity is established at d5 of 3D culture after cells have ceased to divide strongly suggest that luminogenesis is driven by planar cell divisions, independently of apical polarity. Interestingly, our proteomic data identify another established LGN-interacting complex, including the LGN adapter INSC and the apical polarity protein PAR3[3]. While these proteins are essential regulators of the apical localisation of LGN and asymmetric/perpendicular cell division in the mouse skin epidermis and neuroepithelium[3,4], a study in the mammary gland has shown that mInsc promotes symmetric/planar cell divisions[25], pointing to a context-dependent action of the mechanisms that regulate the localisation and function of the LGN-NuMA complex to adequately position cell fate determinants and daughter cells during symmetric/planar or asymmetric/perpendicular cell divisions. Given evidence showing that INSC competes NuMA from LGN[92], it will be interesting to investigate how ANXA1 regulates the interplay between LGN-NuMA and INSC-PAR3 and determine how this influences the switch between perpendicular and planar divisions in the differentiating mammary epithelium.

Our immunofluorescence analyses in hMEC acini show that while ANXA1 localises at the cell cortex, the protein is expressed in both polarised and non-polarised cells that comprise heterogeneous multi-layered acini, which rarely form a central lumen. The inherent morphological heterogeneity observed in hMEC 3D models that we and another study report[93], poses a challenge to the visualisation of OCDs and the characterisation of their relative contribution to *human* mammary epithelial morphogenesis. Novel culture methods will be needed to assess the functional requirement of ANXA1 in mitotic spindle orientation during hMEC cystogenesis. Our in situ experiments reveal that ANXA1 marks a subset of luminal cells in the *mouse* mammary gland, whereas in *human* the protein is expressed in both the luminal and basal compartments, further reflecting the intrinsic cellular differences between the *mouse* and *human* mammary epithelia. Commonly used *mouse* cellular lineage markers, such as K5 and K14, have already been described as bimodal in the *human* breast[94]. Further studies will be necessary to determine whether ANXA1 has distinct or additional functions in the *human* mammary epithelium from those that we here characterise in *mouse*.

The balance between symmetric and asymmetric division is broken in breast cancer[3,24,26,27,95]. Yet, it remains unknown how incorrect mitotic spindle orientation can initiate breast tumorigenesis. Studies in the normal prostate epithelium have shown that depletion of E-cadherin affects the localisation of the LGN-NuMA complex and randomises mitotic spindle orientation, resulting in prostatic hyperplasia that progresses to invasive adenocarcinoma[14]. Thus, defects in OCDs can be envisioned as a key initiating oncogenic event. Our 3D culture experiments show that depletion of ANXA1 leads to epithelial multi-layering, resulting in aberrant epithelial architecture and multiple off-centred small or collapsed lumens, reminiscent of the cribriform growth pattern that is often observed in breast ductal carcinoma in situ (DCIS)[96]. Indeed, cell division-driven epithelial multi-layering where cells escape from the epithelial sheet has been proposed to represent one of the early steps in tumorigenesis[97]. ANXA1 is well established as a biomarker of breast cancer[28]. While the underlying mechanisms remain poorly understood, studies have shown that ANXA1 is downregulated in DCIS whereas it is upregulated in the most aggressive and metastatic breast cancer subtypes[28]. It will be particularly interesting to investigate the functional requirement of ANXA1-mediated mitotic spindle orientation for proper mammary epithelial integrity and differentiation in vivo. In parallel, it will also be highly informative to learn more about the fate and dynamics of ANXA1-expressing cells and their relative contributions to mammary gland development and homeostasis, but also to tumour formation and heterogeneity.

## Methods

### Mice
All animal experiments were performed in accordance with Home Office (UK) and to the principles of the 3Rs (Replacement, Refinement and Reduction) and ARRIVE (Animal Research: Reporting of In Vivo Experiments) guidelines. The experiments and protocols are approved by the University of Southampton Local Ethics Committee and registered in the Ethics and Research Governance Online II (ERGO II; ID: 65385). Mice were housed in a specific pathogen-free facility in individually ventilated cages in an ambient temperature- and humidity-controlled room with a 12 h light/12 h dark cycle under standard housing conditions with continuous access to food and water.

### Human breast tissue collection
For experiments using *human* breast tissues from healthy donors, ethical approval was not required as these samples were obtained upon informed voluntary donor consent followed by complete anonymisation. *Human* breast tissues were anonymously retrieved from healthy donors who underwent voluntary cosmetic reduction surgery at Mediclinic (Oud-Heverlee, Belgium) in the absence of any medical indication. All donors provided written informed consent for the use of the resection material and publication of the experimental results. Tissue donors were adequately informed by third party clinicians about the use of the tissue. Limited and fully anonymous donor information (age, parity status, familial cancer risk) was obtained without any link to the donor's medical record. Sample ID numbers and donor anonymised information are reported in Supplementary Dataset 3.

### Cell lines and transfections
MCF-10A, a spontaneously immortalised, nontransformed *human* mammary epithelial cell (MEC) line (ATCC® #CRL-10317) was cultured in DMEM/F12 (Invitrogen) supplemented with 5% donor horse serum (Gibco), 20 ng/ml EGF (Sigma #E9644), 10 μg/ml insulin (Sigma #I1882), 1 ng/ml cholera toxin (Sigma #C8052), 100 μg/ml hydrocortisone (Sigma #H0888), 50 U/ml penicillin, and 50 μg/ml streptomycin (Life Technologies), at 37 °C in a humidified 5% $CO_2$ atmosphere. Cells were transfected with a final concentration of 50 nM si-RNAs

using Lipofectamine RNAiMAX (Invitrogen), according to the manufacturer's protocol. 24 h post-transfection, cells were left for 48 h in culture to polarise and establish cell-cell adhesions, before they were lysed or fixed and immunoprocessed.

HEK293 cells (ATCC® #CRL-3216) provided by Melissa Andrews (University of Southampton), were cultured in DMEM supplemented with 10% FBS (Gibco), 50 U/ml penicillin, and 50 µg/ml streptomycin (Life Technologies), at 37 °C in a humidified 5% $CO_2$ atmosphere. Cells were transfected using standard calcium phosphate transfection protocol[98].

### Primary mammary epithelial cell 3D culture

*Mouse* primary MECs (mMECs) were collected from wild-type C57Bl/6 15.5- and 16.5-day pregnant female (wild-type: *n* = 6; sh-Control: *n* = 6; sh-ANXA1#1: *n* = 6; sh-ANXA1#2: *n* = 6) and cultured as described in ref. [70]. Briefly, once mechanically dissociated, mammary fat pads were digested (90 min, 37 °C) in $CO_2$-independent medium (Invitrogen) containing 5% FBS, 3 mg/ml collagenase A (Roche Diagnostics #10103586001) and 100 U/ml hyaluronidase (Sigma #H3884). Cells were resuspended in 0.25% trypsin-EDTA (1 min), and then in 5 mg/ml dispase (Roche Diagnostics #4942078001) with 0.1 mg/ml DNase I (Roche Diagnostics #11284932001) (5 min). Red blood cells were lysed with 0.17 mM $NH_4Cl$. Cells ($2 \times 10^4$) were plated in each well of 8-well LabTekII chamber slides (Fisher Scientific) pre-coated with Phenol-free Growth Factor Reduced 100% Matrigel (25 µl per well, BD Biosciences) and cultured in DMEM-F12 supplemented with 2% Matrigel, 10% FBS (Gibco), 3 ng/ml EGF (PeproTech #315-09), 1% Glutamine (Sigma), 1 µg/ml Hydrocortisone (Sigma), 10 µg/ml Insulin (Sigma) and 1% Penicillin/Streptomycin (Life Technologies). Cells were grown onto Matrigel to form acini for 2–5 days and fed every 2 days.

*Human* MECs (hMECs) were obtained from cryopreserved breast tissue fragments derived from mammary reduction surgeries from healthy donors under informed consent at Mediclinic (Belgium) as described in ref. [99]. Briefly, cryosections were thawed at 37 °C and washed with DMEM-F12 supplemented with antibiotics. After centrifugation at 350 x *g* for 5 min using DMEM-F12, breast fragments were sheared into smaller fragments using pre-warmed 0.15% Trypsin incubation for 10–15 min at 37 °C. hMEC 3D acini were resuspended in 10 mg/ml Cultrex Reduced Growth Factor Basement Membrane Extract (BME) Type 2 (R&D Systems #3532-005-02) and 25 µl BME domes were plated in each well of µ-Slides 8 (Ibidi). Culture tissue plates were incubated upside down and allowed to polymerise for 30 min at 37 °C. 200 µl of hMECs 3D culture medium (Mammary Epithelial Cell Media, Promocell) supplemented with 0.5% FBS, 3 µM Y-27632 (Sigma #Y0503), Forskolin (Sigma #F6886), 10 ng/ml Heregulin-β–1 (Peprotech #100-03) and 10 ng/ml Amphiregulin (Peprotech #100-55B) were added to each well. Y-27632 and FBS were removed from culture medium after 5 days of culture. Culture medium was refreshed every 3–4 days.

### Constructs, siRNAs and shRNAs

Retroviral plasmids were used to transduce MCF-10A cells. pTK14-GFP-LGN plasmid was obtained from Addgene (Plasmid #37360). pTK-GFP was cloned as follows: The full-length eGFP sequence from pTK14 plasmid was amplified by PCR and inserted into the pTK14 plasmid, where the GFP-LGN sequence was removed by restriction digestion with Bsu36i and PspOMI (New England Biolabs). The primers were 5′-CCGACCTGAGGAAGGGAG-3′ (forward) and 5′-ACAGCGGGGCCCTTACTTGTACAGCTCGTCCATGCC-3′ (reverse). To clone pTK-EB3-GFP, the EB3 sequence was synthesised (Eurofins) and inserted into pTK-GFP digested by AgeI and SacII restriction enzymes (New England Biolabs). To clone pTK-ANXA1-mCherry, the ANXA1-mCherry sequence was synthesised (Eurofins) and inserted into pTK93-Lifeact-mCherry vector (Addgene, Plasmid #46357) where Lifeact-mCherry was removed by restriction digestion with NaeI and ApaI (New England

Biolabs). pTK-mCherry was cloned as follows: The full-length mCherry sequence was amplified by PCR and inserted into pTK93-Lifeact-mCherry vector where the Lifeact-mCherry sequence was removed by restriction digestion with BamHI and SalI (New England Biolabs). The primers were 5′-AATTGGATCCGCCACCATGGTGAGCAAGGGCGAG-3′ (forward) and 5′-CTGACACACATTCCACAGGGTCG-3′ (reverse). For all plasmids, correct insertions were verified by Sanger sequencing.

Transient knockdown of *human* ANXA1 or LGN in MCF-10A cells was achieved by transfection of MISSION® Predesigned siRNAs from Sigma. The following products were used: SASI_Hs01_00157996 (si-ANXA1#1), SASI_Hs01_00157997 (si-ANXA1#2) for ANXA1 knockdown; SASI_HS01_00121831 (si-LGN#1), SASI_HS01_00121832 (si-LGN#2) for LGN knockdown. siGENOME RISC-Free® (Dharmacon) was used as a negative control (si-Control).

Stable knockdown of mouse ANXA1 in mMECs was performed by small-hairpin RNAs (sh-RNAs). shRNA sequences were annealed and cloned into the pLKO.4-Mem-Venus lentiviral vector, provided by David Bryant (CRUK Beatson Institute University of Glasgow). The following sh-RNA sequences were used: 5′-CCGGGCTTTGGCAGATAAGTCTAATCTCGAGATTAGACTTATCTGCCAAAGCTTTTTG-3′ (sh-ANXA1#1 forward) and 5′-CCGGCCGTTCGGAAATTGACATGAACTCGAGTTCATGTCAATTTCCGAACGGTTTTTG-3′ (sh-ANXA1#1 reverse); 5′-AATTCAAAAAGCTTTGGCAGATAAGTCTAATCTCGAGATTAGACTTATCTGCCAAAGC-3′ (sh-ANXA1#2 forward) and 5′-AATTCAAAAACCGTTCGGAAATTGACATGAACTCGAGTTCATGTCAATTTCCGAACGG-3′ (sh-ANXA1#2 reverse). As a negative control, the following sequences were used: 5′-CCGGCCTAAGGTTAAGTCGCCCTCGCTCGAGCGAGGGCGACTTAACCTTAGGTTTTTG-3′ (sh-Control forward) and 5′-AATTCAAAAACCTAAGGTTAAGTCGCCCTCGCTCGAGCGAGGGCGACTTAACCTTAGG-3′ (sh-Control reverse).

### Lentivirus and retrovirus production and infection

To generate stable MCF-10A cell lines, plasmids were introduced by retroviral transduction. Retroviruses were prepared in HEK293 cells by calcium phosphate co-transfection of 10 µg retroviral plasmid, 6.5 µg envelope plasmid pCMV-VSV-G (Addgene, Plasmid #8454) and 5 µg packaging plasmid pUMVC (Addgene, Plasmid #8449). Virus particles were collected 48 h after transfection, filtered (45 µm) and used to infect MCF-10A cells in the presence of 8 µg/ml polybrene (Sigma). Clones of interest were selected using 2 µg/ml blasticidin (Sigma) or 1 µg/ml puromycin (Sigma). To further enrich for positive cells, selected cells were subjected to flow-cytometry to sort for fluorescent-positive cells (see gating strategy in Supplementary Information).

Knockdown of ANXA1 in mMECs was achieved by lentiviral transduction. Lentiviruses were generated similarly to retrovirus production by co-transfecting 10 µg lentiviral plasmid, 10 µg envelope plasmid pCMV-VSV-G and 10 µg packaging plasmid (pPAX2). pPAX2 plasmid was provided by David Bryant (CRUK Beatson Institute, University of Glasgow). Freshly isolated mMECs were transduced as described in ref. [100], and cultured in 8-well slide chambers pre-coated with Matrigel to form acini until immunoprocessing.

### Drug treatment

Drugs were dissolved in DMSO and kept at −20 °C as 2 mM or 10 mM stock solutions. To synchronise MCF-10A cells in G2/M phase, cells were treated with 9 µM RO-3306 (Sigma #SML0569) for 18 h to allow CDK1 inhibition. To further arrest cells in metaphase, MCF-10A cells were released from the G2/M block by washing three times with pre-warmed drug-free medium and immediately treated with 10 µM proteasome inhibitor MG-132 (Sigma #474787) for 6 h. Cells were washed twice with PBS before processing for experiments. To inhibit TRPM7 kinase activity, MCF-10A cells were treated with 20 µM TG100-115 (Stratech #A2754) for 2 h before immunofluorescence. To depolymerise F-actin, cells were treated with 1 µM latrunculin A (Sigma #L5163) for 30 min before immunofluorescence.

## Cell extracts and immunoblotting

MCF-10A cells and MEC 3D acini were lysed in NP-40 buffer [50 mM Tris, pH 7.4, 250 mM NaCl, 5 mM EDTA, 50 mM NaF, 1 mM Na3VO4, 1% Nonidet P40 (NP40)], supplemented with protease inhibitor cocktail (Sigma, #P2714). MEC acini were first treated with trypsin 0.25% for 30 min to break the Matrigel, then washed with PBS1X and resuspended in NP40 lysis buffer, containing protease inhibitor cocktail, and centrifuged at 21,100 x $g$ for 10 min at 4 °C. Protein concentration of lysates was determined using Pierce™ BCA Protein Assay (ThermoScientifc) and protein were subjected to SDS-PAGE and subsequent Western blot analysis. The following primary antibodies were used: anti-α-tubulin DM1A (0.2 µg/ml, Sigma #T6199), anti-LGN (1:500, Sigma #ABT174), anti-GFP (2 µg/ml, Invitrogen #A-11122), anti-phospho-Histone3 (0.2 µg/ml, Sigma #06-570), anti-ANXA1 (0.6 µg/ml, Proteintech, #55018-1-AP), anti-mCherry (1:1000, Abcam #ab167453). Secondary antibodies conjugated to horseradish peroxidase (Invitrogen, #32430, #32460) were used at 1:10,000. Protein bands were visualised using SuperSignal™ West Femto Substrate (Thermo Scientific) on a Syngene PXi detection system (Syngene).

## Immunoprecipitation and sample preparation for mass spectrometry

MCF-10A cells were plated in 15 cm dishes and washed twice with ice-cold PBS before protein extraction. About $10 \times 10^7$ cells were lysed in a mild lysis buffer [50 mM Tris, pH 7.4, 150 mM NaCl, 0.5 mM EDTA, 10 mM NaF, 1 mM Na3VO4, 0.5% Nonidet P40 (NP40)], containing protease inhibitor cocktail. Cell lysates were cleared by centrifugation at 17,000 x $g$ for 30 min at 4 °C. Co-immunoprecipitation was performed using a GFP- or RFP-Trap Kit (Chromotek, #gtma-20, #rtma-20) as per manufacturer's instructions. For immunoblotting, washed beads were eluted by boiling in Laemmli sample buffer (Bio-Rad) containing 5% 2-mercaptoethanol. For mass spectrometry analysis, washed beads were eluted by on-bead tryptic digestion as described in refs. [101,102]. Briefly, proteins on the beads were partially denatured with a buffer containing 2 M urea in 50 mM Tris, pH 7.5, 1 mM DTT. Beads were further treated with a buffer containing 2 M urea in 50 mM Tris, pH 7.5, 5 mM iodoacetamide (IAA) and proteins were eluted by digestion with 5 µg/ml trypsin (Promega). Eluted proteins were fully digested overnight. C18 reverse phase clean-up was performed on the tryptic peptides using a Waters Oasis C18 plate (Waters, UK).

## Mass spectrometry and data analysis

Peptide extracts were separated on an Ultimate 3000 RSLC nano system (Thermo Scientific) using a PepMap C18 EASY-Spray LC column, 2 µm particle size, 75 µm x 75 cm column (Thermo Scientific) over a 140 min (single run) linear gradient of 3–25% buffer B (0.1% formic acid in acetonitrile (v/v)) in buffer A (0.1% formic acid in water (v/v)) at a flow rate of 300 nL/min. Peptides were introduced using an EASY-Spray source at 2000 V to a Fusion Tribrid Orbitrap mass spectrometer (Thermo Scientific). Full MS spectra were recorded from 300 to 1500 $m/z$ in the Orbitrap at 120,000 resolution with an automatic was performed using TopSpeed mode at a cycle time of 3 s. Higher-energy collisional dissociation (HCD) fragmentation was induced at an energy setting of 28 for peptides with a charge state of 2–4. Fragments were analysed in the orbitrap at 30,000 resolution. Analysis of raw data was performed using Proteome Discoverer software (Thermo Scientific) and the data processed to generate reduced charge state and deisotoped precursor and associated product ion peak lists. These peak lists were searched against the *human* Uniprot KB database protein database (42,186 entries, 2018-05-18). A maximum of one missed cleavage was allowed for tryptic digestion and the variable modification was set to contain oxidation of methionine and N-terminal protein acetylation. Carboxyamidomethylation of cysteine was set as a fixed modification. The false discovery rate (FDR) was estimated with randomised decoy database searches and were filtered to 1% FDR. Proteins present in the control were removed and further contaminants were also removed by searching against the Contaminant Repository of Affinity Purification (CRAPome) database[30]. The interactions network was constructed with the online STRING protein-protein interaction (PPI) database (v. 11)[103] using an interaction confidence score of ≥ 0.6. The PPI networks were visualised and analysed using Cytoscape software (version 3.8.2)[104]. Gene enrichment analyses were performed to identify Gene Ontology (GO) biological process as well as Reactome and KEGG (Kyoto Encyclopedia of Genes and Genomes) pathways associated with the PPI networks using ClueGO plugin of Cytoscape (version 2.5.7)[105].

## Immunofluorescence

The following primary antibodies were used: anti-LGN (1:200, Sigma #ABT174), anti-NuMA (1:100, Novus Biologicals #NB500-174), anti-p150$^{Glued}$ (2.5 µg/ml, BD Biosciences #610473), anti-γ-tubulin AK-15 (1:300, Sigma #T3320), anti-G$_{\alpha i1}$ (10 µg/ml, Santa Cruz #sc-56536), anti-ANXA1 (10 µg/ml, Proteintech #66344-1-Ig; 3 µg/ml, Proteintech #21990-1-AP), anti-S100A11 (10 µg/ml, Proteintech #60024-1-Ig), anti-α-tubulin DM1A (1 µg/ml, Sigma #T6199), anti-E-cadherin (20 µg/ml, Invitrogen #13-1900), anti-Par6 (1 µg/ml, Santa-Cruz #sc-166405), anti-α-Smooth muscle actin (α-SMA) (10 µg/ml, Sigma #A5528), anti-keratin 8 (K8) (5 µg/ml, Sigma #MABT329), anti-K14 (1:300, Origene #BP5009) and anti-GFP (6 µg/ml, Sigma #11814460001). AlexaFluor555 phalloidin (1:100, Life Technologies #A34055) was used to label F-actin. Secondary antibodies (Life Technologies) used were *goat* anti-*mouse* (#A-32723), anti-*rabbit* (#A-11037 and #A-11008), anti-*rat* (#A-11007) and anti-*guineapig* (#A-21450) conjugated to AlexaFluor488, AlexaFluor594 or AlexaFluor647, at 5 µg/ml.

To visualise LGN, NuMA, p150$^{Glued}$, ANXA1, GFP and γ-tubulin, MCF-10A cells were fixed with anhydrous methanol at −20 °C for 10 min followed by permeabilization with 0.1% Triton X-100 in PBS for 2 min. Alternatively, cells were fixed with 10% trichloroacetic acid (TCA, Sigma #T6399) for 7 min on ice followed with anhydrous methanol at −20 °C for 10 min. Cells were washed three times before permeabilization with 0.1% Triton X-100 in PBS for 2 min. G$_{\alpha i1}$ and α-tubulin were visualised by fixing cells with 4% paraformaldehyde (PFA) for 20 min at RT followed by permeabilization with 0.5% Triton X-100-PBS for 10 min. For all fixation methods, cells were blocked with 3% BSA in 0.1% Triton X-100 in PBS for 1 h at RT and immunostained with primary antibodies overnight at 4 °C. Cells were washed and incubated with appropriate secondary antibodies for 1 h at RT.

To visualise astral microtubules, MCF-10A cells were fixed with 3% PFA, containing 0.25% glutaraldehyde and 0.2% NP-40 in Brinkley buffer 1980 [(BRB80) 80 mM PIPES, 1 mM MgCl2 hexahydrate, 1 mM EGTA] for 1 min, followed by incubation with 3% PFA and 0.25% glutaraldehyde in BRB80 for 10 min and with 0.1 M NH₄Cl in BRB80 for 10 min. Cells were washed twice for 5 min with BRB80, permeabilised with 0.5% Triton X-100 in PBS for 10 min, washed again twice in BRB80 and finally blocked in 3% BSA and 0.2% NP-40 in BRB80 for 1 h.

For all MCF-10A cell immunostaining, cells were counterstained with 2.5 µg/ml DAPI (Sigma) and mounted with Vectashield antifade mounting medium (Vector Laboratories).

mMEC-derived acini were fixed with 4% PFA for 30 min and permeabilised with 0.5% Triton X-100 in PBS for 10 min. Acini were blocked with 1% BSA in PBS and 0.1% Triton X-100 for 2 h and then incubated with ANXA1, LGN, α-tubulin, E-cadherin primary antibodies overnight at 4 °C. Then, acini were immunostained with *mouse, rabbit,* and *rat* AlexaFluor488-, AlexaFluor594- or AlexaFluor647-conjugated secondary antibodies for 2 h at RT. Acini with F-actin staining at the apical surface of cells surrounding a single lumen were identified as acini with normal lumens. Cells were counterstained with DAPI-containing Fluoroshield (Sigma).

hMEC-derived acini were fixed at day 10 in 2% PFA for 30 min and washed in 1X PBS three times for 10 min. Permeabilization was

performed with 0.5% Triton-100X in 1X PBS for 1 h at RT. Acini were then incubated with 0.5% Triton-blocking buffer (5% FBS and 2% BSA in 1X PBS) for 2 h at RT. Next, acini were incubated at RT overnight with ANXA1, K8 and Par6 primary antibodies, following washing and incubation with *rabbit*, *rat* and *mouse* AlexaFluor488-, AlexaFluor555-, AlexaFluor647-conjugated secondary antibodies and DAPI for 2 h at RT and mounted with Vectashield antifade mounting medium.

*Mouse* mammary gland sections (50 μm thick) were cut and air-dried for 30 min, then fixed with 4% PFA for 20 min at RT. Following 45 min permeabilization with 0.2% Triton X-100 in PBS, slides were blocked for 2 h in 2% BSA, 5% foetal bovine serum (FBS), 0.2% Triton X-100 in PBS. Sections were incubated at 4 °C overnight in ANXA1, E-cadherin and K14 primary antibodies, washed and incubated with *rabbit*, *rat* and *guineapig* AlexaFluor488-, AlexaFluor594- and AlexaFluor647-conjugated secondary antibodies for 2 h at RT. Sections were counterstained with DAPI-containing Fluoroshield.

Freshly dissected *human* breast tissue pieces derived from mammary reduction surgeries from healthy donors were washed in 1X PBS and fixed at 4 °C with 2% PFA for 48 h. Next, fixed breast samples were impregnated with 30% Sucrose solution in 1X PBS for at least 48 h and embedded in OCT (VWR, #361603E). 70 μm-thick sections were incubated with 0.5% Triton-blocking buffer (5% FBS and 2% BSA in 1X PBS) for 2 h at RT. Sections were incubated at RT overnight with ANXA1, E-cadherin and SMA primary antibodies, following washing and incubation with *rabbit*, *rat* and *mouse* AlexaFluor488-, Alexa-Fluor555-, AlexaFluor647-conjugated secondary antibodies and DAPI for 2 h at RT and mounted with Vectashield antifade mounting medium.

## Proximity ligation assay

To detect proximal association of ANXA1 and LGN, the Duolink™ proximity ligation assay (PLA) (Sigma, #DUO92102) was used according to the manufacturer's protocol. Briefly, MCF-10A cells expressing GFP-LGN were fixed and stained as described in the standard immunofluorescence protocol above. Samples were incubated with ANXA1 and GFP primary antibodies overnight at 4 °C. Negative controls were performed by incubating cells with one of the two primary antibodies only. The positive control was performed by visualising the co-localisation of GFP-LGN and LGN. Cells were incubated with Duolink PLA probes for 2 h, washed and incubated with the ligation mixture for 30 min, washed again and incubated with the amplification mixture for 2.5 h. Cells were counterstained with DAPI-containing Fluoroshield.

## Quantitative confocal microscopy

Immunofluorescence images were captured with an inverted Leica TCS SP8 inverted laser scanning microscope (Leica Microsystems) using x40 oil immersion (40x HC Plan/Apo CS2 1.30 numerical aperture (NA)) and x63 glycerol immersion (63x HC Plan/Apo CS2 1.30 NA) objectives. Z-stacks at 16-bit depth and 2048 × 2048 pixels were collected at 0.2 or 0.3 μm (MCF-10A), 0.5 μm (mMECs) and 1 μm (mammary sections) intervals. Images were processed with Fiji software[106]. Astral microtubule images were denoised using the MATLAB-based ND-Safir software[107]. hMEC acini and *human* breast tissue images were collected at 12-bit depth with 1024 × 1024 pixels per tile and then processed using the LasX 3D visualisation software module (Leica Microsystems), and signal intensities were linearly adjusted for visualisation purposes.

ANXA1, LGN and $G_{\alpha i1}$ profiles at the cell cortex of metaphase cells were measured in Fiji using a custom macro as described in ref.[108]. Fluorescence intensity values were reported along the cortex starting (and finishing) from a point facing the metaphase plate. Position 0 and 180 face the chromosome plate (central cortex) and position 90 and 270 face spindle poles (lateral cortex): 180 positions were scanned (every 2°) along the cortex. Profiles of ANXA1 and LGN at the cell cortex of prometaphase cells were measured similarly in Fiji with a custom

macro (see Supplementary Information). To account for more elongated cell shapes, a line was drawn along the cell contour using the Freehand tool and the macro fits an ellipse to this contour. Fluorescence intensities were calculated along a 30 pixel-long radial line overlapping the cortex and the maximum intensity was reported. The whole cell contour was scanned at 180 successive positions (every 2°) starting (and finishing) at the ellipse's short axis assuming the metaphase plate will align with the long axis of the ellipse. To compare signals from ANXA1 and LGN in prometaphase and metaphase cells, intensity values were corrected by subtracting background measurements. Average profiles were calculated from individual profiles.

Astral microtubule intensity (α-tubulin signal) was measured in Fiji. Maximum projections of images were generated and the fluorescence intensity of the whole cell ($I_{total}$) and the spindle ($I_{spindle}$) excluding spindle poles with astral microtubules were measured. Background signal was subtracted, and the relative fluorescence intensity of astral microtubules ($I_{astral, rel}$) was calculated as $I_{astral, rel} = (I_{total} - I_{spindle})/I_{spindle}$. Length of astral microtubules and EB3-GFP comets were measured by drawing a line along the astral microtubule and EB3-GFP comet extending towards the cell cortex on both sides of the poles. Similarly, the pole-to-cortex distance was measured by drawing a line towards the closest cell cortex in line with the spindle axis. Ratios were calculated by dividing the longer length or distance by the shorter length or distance.

To measure relative fluorescence intensities at the cell cortex and cytoplasm, a 30-pixel line was drawn across the lateral surface and the cytoplasm using Fiji software. The line scan function of Fiji was used to reveal the relative fluorescence intensity across the line.

Ratio of cortical-to-cytoplasmic F-actin fluorescence intensities was measured in Fiji. Maximum projections of images were generated and the fluorescence intensity of the whole cell ($I_{total}$) and the cytoplasm ($I_{cytoplasm}$) were measured. Background signal was subtracted, and the cortical-to-cytoplasm fluorescence intensity ratio of F-actin ($I_{cortex}/I_{cytoplasm}$) was calculated as $I_{cortex}/I_{cytoplasm} = (I_{total} - I_{cytoplasm})/I_{cytoplasm}$.

Quantifications of number of lumen and shape descriptors (circularity, area/size, roundness), in mMEC acini were performed using Fiji. Acini and lumen were outlined using the Fiji polygon selection tool, followed by shape descriptors extraction. Fluorescence intensities of E-cadherin and F-actin at the cell cortex were analysed using a semi-automated Fiji macro (Cell-o-Tape)[109]. Cell cortex were manually outlined using the Fiji segmented line tool in the E-cadherin channel, and signal intensities of the cellular perimeter were measured. Using the same perimeter, F-actin intensities were measured for the same cell. Graphic representation of both fluorescence intensities was performed by averaging the intensities of all cells per acinus.

Mitotic spindle orientation was measured using Fiji in metaphase cells stained for γ-tubulin (MCF-10A cells) or α-tubulin (acini). In MCF-10A, the spindle axis was defined by drawing a 30-pixel wide line across both spindle poles and repositioned along the z axis. The spindle axis angle was measured in respect to the substratum using the angle tool. Similarly, the spindle axis angle in acini was measured in respect to the plan of the basement membrane.

## Quantitative live cell imaging

For live cell imaging, MCF-10A or GFP-LGN-expressing MCF-10A cells were plated in glass bottom dishes (Nunc). Prior to imaging, cells were incubated in cell culture medium supplemented with 100 ng/ml Hoechst 33342 (Sigma) for 30 min. When mitotic spindles were observed, cells were further incubated in cell culture medium supplemented with 100 nM SiR-tubulin (Spirochrome #251SC002) for 3 h. Cells were imaged at 37 °C in $CO_2$ independent medium (Gibco) using a DeltaVision Elite microscope (GE Healthcare) coupled to a sCMOS max chip area 2048 × 2048 camera (G.E. Healthcare). For each recording,

image stacks of 30 to 40 planes at 0.6 μm increments were acquired using a PlanApo 60x/1.42 Oil immersion objective (Olympus) with 2 × 2 binning. Images were taken at 3 stage positions every 2.5 or 3 min for 3 or 5 h. Exposure times were 250 msec and 5% laser power for GFP, 80 msec and 2% laser power for labelled DNA and 80 msec and 2% laser power for labelled microtubules using the DAPI-FITC-mCh-Cy5 filter set. Images were deconvolved using the DeltaVision software SoftWoRx and further processed using Fiji.

Mitotic spindle oscillations were calculated from time-lapse videos of MCF-10A cells labelled with SiR-tubulin and Hoechst 33342. Metaphase spindle angles in the z and xy plane were manually determined for every frame using Fiji. Measurements of the spindle angle axis in the z-plane were performed as described above in fixed cells. Spindle angles were reported as positive values unless spindle poles changed direction. In this case, angles were displayed as negative values. Mitotic spindle orientation in the xy plane was measured by drawing a line crossing both spindle poles of the first frame of metaphase. This line was embedded to all the following frames to mark the initial position of the mitotic spindle. In the next frame, another line was overlaid across the poles to define the new spindle axis. The angle between the initial spindle position and the current spindle axis was measured using the angle tool. Spindle movement directions were considered by reporting spindle rotations clockwise as positive angles and reverse movements as negative angles. Spindle angle deviations >10° between two frames were counted as oscillation events. The oscillation index was determined as the percentage of oscillation events in respect to the total number of frames.

Metaphase plates were tracked using IMARIS software (Bitplane). First, a maximum intensity z-projection was created. The DNA metaphase plates were segmented using the Surface module and the centre of the DNA was marked with a point. The movement of this centre point was then tracked automatically over time starting from the first frame of metaphase to the last before anaphase. Movement tracks in the xy plane were reported. Displacement length was the length of DNA displacement from starting point. Velocity was calculated by the total track length divided by tracking time (metaphase plate congression time).

### Statistical analysis

Most of the experiments were repeated at least three times and the exact n is stated in the corresponding figure legend. Statistical significance of the overrepresentation of enriched proteins in biological processes and pathways were analysed with the ClueGO App (version 2.5.7) using hypergeometric test (Fisher's Exact Test) and the Bonferroni step-down correction method for multiple testing. Bar graphs of enrichment analyses were created using R statistical software (version 3.6.3) and RStudio (version 1.2.1335). All other statistical analysis were performed with GraphPad Prism 9.0 software. Multiple groups were tested using analysis of variance (ANOVA) with post hoc Tukey test, and comparisons between two groups were performed using t-tests. Data are shown as mean ± standard error of the mean (s.e.m.). $P \leq 0.05$ was considered statistically significant. Asterisks indicate levels of significance ($*P \leq 0.05$; $**P \leq 0.01$; $***P \leq 0.001$; $****P \leq 0.0001$).

### Reporting summary

Further information on research design is available in the Nature Portfolio Reporting Summary linked to this article.

## Data availability

The raw mass spectrometry proteomic data generated in this study have been deposited in the ProteomeXchange Consortium via the PRIDE partner repository[29] under accession code "PXD027452". The processed proteomic datasets are available in Supplementary Dataset 1 and Supplementary Dataset 2 files within the paper. Swiss-Prot human protein database used in this study is available at UniProt "UP000005640". CRAPome database (version 2.0) used in this study is available at (https://reprint-apms.org/?q=wk_1_1_search). STRING protein-protein interaction (PPI) database (version 11) used in this study is available at (https://string-db.org/cgi/input?sessionId= bCCp1b58NbtT&input_page_active_form=multiple_identifiers). All other relevant data supporting the key findings of this study are available within the article and its Supplementary Information files or from the corresponding author upon reasonable request. Details on the human breast samples used in this study are included in Supplementary Dataset 3. The source data that support the findings in all Figures and Supplementary Figures are provided as a Source Data file within the paper. All reagents generated in this study are available from the corresponding author upon reasonable request. Source data are provided with this paper.

## Code availability

The Fiji custom macro code generated in this study is provided in the Supplementary Information file within the paper.

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

## Acknowledgements

We would like to thank Dr. Mark Willett at the Imaging and Microscopy Centre for valuable assistance with fluorescence microscopy; Andrew Crocker for *mouse* husbandry; Dr. Melissa Andrews for providing the HEK293 cell line; Dr. David Bryant for providing the pLKO.4-Mem-Venus lentiviral plasmid. We would like to thank Wim De Maerteleire, MD for kindly providing us with *human* healthy breast material from healthy women undergoing reduction mammoplasty. In addition, we thank Dr. Marcin Przewloka for critically reviewing the manuscript. Cartoons in Figs. 1g, 2a, 2f, 3d, 4m, 5c–e, 5g, 6c, 6l, Supplementary Figs. 1c, 3b, 4c were created using BioRender.com. This work was supported by a Wellcome Trust Seed Award in Science (210077/Z/17/Z) and MRC New Investigator Research Grant (MR/R026610/1) awarded to SE. MF was supported by a Gerald Kerkut Trust PhD studentship.

## Author contributions

M.F. designed and performed experiments, analysed, and interpreted the data. F.S.G.H. performed the bioinformatic analyses. L.M. and C.L.G.J.S. performed immunofluorescence and *human* 3D culture experiments and analysed the data. E.L. performed rescue and immunofluorescence experiments. M.M.H. performed immunofluorescence and *mouse* 3D culture experiments. X.M. designed custom macros for the quantification of cortical fluorescence intensities. P.S. performed the LC-MS/MS experiments. S.E. conceived and designed the project, performed experiments, analysed, and interpreted the data, and wrote the manuscript. All the authors provided intellectual input, edited, and approved the final manuscript.

## Competing interests

The authors declare no competing interests.
