## [Peer Review File · Nature Communications]

Reviewers' comments:

Reviewer #1 (Remarks to the Author):

The accurate positioning of the mitotic spindle is critical for proper development and morphogenesis. In most instances, proper spindle positioning requires interaction between the astral microtubules and the cortical dynein/dynactin complex. The cortical dynein/dynactin distribution in metaphase is regulated by evolutionarily conserved machinery comprising G/LGN/NuMA. However, the mechanisms by which the localization of this complex is spatiotemporally regulated during mitosis remains incompletely known. In this manuscript, using a proteomic approach, the authors uncovered a novel interacting partner of LGN, i.e., Annexin A1 (ANXA1). They show that depletion of ANXA1 affects the alignment of the mitotic spindle, which eventually disrupt epithelial architecture and lumen formation in primary mammary organoids.

Here, the authors have attempted to characterize a novel protein, ANXA1 and linked it to spindle positioning. However, the major weakness is we learn very little about the molecular mechanism by which ANXA1 influence LGN/NuMA localization (see a few major points related to that). Therefore, I feel that in the absence of molecular mechanisms, the scope of this study is limited until authors put substantial efforts into clarifying the molecular mechanism by which ANXA1 regulates spindle positioning in mammalian cells and in their organoid model.

Major points:

1. In Fig. 1, and the associated Supplementary Fig. 1, the authors established that ANXA1 is an interacting partner of LGN. They further show that ANXA1 is enriched at the equatorial membrane, and LGN localization is more restricted to the polar region of the cell cortex. The authors identified ANXA1 as a potential interacting partner of LGN, then why LGN and ANXA1 show this 'mutually exclusive cortical-lateral distribution' is not clear to me. The co-localization studies were done using fixed cells. Why were these analyses not performed using cells that co-expressed ANXA1 and LGN equivalent to the endogenous proteins? Also, as claimed by the authors, this data alone does not reveal that ANXA1 is a potential regulator of LGN.
2. In Fig. 2, the authors have analyzed the impact of ANXA1 depletion on LGN distribution by live imaging and in fixed cells. Here, they uncovered that siRNA-mediated depletion of ANXA1 affects LGN/NuMA and p150Glued cortical distribution. However, cells depleted of ANXA1 also reveal chromosomes congression defects (for instance, check Fig. 2A and 2D). It may well be that chromosomes congression defects because of the ANXA1 loss is impacting LGN/NuMA and p150Glued cortical distribution. Previous work has shown that kinetochore localized Plk1 affect cortical LGN/NuMA distribution (see Tame et al., 2016; and Sana et al., 2019). Thus the impact of ANXA1 depletion on LGN/NuMA distribution could be because of chromosomes congression defects in their settings rather than its direct effect on cortical LGN/NuMA. Also, the mitotic delay observed in Fig. 3 could be because of chromosomes misalignment upon ANXA1 KD.
3. As a follow up on the above point, the spindle positioning phenotypes (Z-rotation during metaphase, oscillation index, and cells with spindle oscillations) observed in Fig. 3 could be because of an indirect impact of ANXA1 depletion on chromosomes alignment. They have further linked the role of ANXA1 in planer cell division and morphogenesis (Fig 4); here again, the inability of the ANXA1 depleted cells to align the mitotic spindle could be due to global defects in chromosomes alignment and that is leading to spindle positioning and morphogenesis defects in their organoids system.
4. I suggest that the authors should test if ANXA1 is sufficient to impact the cortical distribution of LGN/NuMA by ectopically target ANXA1 using optogenetics or a similar approach. This experiment will be useful in directly linking the contribution of ANXA1 in modulating cortical LGN/NuMA.

Minor comments:

1. It remained unclear in this work if the interaction of ANXA1 with LGN is independent of NuMA.
2. In the initial part of their introduction, the authors have only cited the review articles and missed mentioning the original contribution to the field. It is a matter of taste, but I would prefer to cite the original work instead of only reviews.
3. In mammalian cells, the Plk1/and Ran-based mechanisms are shown to exclude LGN/NuMA from the membrane in the proximity of chromosomes (Kiyomitsu and Cheeseman, 2012; Tame et al., 2016; Sana et al., 2019). However, the authors mentioned that 'the mechanisms that polarize LGN at the cell cortex remain unfolding'. I felt that the authors did not discuss in depth the previously published work.
4. On page 7, the authors mentioned 'ANXA1, LGN, NuMA, and S100A11 are part of the same macromolecular complex. However, in their IP study with mCherry-ANXA1, they only detected LGN (Fig. 1d), and thus it is not clear why they claim that ANXA1, LGN, NuMA, and S100A11 is part of the same complex.

Reviewer #2 (Remarks to the Author):

The manuscript submitted by Fankhaenel et al., is aimed at bringing additional clues to the fundamental regulation of mitotic spindle orientation in epithelia.

The authors have clearly presented the status of the knowledge on mechanisms underlying spindle orientation, and they have focused their investigation on the cause of polarized cortical LGN anchoring. They have used a proteomics approach via affinity purification and mass spectrometry to unravel the cortical interactome of LGN. They chose the mammary gland as a working model in light of the importance of its development throughout life and the previously demonstrated link between spindle misorientation and the risk of mammary tumor onset. Using GFP-LGN-expressing MCF10A cells, they have analyzed interactomes of LGN in cells in division (arrested in metaphase) and in interphase (arrested in G2) to identify membrane-associated annexin A1 ANXA1 as a previously unrecognized interactor of LGN and actor in cell division. Important controls for the coimmunoprecipitation and for the potential interaction between LGN and ANXA1 are included in the report. They claim based on results with fluorescence microscopy and siRNA against ANXA1 that the interaction between LGN and ANXA1 restricts LGN to the lateral cortex, a condition necessary for the correct division plane of mammary epithelial cells. An abundance of microscopic analyses of mitotic spindle movements and position confirms the impact of ANXA1 on this specific aspect of mitosis. Validation for the role of ANXA1 in mitosis of the glandular epithelium is brought by 3D cell culture of murine acini.

Although the report reveals ANXA1 as a new player in mitotic spindle orientation, it does not really highlight a significant step in understanding why the LGN-NuMA-dynactin complex and spindle locate parallel to the epithelium in order to ensure the maintenance of a monolayer of luminal cells. The new knowledge appears incremental since there are other proteins already shown to regulate the cortical localization of LGN. Moreover, the authors make several statements regarding the importance of understanding the link between mitotic homeostasis and breast cancer onset; specifically, they stress the fact that mitotic spindle misorientation might be linked to cancer development via piling up of cells, which several other teams have suggested and partly demonstrated, but they do not bring exciting new information about this, including in the discussion. There is a tendency to 'overinterpret' some results, notably when related to validation experiments, and there is a misuse of major terms that have a strict biological meaning, like apical polarity and organoids. These issues need to be resolved so that the readers are not misled regarding the findings. Moreover, the discussion needs to

be more meaningful in terms of how ANXA1 fits in the mechanistic knowledge of spindle pole orientation and the mechanisms that participate in homeostasis necessary to prevent tumor onset, since the latter is a theme that is present on several occasions in the report.

Major Comments

Comment #1: The authors use cells transfected with LGN or ANXA1 for all immunoprecipitation experiments. It is not clear why no validation experiments were performed with only endogenous proteins, notably with human cells since this is the species of interest in light of the mention of breast cancer onset and the link to mitotic spindle orientation.

Comment #2: Some of the conclusions or statements from the results need to be reconsidered.

-Page 7: When listing the proteins that copurify with NuMA, it is important to distinguish those that are indeed known to be part of a complex with several proteins involved together (see use of term subcomplex in the text) and those for which only copurification is shown.

-Page 8: It is not clear how the microscopy analysis presented in the first half of page 8 may already lead to the conclusion that ANXA1 is a potential regulator of LGN. This conclusion can only be made later in the manuscript when an induced change in ANXA1 location and/or expression also changes LGN location (shown in the next section of the manuscript). Also, how can we exclude that ANXA1 decrease in expression with siRNA is indirectly responsible for the change in LGN location. What happens to other elements upstream of LGN for spindle orientation? It seems that only Gα protein is analyzed.

-Page 10: The title of this subsection is not representative of the content. Only one experiment is focused on determining whether ANXA1 acts upstream of LGN in this entire section.

Comment #3: The validation step with real tissues and 3D cell culture does not use the most appropriate model or it is at least incomplete.

-Page 12: Why only use murine tissues to validate the location of ANXA1 in the mammary gland and not human normal breast tissue that is abundantly available now?

-Also, the term organoid is wrongly used throughout the manuscript and needs to be removed and replaced with acinar differentiation or 3D culture of mammary epithelial cells. An organoid normally means the formation of groups of cells that include ALL the cell types in an organ or part of an organ. The authors isolate mammary epithelial cells from the murine mammary gland to culture them within Matrigel to make acinus-like structures, so they do not form organoids.

-It is not clear why the authors they did not use human cells in addition to the murine model to validate their findings in the species in which breast cancer development is an issue. Even just cell lines or primary cells shown by others to form polarized acini in Matrigel would have been an important complement to the murine model. The MCF10A cells would not be a good model for proper orientation of the mitotic spindle: they do not form fully polarized acini and the lumen is made by cell death probably because cells accumulate in the wrong direction via random mitotic spindle orientation, as shown by others. In fact, it would be a strong validation for human cells to place MCF10A cells in 3D culture in Matrigel and check the distributions of ANXA1 and LGN with regards to mitotic spindle orientation and compare to cells that only form one layer of cells thanks to the correct orientation of the mitotic spindle (see Bazzoun et al. J Cell Sci 2019; TenVooren et al., oncogene 2019).

- F-actin is not a true marker of apical polarity; instead, tight junction markers should be used when assessing apical polarity.

Comment #4: The discussion is not strong enough and the novelty of the findings is not discussed.

-The discussion as it stands currently does not bring much to the knowledge regarding the control of mitotic spindle orientation. A major discussion point should be how ANXA1 distinguishes itself from other known controllers of mitotic spindle orientation that are listed by the authors. The authors mention that spindle control might be context dependent (see end of page 16), but there is no explanation regarding why they make such a statement.

-Is it unique for ANXA1 to control lateral/polarized cortical location of LGN rather than cortical location overall? Most of the experiments are performed in 2D culture of MCF10A cells without the possibility of checking for an impact from polarity. It is not sure that the little amount of work done with a polarized model is sufficient to claim an impact from ANXA1 on polarized cortical location of LGN.... especially since the work in 3D culture is done only with murine cells that are not totally similar to the human mammary epithelium.

-It would also be interesting to have a short discussion on ANXA1-LGN relatively close interaction as observed by immunoprecipitation and one of the microscopy tests, and their nonoverlapping distribution shown by immunostaining.

-page 17, it is stated that ANXA1 marks a subset of luminal epithelial cells in the murine mammary gland. But there is no further discussion about this fact. Most importantly, it would have been essential to check if it is also the case in the human mammary gland, especially in light of the interest for the mechanisms of cancer onset.

-A major gap in the discussion is the relation between Annexin A1 and the Pi3K pathway that controls polarity and that has been linked by others to ANXA1 and also to mitotic spindle orientation control in models of human breast epithelial differentiation. This is an important aspect of the discussion because Pi3K is an essential pathway in breast cancer development and the authors emphasize the role of misorientation of the spindle in cancer onset. Yet, others have shown that Pi3K controls spindle orientation and there is no discussion based on that literature. The authors state that lumen expansion is driven by planar cell divisions independently of apical polarity (see page 16), but there is no strong or detailed explanation regarding why they can make such a conclusion.

-The next important question to address in the discussion is possibly what makes ANXA1 locate to the lateral cell cortex.

Minor comments

1. Materials and Methods: Cell extracts and immunoblotting-- give antibody concentrations instead of dilution as the latter may differ depending on the concentration of the stock of antibodies;
2. Materials and Methods: it is cell culture MEDIUM when singular, not media! Media means the Press.
3. Page 25: it should be fluorescence intensities instead of fluorescent intensities.
4. Figure 1: results of sections c through g are in general difficult to interpret.
5. Figure 2a: images are not very convincing with regards to the statement made in the text.
6. Figure 3a: tell what the arrowheads indicate in the figure legend; anomalies listed in 'e' should be clearly shown in representative images in 'a'.
7. Figure 4i: The abnormal piling up of cells in the acinus should be shown with an arrow.

Reviewer #3 (Remarks to the Author):

For transparency and ease of interpretation, please include a supplementary table with all proteins identified in the IP experiments, including the number of peptides identified per protein. Raw files were deposited to PRIDE, but there are no summary tables with this information.

The text on page 6 suggests that 'singletons' were removed from consideration. My understanding of 'singleton' is a protein identified by a single peptide hit. 10 out of 18 proteins in the 18 protein network are detected with only one peptide in experiment 1 (NUMB, PLK1, CCNB1, DCTN2, S100A11, SAA1, PARD3, HAUS6, TUBG1). What does singleton mean in this context? Please clarify.

Page 7, "...were found in the LGN protein network by co-purifying with with NuMA. We also identified the oncogene EIF3E (Eukaryotic translation initiation factor 3 subunit E)46, 47, in the NuMA subcomplex. Finally, we identified a ternary complex comprising the membrane-associated protein Annexin A1 (ANXA1) 48 and its partner S100A11 (S100 Ca²⁺-binding protein A11) 49, as well as the Serum Amyloid A-1 protein (SAA1) 50, that associates with LGN." Please refer to these protein-protein interactions as sub-networks. STRING connections do not necessarily indicate physical protein complexes, which is what is implied in the text.

Point-by-point response to the Reviewer comments

Thanks very much for sending along the reviews of our paper. We are grateful for the Reviewers comments, which we have all addressed to improve the initial manuscript. Please find below our point-by-point response to the Reviewers comments. We have performed additional experiments and analyses, revised the figures, added new figures, and modified the text to address all the Reviewers' concerns. Additionally, we have improved our manuscript's discussion to highlight the significance and novelty of our findings and proposed ANXA1-mediated mechanism regulating LGN polarised cortical accumulation and planar mitotic spindle orientation in mammalian epithelial cells, in comparison with those reported in the literature.

Reviewer #1 (Remarks to the Author)

The accurate positioning of the mitotic spindle is critical for proper development and morphogenesis. In most instances, proper spindle positioning requires interaction between the astral microtubules and the cortical dynein/dynactin complex. The cortical dynein/dynactin distribution in metaphase is regulated by evolutionarily conserved machinery comprising G/LGN/NuMA. However, the mechanisms by which the localization of this complex is spatiotemporally regulated during mitosis remains incompletely known. In this manuscript, using a proteomic approach, the authors uncovered a novel interacting partner of LGN, i.e., Annexin A1 (ANXA1). They show that depletion of ANXA1 affects the alignment of the mitotic spindle, which eventually disrupt epithelial architecture and lumen formation in primary mammary organoids.

Here, the authors have attempted to characterize a novel protein, ANXA1 and linked it to spindle positioning. However, the major weakness is we learn very little about the molecular mechanism by which ANXA1 influence LGN/NuMA localization (see a few major points related to that). Therefore, I feel that in the absence of molecular mechanisms, the scope of this study is limited until authors put substantial efforts into clarifying the molecular mechanism by which ANXA1 regulates spindle positioning in mammalian cells and in their organoid model.

Major points:

1. In Fig. 1, and the associated Supplementary Fig. 1, the authors established that ANXA1 is an interacting partner of LGN. They further show that ANXA1 is enriched at the equatorial membrane, and LGN localization is more restricted to the polar region of the cell cortex. The authors identified ANXA1 as a potential interacting partner of LGN, then why LGN and ANXA1 show this 'mutually exclusive cortical-lateral distribution' is not clear to me. The co-localization studies were done using fixed cells. Why were these analyses not performed using cells that co-expressed ANXA1 and LGN equivalent to the endogenous proteins? Also, as claimed by the authors, this data alone does not reveal that ANXA1 is a potential regulator of LGN.

- As Reviewer #1 rightly pointed out, while we describe different cortical dynamic distributions of LGN and ANXA1 particularly during prometaphase, our experiments in fixed cells do not allow us to conclude that the proteins are "mutually exclusive". We have amended the text to remove this statement (page 8). To address the Reviewer's point further, we have toned-down our conclusion by removing the statement "...and suggest ANXA1 as a potential regulator of LGN" (page 8).

We have generated clonal MCF-10A cells co-expressing GFP-LGN and ANXA1-mCherry to assess the dynamic distribution of both proteins. However, as shown below (Reviewer Figure 1), our live imaging experiments using a wide-field deconvolution microscope (DeltaVision Elite) which was used for all the live imaging experiments in this manuscript, did not allow us to distinguish between cytoplasmic and cortical ANXA1-mCherry. ANXA1 has been shown to have a widespread subcellular localisation in the cell, including all vesicular organelles, mitochondria, nucleus, in addition to the plasma membrane. To allow for an accurate live imaging analysis of the spatiotemporal dynamic assembly and co-distribution of GFP-LGN and ANXA1-mCherry, we would require a higher resolution microscope. Based on our experience of live imaging of mitosis,

devising these experiments will require extensive and time-consuming optimisations, which is beyond the initial scope of this study.

To show that immunofluorescent detection of ANXA1 faithfully represents its distribution throughout the cell cycle, we have added new confocal images showing the co-distribution of LGN and ANXA1 throughout the cell cycle in our clonal MCF-10A co-expressing GFP-LGN and ANXA1-mCherry (new Supplementary Fig. 2d). These new results are also described in the text (page 8). Our confocal imaging shows that ectopically expressed LGN and ANXA1 behave as their endogenous counterparts.

Reviewer Figure 1. Live imaging of GFP-LGN and ANXA1-mCherry in MCF-10A cells.

2. In Fig. 2, the authors have analyzed the impact of ANXA1 depletion on LGN distribution by live imaging and in fixed cells. Here, they uncovered that siRNA-mediated depletion of ANXA1 affects LGN/NuMA and p150Glued cortical distribution. However, cells depleted of ANXA1 also reveal chromosomes congression defects (for instance, check Fig. 2A and 2D). It may well be that chromosomes congression defects because of the ANXA1 loss is impacting LGN/NuMA and p150Glued cortical distribution. Previous work has shown that kinetochore localized Plk1 affect cortical LGN/NuMA distribution (see Tame et al., 2016; and Sana et al., 2019). Thus the impact of ANXA1 depletion on LGN/NuMA distribution could be because of chromosomes congression defects in their settings rather than its direct effect on cortical LGN/NuMA. Also, the mitotic delay observed in Fig. 3 could be because of chromosomes misalignment upon ANXA1 KD.

- The Reviewer raises an important point that we addressed in the discussion of the initial manuscript, where we cited one of the papers they mentioned (Tame et al., 2016 EMBO Rep) (page 14-15). As rightly pointed out by the reviewer, the effect of ANXA1 depletion on the cortical localisation of LGN/NuMA could be due to the defects in chromosome alignment. While we cannot formally rule out this possibility, in our study, we did not detect ANXA1 on chromosomes, which makes it unlikely that chromosome misalignments are a direct effect of ANXA1 depletion. Moreover, recent studies have shown that perturbation of the astral microtubule-cortex crosstalk causes defects in chromosome alignment and segregation (di Pietro et al., 2017 Current Biology; Yu et al., 2019 Cell Research). Another elegant study from Angelika Amon lab (Knouse et al., 2018 Cell) has shown that epithelial cell geometry and polarity affects faithful chromosome segregation. Our proteomic data also reveal several proteins co-purifying with NuMA (such as Dynein, CLASP, RAN, HAUS6) that have been shown to directly regulate the dynamics of astral microtubules. We have extended our discussion to further address this important point and included the paper by Sana et al., 2018 (Life Sci Alliance) (pages 17-18). Based on these studies, it is likely that ANXA1 may regulate astral microtubule dynamics. It will be interesting to further explore the molecular mechanisms linking ANXA1-mediated regulation of astral microtubule dynamics to accurate chromosome segregation, however, we feel that this is out of the scope of this manuscript.

3. As a follow up on the above point, the spindle positioning phenotypes (Z-rotation during metaphase, oscillation index, and cells with spindle oscillations) observed in Fig. 3 could be because of an indirect impact of ANXA1 depletion on chromosomes alignment. They have further linked the role of ANXA1 in planer cell division and morphogenesis (Fig 4); here again, the inability of the ANXA1 depleted cells to align the mitotic spindle could be due to global defects in chromosomes alignment and that is leading to spindle positioning and morphogenesis defects in their organoids system.

- We agree with the Reviewer that excessive oscillations of the spindle could be due to defects in the alignment of the metaphase plate, which could prolong the spindle assembly checkpoint, thereby affecting the dynamics of the mitotic spindle. Nonetheless, our observations showing elongation and buckling of astral microtubules (new Supplementary Fig. 5a-d), in addition to defects in cortical F-actin organisation (new Supplementary Fig. 5e, f) in ANXA1-depleted cells further indicate that ANXA1 acts on the astral microtubule-cortex crosstalk to regulate mitotic spindle dynamics. Our findings are consistent with studies demonstrating the influence of cortical cytoskeleton in 2D culture (di Pietro et al., 2017, Current Biology; Yu et al., 2019 Cell Research) and epithelial cell architecture in 3D culture (Knouse et al., 2018 Cell) on mitotic spindle orientation and assembly. Our findings reinforce the idea that defects in the F-actin-astral microtubules crosstalk caused by ANXA1 depletion impairs spindle orientation and assembly, which in turn could also cause incorrect chromosome segregation. As discussed above (Major Point 2) and in the manuscript's text, it will be interesting to investigate the mechanisms linking ANXA1-mediated regulation of cell cortex organisation to mitotic spindle assembly and chromosome segregation.

4. I suggest that the authors should test if ANXA1 is sufficient to impact the cortical distribution of LGN/NuMA by ectopically target ANXA1 using optogenetics or a similar approach. This experiment will be useful in directly linking the contribution of ANXA1 in modulating cortical LGN/NuMA.

- To address this important point also raised by Reviewer #2, we have added rescue experiments in si-ANXA1-treated MCF-10A cells where we expressed an si-ANXA1-resistant ANXA1-mCherry. Our new data show that expression of ANXA1-mCherry in ANXA1-depleted MCF-10A cells restores the accumulation of the LGN-NuMA-Dynein-Dynactin complex at the lateral cortex (new Fig. 2d-f) as well as planar mitotic spindle orientation (new Supplementary Fig 4a, b). The optogenetic experiments requested by Reviewer #1 to induce ectopic relocalization of ANXA1 during mitosis are challenging to devise given that ANXA1 has a widespread subcellular localisation in the cell, as discussed above. Additionally, there are no direct inhibitors of ANXA1 available to perform similar experiments. Alternatively, we have added new experiments in which we treated MCF-10A cells with TG100-115 (20 μ M for 2h), a potent and specific inhibitor of the TRPM7 kinase that phosphorylates ANXA1 on Ser5 and that has been suggested to regulate ANXA1 recruitment to membrane phospholipids along with its partner S100A11. Our new data (new Supplementary Fig.3) show that MCF-10A treatment with TG100-115 abrogates ANXA1 localisation to the plasma membrane and results in its translocation to the cytoplasm, leading to aberrant cortical accumulation of LGN, NuMA and p150Glued, which is in line with the results we obtained in ANXA1-depleted cells (Fig. 2d). Together, our new data indicate that ANXA1 localisation to the plasma membrane is sufficient to control the polarised accumulation of LGN-NuMA-Dynein-Dynactin at the lateral cortex.

Minor comments:

1. It remained unclear in this work if the interaction of ANXA1 with LGN is independent of NuMA.

- In the present work, we show that ANXA1 acts upstream of LGN. We also show that upon ANXA1 depletion or inhibition both LGN and NuMA remain at the cell cortex where their restricted lateral accumulation is similarly impaired, suggesting that ANXA1 does not act on the assembly of the LGN-NuMA complex, but rather controls its polarized cortical distribution. Furthermore, our proteomic data show that NuMA co-purifies with the ANXA1-LGN complex, suggesting that ANXA1, LGN and NuMA are likely to form a ternary complex. However, we fully agree with the

Reviewer that future experiments, such as those using si-NuMA and in vitro competition binding assays, will be needed to address whether the ANXA1-LGN complex forms independently of NuMA, whether NuMA helps the formation of the complex or competes LGN from ANXA1. Additionally, structure-function studies will be key to dissect the mechanisms underlying the assembly and spatiotemporal dynamics of the ANXA1-LGN-NuMA complex at the cell cortex and understand how ANXA1 acts with NuMA and synergizes with $G_{\alpha i}$ to restrict the cortical accumulation of LGN. Although these experiments fall beyond the scope of this initial study, we now discuss this important point raised by the Reviewer in the revised text (pages 16-17).

2. In the initial part of their introduction, the authors have only cited the review articles and missed mentioning the original contribution to the field. It is a matter of taste, but I would prefer to cite the original work instead of only reviews.

- We share Reviewer's taste and prefer to cite original articles. Due to the reference number limit, we had to remove a few papers to replace them with excellent reviews. Nonetheless, we have cited, in an unbiased manner, most of the original papers that are essential to the present study. Additionally, in response to both Reviewers #1 and Reviewer #2 comments we have included additional original papers throughout, to improve our revised manuscript.

3. In mammalian cells, the Plk1/and Ran-based mechanisms are shown to exclude LGN/NuMA from the membrane in the proximity of chromosomes (Kiyomitsu and Cheeseman, 2012; Tame et al., 2016; Sana et al., 2019). However, the authors mentioned that 'the mechanisms that polarize LGN at the cell cortex remain unfolding'. I felt that the authors did not discuss in depth the previously published work.

- We apologise for not discussing further these important original studies. We kept the focus of the introduction on the cortical polarity cues that have been shown to regulate the localisation of LGN. We have now extended the discussion of these papers and included more details about how the PLK1 and RANGTP-mediated mechanisms regulate the restricted cortical accumulation of LGN during mitosis (pages 16-17).

4. On page 7, the authors mentioned 'ANXA1, LGN, NuMA, and S100A11 are part of the same macromolecular complex. However, in their IP study with mCherry-ANXA1, they only detected LGN (Fig. 1d), and thus it is not clear why they claim that ANXA1, LGN, NuMA, and S100A11 is part of the same complex.

- We have suggested that ANXA1, LGN, NuMA and S100A11 could be part of the same complex based on our combined IP, PLA, proteomic and immunofluorescence observations (Fig 1a, c-f; Supplementary Fig. 2a). In response to the Reviewer's point, we have removed the statement suggesting ANXA1, LGN, NuMA and S100A11 are part of the same complex (page 7).

Reviewer #2 (Remarks to the Author)

The manuscript submitted by Fankhaenel et al., is aimed at bringing additional clues to the fundamental regulation of mitotic spindle orientation in epithelia.

The authors have clearly presented the status of the knowledge on mechanisms underlying spindle orientation, and they have focused their investigation on the cause of polarized cortical LGN anchoring. They have used a proteomics approach via affinity purification and mass spectrometry to unravel the cortical interactome of LGN. They chose the mammary gland as a working model in light of the importance of its development throughout life and the previously demonstrated link between spindle misorientation and the risk of mammary tumor onset. Using GFP-LGN-expressing MCF10A cells, they have analyzed interactomes of LGN in cells in division (arrested in metaphase) and in interphase (arrested in G2) to identify membrane-associated annexin A1 ANXA1 as a previously unrecognized interactor of LGN and actor in cell division. Important controls for the coimmunoprecipitation and for the potential interaction between LGN and ANXA1 are included in

the report. They claim based on results with fluorescence microscopy and siRNA against ANXA1 that the interaction between LGN and ANXA1 restricts LGN to the lateral cortex, a condition necessary for the correct division plane of mammary epithelial cells. An abundance of microscopic analyses of mitotic spindle movements and position confirms the impact of ANXA1 on this specific aspect of mitosis. Validation for the role of ANXA1 in mitosis of the glandular epithelium is brought by 3D cell culture of murine acini.

Although the report reveals ANXA1 as a new player in mitotic spindle orientation, it does not really highlight a significant step in understanding why the LGN-NuMA-dynactin complex and spindle locate parallel to the epithelium in order to ensure the maintenance of a monolayer of luminal cells. The new knowledge appears incremental since there are other proteins already shown to regulate the cortical localization of LGN. Moreover, the authors make several statements regarding the importance of understanding the link between mitotic homeostasis and breast cancer onset; specifically, they stress the fact that mitotic spindle misorientation might be linked to cancer development via piling up of cells, which several other teams have suggested and partly demonstrated, but they do not bring exciting new information about this, including in the discussion. There is a tendency to 'overinterpret' some results, notably when related to validation experiments, and there is a misuse of major terms that have a strict biological meaning, like apical polarity and organoids. These issues need to be resolved so that the readers are not misled regarding the findings. Moreover, the discussion needs to be more meaningful in terms of how ANXA1 fits in the mechanistic knowledge of spindle pole orientation and the mechanisms that participate in homeostasis necessary to prevent tumor onset, since the latter is a theme that is present on several occasions in the report.

Major Comments

Comment #1: The authors use cells transfected with LGN or ANXA1 for all immunoprecipitation experiments. It is not clear why no validation experiments were performed with only endogenous proteins, notably with human cells since this is the species of interest in light of the mention of breast cancer onset and the link to mitotic spindle orientation.

- We thank the Reviewer for raising this important point, we would like to clarify that all our validation experiments were performed in clonal human MCF-10A mammary cells stably expressing GFP-LGN and/or ANXA1-mCherry. We have included several controls throughout showing that GFP-LGN and ANXA1-mCherry behave like their endogenous counterparts. Our reciprocal immunoprecipitation experiments in our clonal cells stably expressing either GFP-LGN or ANXA1-mCherry allowed us to detect both endogenous and ectopically expressed ANXA1 and LGN, thereby rigorously validate our proteomic and mass spectrometry results. These validation experiments were reinforced by PLA (Fig 1e) as well as immunofluorescence where we co-labelled endogenous ANXA1 and LGN (Supplementary Fig. 2a).

Comment #2: Some of the conclusions or statements from the results need to be reconsidered.

-Page 7: When listing the proteins that copurify with NuMA, it is important to distinguish those that are indeed known to be part of a complex with several proteins involved together (see use of term subcomplex in the text) and those for which only copurification is shown.

- We apologise for the confusion; we have now amended the text to address the Reviewer's point to distinguish between proteins known to interact with NuMA and those co-purified in our proteomic experiments (page 7).

-Page 8: It is not clear how the microscopy analysis presented in the first half of page 8 may already lead to the conclusion that ANXA1 is a potential regulator of LGN. This conclusion can only be made later in the manuscript when an induced change in ANXA1 location and/or expression also changes LGN location (shown in the next section of the manuscript). Also, how can we exclude that ANXA1 decrease in expression with siRNA is indirectly responsible for the change in

LGN location. What happens to other elements upstream of LGN for spindle orientation? It seems that only Gα protein is analysed.

- As rightly pointed out by the Reviewer our results presented in Fig. 1 and Supplementary Fig. 2 do not allow us to suggest that ANXA1 acts as a potential regulator of LGN. To address this point also raised by Reviewer #1 we removed the claim to tone down our conclusion (page 8).

The Reviewer's point about the specificity of the effect of ANXA1 on LGN cortical accumulation was also raised by Reviewer #1. As discussed above in response to Reviewer #1 (Major Point 4), we have now included new rescue experiments showing that expression of an si-ANXA1-resistant ANXA1-mCherry in ANXA1-depleted MCF-10A cells restores the accumulation of the LGN-NuMA-Dynein-Dynactin complex at the lateral cortex (new Fig. 2d-f) as well as planar mitotic spindle orientation (new Supplementary Fig 4a, b). To further assess the sufficiency of ANXA1, we treated MCF-10A cells with TG100-115 (20 μM for 2h), a potent and specific inhibitor of the TRPM7 kinase that phosphorylates ANXA1 on Ser5, regulating its recruitment to membrane phospholipids. We show that inhibition of ANXA1 localisation to the plasma membrane leads in aberrant cortical accumulation of LGN, NuMA and p150^{Glued} (new Supplementary Fig.3), similarly to the results in ANXA1-depleted cells. Together, these new data reinforce the model whereby ANXA1 serves as a cortical landmark to ensure proper cortical distribution of LGN and its effectors NuMA and Dynein-Dynactin during metaphase.

We assessed the effect of ANXA1 depletion on Gai a key upstream partner of LGN that directs its recruitment to the cell cortex during mitosis. To address the Reviewer's point further, we have added new experiments to examine the impact of ANXA1 depletion on cortical F-actin organisation. F-actin has recently been shown to interact with LGN and Afadin to regulate the LGN-NuMA complex during mitosis (Carminati et al., 2016 Nat SMB). We found that ANXA1 depletion impairs the integrity of cortical F-actin (new Supplementary Fig. 5e, f), further indicating that ANXA1 acts as an upstream polarity cue to ensure proper cortex-astral microtubule crosstalk for correct mitotic spindle orientation.

-Page 10: The title of this subsection is not representative of the content. Only one experiment is focused on determining whether ANXA1 acts upstream of LGN in this entire section.

- The Reviewer raises an important point which we have addressed in the revised manuscript. In the initial manuscript, we show that LGN knockdown does not affect ANXA1 cortical accumulation, while it abrogates NuMA recruitment at the cortex (new Supplementary Fig. 4c-f). These findings together with our new successful rescue experiments using si-ANXA1-resistant ANXA1-mCherry (new Fig 2d-f), preventing ANXA1 plasma membrane localisation (new Supplementary Fig. 3) and those showing that cortical F-actin integrity is affected by ANXA1 depletion (new Supplementary Fig. 5e, f), reinforce our conclusion that ANXA1 acts upstream of LGN.

Comment #3: The validation step with real tissues and 3D cell culture does not use the most appropriate model or it is at least incomplete.

-Page 12: Why only use murine tissues to validate de location of ANXA1 in the mammary gland and not human normal breast tissue that is abundantly available now?

- As requested by the Reviewer, we have included new data showing immunofluorescence experiments in human healthy breast tissue where we confirm that ANXA1 is mostly detected at the cortex of luminal cells similarly to the mouse mammary epithelium, but also reveal ANXA1 expression in a few α-smooth muscle actin (α-SMA)-positive basal cells (Supplementary Fig. 6a). We discuss this difference in ANXA1 expression profile and emphasise the established intrinsic differences between the mouse and human mammary epithelium regarding the expression of cell lineage markers (pages 19-20).

-Also, the term organoid is wrongly used throughout the manuscript and needs to be removed and replaced with acinar differentiation or 3D culture of mammary epithelial cells. An organoid normally

means the formation of groups of cells that include ALL the cell types in an organ or part of an organ. The authors isolate mammary epithelial cells from the murine mammary gland to culture them within Matrigel to make acinus-like structures, so they do not form organoids.

- To address the Reviewer's point, we have replaced the term "organoid" by "acinus" to refer to the 3D cultures of MECs throughout the revised manuscript.

-It is not clear why the authors they did not use human cells in addition to the murine model to validate their findings in the species in which breast cancer development is an issue. Even just cell lines or primary cells shown by others to form polarized acini in Matrigel would have been an important complement to the murine model. The MCF10A cells would not be a good model for proper orientation of the mitotic spindle: they do not form fully polarized acini and the lumen is made by cell death probably because cells accumulate in the wrong direction via random mitotic spindle orientation, as shown by others. In fact, it would be a strong validation for human cells to place MCF10A cells in 3D culture in Matrigel and check the distributions of ANXA1 and LGN with regards to mitotic spindle orientation and compare to cells that only form one layer of cells thanks to the correct orientation of the mitotic spindle (see Bazzoun et al. J Cell Sci 2019; Tenvooren et al., oncogene 2019).

- We would like to thank the Reviewer for these suggestions; we have added new experiments assessing the orientation of the mitosis spindle in 3D cultures of MCF-10A (at 96h and 192h) and human primary MEC-derived acini (at 240h) (new Supplementary Fig. 6).

Our experiments in MCF-10A-derived acini show that ~60% and ~70% of cells align the mitotic spindle planarly to the basement membrane at 96h and 192h of 3D culture, respectively (Supplementary Fig. 6b, c). ANXA1 localises to the cell cortex of mitotic MCF-10A cells (Supplementary Fig. 6d), further indicating that cortical ANXA1 is required for planar mitotic spindle orientation in MCF-10A 3D cultures. However, planar cell divisions in MCF-10A cell-derived acini do not lead to lumen formation (Supplementary Fig. 6b, c), consistent with studies showing that lumen forms by apoptosis of inner cells at the end of cystogenesis in this 3D culture system (Debnath et al., 2002 Cell). Thus, as also rightly pointed by the Reviewer, 3D cultures of MCF-10A are not suitable for the study of the contribution of oriented cell divisions to lumen formation. Another human mammary S1 cell line, which as opposed to MCF-10A cells, can establish an apical polarity but fail to expand a central lumen (Bazzoun et al., 2019 J Cell Sci; Tenvooren et al., 2019 Oncogene; Plachot et al., 2009 BMC Biol), making these cells again not suitable to address our question of the link between mitotic spindle orientation and lumen formation.

Our new experiments in primary human MECs (hMECs) reveal multi-layered acini displaying high architectural heterogeneity (Supplementary Fig. 6e). This architectural heterogeneity inherent to hMECs has been described previously (Rosenbluth et al., 2020 Nature Communications). Out of 601 acini, only 15 (~2.5%) form a central lumen and establish an apico-basal polarity with ANXA1 distributing at the cell cortex and Par6 accumulating at the apical surface (Supplementary Fig. 6e, f), suggesting that hMECs are also not suitable to study the contribution of oriented cell divisions (OCDs) to lumen formation in 3D culture. These 3D cultures have been shown to rarely form a lumen (Dekkers et al., 2019 J Natl Cancer Int; Rosenbluth et al., 2020 Nature Communications), whether they are grown as spheres or as branched structures in Collagen or Matrigel (Sachs et al., 2018 Cell; Dekkers et al., 2019 J Natl Cancer Int; Rosenbluth et al., 2020 Nature Communications). It is important to note that to our knowledge the mechanisms of epithelial polarity and oriented cell divisions have not been characterised in hMEC-derived acini, indicating the need for novel 3D culture methods to accurately assess the functional requirement of ANXA1 in mitotic spindle orientation during hMEC cystogenesis.

We and others have shown that mouse MECs (mMECs) grown in 3D culture self-organise and polarise around a central lumen (Akhtar et al., 2013 Nat Cell Biol; Ahmed et al., 2016 Development), which is formed in an apoptosis-independent manner. To our knowledge, our present study establishes for the first time mMEC-derived acini as a unique 3D culture system to study the mechanisms linking planar spindle orientation to lumen formation in the mammary

epithelium. Based on our findings described in MCF-10A and hMEC acini, mMEC-derived acini are the most appropriate 3D culture model in our hands to address this question in the mammary epithelium. We have highlighted this important technical novelty in the revised text (pages 13-14 and 18).

- F-actin is not a true marker of apical polarity; instead, tight junction markers should be used when assessing apical polarity.

- We agree with the Reviewer that although F-actin accumulates at the apical cortex of polarised epithelial cells, it is not a true marker of apical polarity. However, F-actin is the most widely used marker to accurately and rigorously assess lumen formation during cystogenesis (*i.e.* compare single vs multi-lumen formation). To address the Reviewer's comment, we have added images of 3D acini labelled for the apical polarity protein Par6 (Fig. 4b; Supplementary Fig. 6e, f).

Comment #4: The discussion is not strong enough and the novelty of the findings is not discussed.

-The discussion as it stands currently does not bring much to the knowledge regarding the control of mitotic spindle orientation. A major discussion point should be how ANXA1 distinguishes itself from other known controllers of mitotic spindle orientation that are listed by the authors. The authors mention that spindle control might be context dependent (see end of page 16), but there is no explanation regarding why they make such a statement.

- We would like to thank the Reviewer's suggestions to improve our manuscript's discussion. We have now put more emphasis on the novelty of the ANXA1-mediated mechanisms regulating the cortical distribution of the LGN complex and mitotic spindle orientation, we discuss how these mechanisms differ from others involving key signalling such as PLK1 or RANGTP chromosome gradients, but also polarity cues such as Afadin, ABL1, SAPCD2 or E-cadherin. Our new data included in the revised manuscript have also help strengthen our discussion. In light of our findings and their novelty, we discuss the importance of future experiments to 1) dissect the mechanisms underlying the assembly and spatiotemporal dynamics of the ANXA1-LGN-NuMA complex at the cell cortex and understand how ANXA1 acts with NuMA and synergizes with Gai to restrict the cortical accumulation of LGN; 2) uncover the molecular mechanisms linking ANXA1 to LGN and F-actin will be key to understand how ANXA1 regulates astral microtubule-cortical actin crosstalk to ensure balanced pulling forces that orient the mitotic spindle to its correct position; 3) explore further the molecular mechanisms linking ANXA1-mediated regulation of astral microtubule dynamics to accurate chromosome segregation; and 4) investigate how ANXA1 regulates the interplay between LGN-NuMA and INSC-PAR3 and determine how this influences the switch between perpendicular and planar divisions in the differentiating mammary epithelium.

In response to the Reviewer's point, we clarified our statement that spindle control might be context dependent (page 19).

-Is it unique for ANXA1 to control lateral/polarized cortical location of LGN rather than cortical location overall? Most of the experiments are performed in 2D culture of MCF10A cells without the possibility of checking for an impact from polarity. It is not sure that the little amount of work done with a polarized model is sufficient to claim an impact from ANXA1 on polarized cortical location of LGN.... especially since the work in 3D culture is done only with murine cells that are not totally similar to the human mammary epithelium.

- The Reviewer made an important point about the role of ANXA1 in the regulation of the polarised accumulation of LGN at the lateral cortex. Our findings show that loss of ANXA1 does not result in a loss of LGN from the cell cortex, but rather affects its polarised distribution to the lateral cortex. This suggests that ANXA1 acts as a cortical landmark that restricts LGN accumulation to the lateral cortex but does not affect its cytoplasm-to-cortex recruitment. This is supported by our results showing that ANXA1 knockdown does not affect G_{ai}, which is a key partner of LGN ensuring its cytoplasm-to-cortex recruitment. Interestingly, NuMA is affected similarly to LGN upon ANXA1 depletion and remains at the cortex. The mechanism that we describe here is different

from those described for the polarity cues Afadin or E-cadherin, which instruct the recruitment of LGN to the lateral cortex. In sharp contrast to our results with ANXA1, loss of E-cadherin or Afadin impairs the recruitment of LGN and NuMA to the cortex. Similarly, loss of SAPCD2 abrogates the cortical recruitment of NuMA. Our results also suggest that in contrast to Afadin, E-cadherin and SAPCD2, ANXA1 does not act on the assembly of LGN-NuMA complex, but rather instructs their proper distribution to the lateral cortex after G_{α} -mediated cortical recruitment of LGN. Our revised discussion now addresses these important differences.

As discussed above in response to the Reviewer's Comment #3, we demonstrate that our 3D mMEC culture model is the most appropriate to accurately investigate the mechanisms of mitotic spindle orientation in the mammary epithelium and to determine how this links to lumen formation. This 3D culture model derives from primary luminal progenitors that are purified from pregnant mice using challenging protocols that require multiple optimisation rounds. Additionally, we used with success a lentiviral transduction assay to knockdown ANXA1 in mMECs, using shRNAs. We developed and optimised immunofluorescence protocols combined with high-resolution imaging to visualise with high precision mitotic spindle orientation, epithelial polarity, and the localisation of LGN and ANXA1 which is already challenging in 2D culture.

To our knowledge, we provide the first characterisation of the dynamics of mitotic spindle orientation throughout mammary cystogenesis and establish mMEC 3D cultures as a unique model system to study the link between OCDs and lumen formation. We would like to emphasise here that we have extensive experience with 3D cultures of MDCK, MCF10A, mMEC and hMEC (Elias et al., 2015 PLOS Biology; Ahmed et al., 2016 Development; Caruso et al., 2022 Frontiers in Physiology). We have tested several of these 3D culture systems in the context of our manuscript and as discussed above (in response to Comment #3), we demonstrate in the revised manuscript that both MCF-10A and hMEC derived 3D cultures are not suited to address these important questions in mammary epithelial biology.

In response to the Reviewer's comment about the impact of ANXA1 on polarity, we would like to highlight our results in the initial submission showing that ANXA1 depletion in mMEC acini affects the localisation of E-cadherin during cystogenesis, which indicates defects in cell polarity (Fig. 4j, l). We also include new data below for the Reviewer showing that ANXA1 depletion in 3D mMEC acini not only results in epithelial multi-layering (Fig 4i), but this also leads to epithelial cell architecture defects, represented below by measures of cell size (Reviewer Figure 2), reflecting impaired epithelial integrity.

Reviewer Figure 2. Effect of ANXA1 depletion in 3D mMEC acini on epithelial cytoarchitecture. ANXA1 depletion leads to epithelial multilayering generating a new population of cells that are smaller than those that contribute to the lumen.

-It would also be interesting to have a short discussion on ANXA1-LGN relatively close interaction as observed by immunoprecipitation and one of the microscopy tests, and their nonoverlapping distribution shown by immunostaining.

- As the Reviewer correctly pointed out, our immunoprecipitation and PLA experiments validate the association between ANXA1 and LGN. Our high-resolution fluorescence imaging adds information about the spatiotemporal and dynamic co-distribution of the two proteins throughout mitosis. We would like to clarify that while their accumulation during prometaphase is different, LGN and ANXA1 remain together and co-localise. Our findings suggest that ANXA1 may regulate the amount of LGN and mediate its translocation from the central cortex to accumulate at the lateral cortex. A similar role has been described for SAPCD2, which interacts with LGN, but negatively regulates its cortical accumulation to ensure a balance between planar and perpendicular divisions (Chiu et al., 2016 Dev Cell). Given that we did not address this observation further in this initial study, additional speculations about the precise mechanisms are unwarranted. It will be interesting to investigate how ANXA1 and LGN assemble and how it synergises G_α-mediated cortical recruitment of LGN and with the action of other polarity cues such as Afadin, E-cadherin and SAPCD2.

-page 17, it is stated that ANXA1 marks a subset of luminal epithelial cells in the murine mammary gland. But there is no further discussion about this fact. Most importantly, it would have been essential to check if it is also the case in the human mammary gland, especially in light of the interest for the mechanisms of cancer onset.

- To address the Reviewer's point, we have added data showing the expression and localisation of ANXA1 in the human healthy breast tissue (Supplementary Fig. 6a) (see also our response to the Reviewer's Comment #3). We have amended the discussion accordingly, as suggested by the Reviewer (page 19-20).

-A major gap in the discussion is the relation between Annexin A1 and the PI3K pathway that controls polarity and that has been linked by others to ANXA1 and also to mitotic spindle orientation control in models of human breast epithelial differentiation. This is an important aspect of the discussion because PI3K is an essential pathway in breast cancer development and the authors emphasize the role of misorientation of the spindle in cancer onset. Yet, others have shown that PI3K controls spindle orientation and there is no discussion based on that literature.

- While we agree with the Reviewer that exploring the functional relationship between ANXA1 and PI3K will be interesting, given the important role also played by Akt-PI3K axis in mitotic spindle orientation in mammary cells as shown, for example, in the two important papers cited by the Reviewer also cite above in their Comment #3 (Bazzoun et al. J Cell Sci 2019; Tenvooren et al., oncogene 2019). However, addressing this pathway is beyond the scope of this study where we focus on the crosstalk between the mitotic spindle and the cortex and how this is mediated by our newly identified ANXA1-LGN axis. Additionally, despite many correlative studies showing the influence of ANXA1 on Akt-PI3K signalling, the precise underlying mechanisms remain unknown: there is no evidence of a direct relationship between ANXA1 and PI3K (see Foo et al., 2019, Trends in Mol Med). We elected not to include the Akt-PI3K signalling in the manuscript's discussion.

The authors state that lumen expansion is driven by planar cell divisions independently of apical polarity (see page 16), but there is no strong or detailed explanation regarding why they can make such a conclusion.

- We thank the Reviewer for raising this important point in our study. As we explain in the discussion that in contrast to canine MDCK, the most widely used cells to study the mechanisms of lumen formation in 3D culture, our observations show that lumen expansion does not require the maturation of the apical surface. In MDCK cells the maturation of the apical membrane at the very early stages of cystogenesis (*i.e.* 2-cell stage) is essential to form a central lumen that expands with the help of planar cell divisions (Roman-Fernandez 2016 Traffic). Therefore, apical polarity

and planar cell divisions are coupled in MDCK to drive lumenogenesis. However, in our mMEC 3D culture model, our observations in Figure 4b suggest that cells first undergo planar cell divisions to expand the lumen (between day 2 to day 4), then cease to divide at day 5 to mature their apical surface around a central lumen. As requested by the Reviewer, we now clarify this point further in the revised discussion (page 18-19).

-The next important question to address in the discussion is possibly what makes ANXA1 locate to the lateral cell cortex.

- We would like to clarify that our findings do not suggest that ANXA1 localises to the lateral cortex. During metaphase ANXA1 has rather a homogeneous cortical distribution. In our initial manuscript we discussed that ANXA1 is a membrane associated protein that is recruited to the plasma membrane through its binding to S100A11 (also identified in our LGN interactome) in a Ca²⁺-dependent manner. As also discussed in our response to Reviewer #1 (Major Point #4), we have now added new experiments where we treated MCF-10A cells with TG100-115 (20 µM for 2h), a potent and specific inhibitor of the TRPM7 kinase that phosphorylates ANXA1 on Ser5, modulating its interaction with S100A11 and recruitment to membrane phospholipids. Our new data (new Supplementary Fig.3) show that MCF-10A treatment with TG100-115 abrogates ANXA1 localisation to the plasma membrane, resulting in aberrant cortical accumulation of LGN, NuMA and p150^{Glued}, similarly to the results in ANXA1-depleted cells. As requested by Reviewer #2, we discuss further the possible mechanisms that may regulate the assembly of ANXA1 and LGN at the cell cortex (page 16-17).

Minor comments

1. Materials and Methods: Cell extracts and immunoblotting-- give antibody concentrations instead of dilution as the latter may differ depending on the concentration of the stock of antibodies.

- We provide in the Methods all the details about commercial antibodies used in this study, including the species, company reference numbers (which upon search provide stock concentration of each antibody) and dilutions used. These are the details that have been provided in our previous studies, but also in most peer-reviewed publications. Therefore, since this does not affect the quality and standard of our experiments, we prefer to keep information about the antibody dilutions as we provided them in the initial manuscript.

2. Materials and Methods: it is cell culture MEDIUM when singular, not media! Media means the Press.

- As requested by the Reviewer, we have now replaced media by medium, as appropriate, in the revised manuscript.

3. Page 25: it should be fluorescence intensities instead of fluorescent intensities.

- We thank the Reviewer for spotting this error, which we corrected.

4. Figure 1: results of sections c through g are in general difficult to interpret.

- We have checked Fig 1c-g legends and the corresponding text in the results section. We have not identified any issues. However, we will be happy to address this further if the Reviewer clarify their point.

5. Figure 2a: images are not very convincing with regards to the statement made in the text.

- We would like to emphasise that we perform all our live imaging of GFP-LGN experiments in fully polarised MCF-10A cells. Therefore, not only we image dividing cells, but also the neighbouring cells, which also express GFP-LGN in their cytoplasm with some varying intensities. Notably, live imaging of a highly dynamic protein such as LGN during mitosis is challenging, but we generated

results of the highest quality as in Kiyomitsu and Cheeseman, 2012 NCB, in HeLa cells. In the control conditions, we see clearly GFP-LGN at the lateral cortex during metaphase, we can also distinguish the different phenotypes induced by ANXA1 depletion and listed in our quantifications.

6. Figure 3a: tell what the arrowheads indicate in the figure legend; anomalies listed in 'e' should be clearly shown in representative images in 'a'.

- As requested by Reviewer #2, we have now explained in the figure legend what the arrowheads shown Fig. 3a indicate. We thank the Reviewer for pointing this omission out.

7. Figure 4i: The abnormal piling up of cells in the acinus should be shown with an arrow

- We would like to clarify that the abnormalities listed in Fig. 3e were obtained from our live imaging experiments. The live-imaging photographs in Fig. 3a are representative of these phenotypes.

Reviewer #3 (Remarks to the Author)

For transparency and ease of interpretation, please include a supplementary table with all proteins identified in the IP experiments, including the number of peptides identified per protein. Raw files were deposited to PRIDE, but there are no summary tables with this information.

- As requested by the Reviewer, we have now included a summary table of our raw proteomic data (new Supplementary Table 1).

The text on page 6 suggests that 'singletons' were removed from consideration. My understanding of 'singleton' is a protein identified by a single peptide hit. 10 out of 18 proteins in the 18 protein network are detected with only one peptide in experiment 1 (NUMB, PLK1, CCNB1, DCTN2, S100A11, SAA1, PARD3, HAUS6, TUBG1). What does singleton mean in this context? Please clarify.

- We apologise for this confusion. We used the term singleton to refer to proteins that did not have any connections/interactions in the network of their subgroup, during the construction of the STRING subnetworks. For more clarity, we have removed this term from the revised text.

Page 7, "...were found in the LGN protein network by co-purifying with with NuMA. We also identified the oncogene EIF3E (Eukaryotic translation initiation factor 3 subunit E)46, 47, in the NuMA subcomplex. Finally, we identified a ternary complex comprising the membrane-associated protein Annexin A1 (ANXA1) 48 and its partner S100A11 (S100 Ca²⁺-binding protein A11) 49, as well as the Serum Amyloid A-1 protein (SAA1) 50, that associates with LGN." Please refer to these protein-protein interactions as sub-networks. STRING connections do not necessarily indicate physical protein complexes, which is what is implied in the text.

- As requested by the Reviewer, we have replaced the term subcomplex by subnetwork to reflect better our proteomic findings (page 7).

Reviewers' comments:

Reviewer #1 (Remarks to the Author):

The authors have added plenty of new data in their revised manuscript and tried to link the function of Annexin A1 with spindle positioning. However, as pointed out earlier this is no clear mechanism for how ANXA1 influences LGN and NuMA localization. To my surprise, the authors did not attempt to address the points I raised earlier and mentioned a few times that 'this is beyond the scope of this manuscript'. For this reason, I cannot recommend publishing the manuscript even after revision.

I will now try to explain my concerns below.

1. As mentioned earlier (previous point 1)-The authors identified LGN as a potential binding partner for ANXA1. Why LGN and ANXA1 show mutually exclusive cortical distribution in prometaphase remained unclear and not addressed.
2. Depletion of ANXA1 significantly impact cortical actin (Supplementary Fig. 5E). It is known that altering actin dynamics significantly affects cortical NuMA. Thus it may well be that impact on LGN or NuMA in mitosis upon ANXA1 depletion is simply because of its effects on the actin cytoskeleton rather than directly related to its binding with LGN. Also, do the authors know if the interaction between LGN and ANXA1 is actin-independent?
3. In cells depleted for ANXA1, the authors noted an abnormal spindle (Fig. 3). Thus it is more likely that ANXA1 siRNA cells have a widespread impact on astral microtubule dynamics, leading to spindle orientation defects. This point was also raised earlier.
4. In general, cells depleted for ANXA1 are generally compromised for actin and microtubules. The phenotype that the authors are reporting could simply be because of its broad impact on the cytoskeleton rather than a specific impact on the LGN-NuMA-dynein-dynactin complex. In fact, Figure 2a shows that cells depleted for ANXA1 take longer for cell rounding, suggesting that cortical actin is significantly perturbed in these cells.

Minor points:

1. It remained unclear to me if GFP-LGN is functional? The author should deplete endogenous LGN and test if the spindle positional defect seen upon endogenous depletion is rescued in cells expressing GFP-LGN.
2. PLA data shown in Figure 1e is not convincing. Authors must add more positive and negative controls in the experiment. Loss of LGN and ANXA1 colocalization could be because of the low level of LGN in interphase as shown previously (it is a cell cycle-regulated protein, see Du and Macara, 2004).
3. Still, I found issues related to proper citations- for instance, the authors mentioned that GFP-LGN progressively decreases in anaphase and cite reference 6-which is incorrect. These errors exist throughout the manuscript.
4. Authors could mutate ANXA1 to its phosphodead form at serine 5 at its N-terminus and could test if its localization/and thus LGN localization is affected in cells depleted of endogenous ANXA1-rather than purely relying on the drug.
5. The cortical localization of ANXA1 in MCF10A cells is not convincing (Supplementary Figure 6D), it appears to me that it localizes to the actin cloud next to the cell cortex rather than at the cell cortex.

Reviewer #2 (Remarks to the Author):

The authors have addressed my concerns with the original manuscript. I have no additional comments on the proteomics experiments conducted.

Reviewer #3 (Remarks to the Author):

The authors are submitting a revised version of a manuscript that, initially, was not given further consideration by this journal. Comments from previous reviewers are addressed in details to support findings that Annexin A1 participates in planar mitotic spindle orientation by influencing the corticolateral distribution of LGN.

The previous version was mainly lacking convincing data to support the claims of the authors; in other words, some of the statements regarding Annexin A1's role were beyond what the data were showing. Moreover, it was not clear that their findings was bringing significantly novel information regarding the mechanisms that control mitotic spindle orientation. The claims regarding Annexin A1 have been toned down to better reflect the results shown, and additional results have been included; however, the manuscript still lacks some rigor in the text to clearly represent the situation with Annexin A1, as described in the major comments below.

Comment 1: The title states 'mammalian epithelial morphogenesis', yet such morphogenesis does not necessarily mean lumen expansion; it can be simply lumen formation. In the resting gland, the lumen might be very tiny as encountered in many cases representing real human tissues. It should be clearly stated from the start (as the authors do in the responses to comments) that it is about lumen expansion and not just any acinar morphogenesis, and a definition of lumen expansion should be given at that time. Indeed, in order to create this situation, the authors need to use mammary luminal cells from pregnant mice since the human models of mammary epithelial morphogenesis do not provide this particular type of morphogenesis. Lumen formation and lumen expansion should not be used interchangeably.

Comment 2: As explained by the authors in their responses to the reviewers' comments, the MCF10A model is not adequate because of the irrelevant apoptosis leading to a large central hole; hence there is no 'lumen expansion' process. In fact, according to the accepted definition of a lumen, there is no lumen in MCF10A structures in 3D culture because there are no tight junctions at the apicolateral membranes. The term lumen should be replaced with a better term in the text when referring to the MCF10A structures. Overall, the lack of correct use of the term lumen and the confusion between any morphogenesis and specifically lumen expansion is misleading to the readers.

Comment 3: The story as presented is still a little convoluted. Many proteins display a different level of expression in cancerous tissues compared to normal tissues; it does not mean that they are essential to the cancerous behavior. It is not clear why it seems so important to the authors to mention that ANXA1 is differentially expressed in breast cancers. Most likely, there is no need to mention this information until the discussion since the manuscript should be focused on the control of mitotic spindle orientation in normal tissue. Without further experiments related to cancer onset, it seems distracting to present the rationale for looking at ANXA1 in the introduction as partly related to cancer onset, especially since key results are mainly obtained with a murine lumen expansion model.

Comment 4: The authors should be mindful of the fact that it is not because a majority of people do something a certain way (like indicating dilutions vs. concentrations of antibodies when this information is available), that it makes this way the correct or appropriate one. Using as an argument the fact that they have always used dilutions and not concentrations and published this way is not a convincing one. The purpose of the materials and methods section is to allow the reader to be able to

use the techniques or approaches with the most pertinent information.

Minor comments

Comment 1: Throughout the manuscript, few sentences are missing a word necessary to comprehend the sentence.

Comment 2: There should be a space between numbers and units: e.g., 10 ng/ml instead of 10ng/ml

Comment 3: The main new experiment to ascertain the role of Annexin A1 on LGN is the rescue of the ANXA1 normal phenotype shown figure 2. It seems that all the other new data are placed in supplementary figures. The rule is that data in supplements are not necessary to bring convincing results to support the main story. It is not sure that new supplementary figure 3 should be considered supplementary, as it is an important added control.

Point-by-point response to Reviewers:

The authors have added plenty of new data in their revised manuscript and tried to link the function of Annexin A1 with spindle positioning. However, as pointed out earlier this is no clear mechanism for how ANXA1 influences LGN and NuMA localization. To my surprise, the authors did not attempt to address the points I raised earlier and mentioned a few times that 'this is beyond the scope of this manuscript'. For this reason, I cannot recommend publishing the manuscript even after revision.

We thank Reviewer #1 for commenting that we have added substantive new data to address the Reviewers comments and to significantly improve our revised manuscript, this was also a remark made by both Reviewers #2 and #3. However, we respectfully disagree with Reviewer #1 when they say we did not attempt to address their points about how ANXA1 influences LGN and NuMA localisation. We have addressed all their points by including new experiments demonstrating that ANXA1 expression and localisation at the plasma membrane is sufficient to control the polarised localisation of LGN and NuMA to the lateral cortex (revised Figure 2d-f and new Figure 3). Reviewer #3 commented that these new experiments allow to ascertain the effect of ANXA1 on LGN. To address Reviewer #1 previous Major Point 1, we also generated a new MCF-10A clonal cell line co-expressing of GFP-LGN and ANXA1-mCherry and performed live imaging during mitosis. However, as shown in the live imaging data that we have included in our previous Point-by-Point Response to Reviewers, we couldn't analyse with enough precision the dynamics of ANXA1-mCherry due to the widespread distribution of ANXA1 in the cytoplasm. We have provided a candid explanation of why those experiments were challenging and required extensive and time-consuming optimisation and new microscopic approaches; therefore, we said 'this is beyond the scope of this manuscript'. Nonetheless, in response to Reviewer #1's previous Major Point 1, we have provided new immunofluorescence data using the same MCF-10A cells co-expressing GFP-LGN and ANXA1-mCherry demonstrating that these ectopically expressed proteins behave faithfully as their endogenous counterparts throughout mitosis (revised Supplementary Figure 2d). Future work developing new approaches for higher resolution live imaging will be essential to address the important question of the spatiotemporal dynamic regulation of ANXA1 along with LGN during mitosis.

I will now try to explain my concerns below.

1. As mentioned earlier (previous point 1)-The authors identified LGN as a potential binding partner for ANXA1. Why LGN and ANXA1 show mutually exclusive cortical distribution in prometaphase remained unclear and not addressed.

We would like to remind Reviewer #1 that in our response to their previous Major Point 1, we clarified that our data, indeed, could not allow us to conclude that ANXA1 and LGN were mutually exclusive at the cortex, and toned-down our conclusion (page 8). Rather, we discussed that our results suggest that ANXA1 facilitates the translocation of LGN from the central cortex during prometaphase and then ensures its stable accumulation at the lateral cortex during metaphase. Importantly, throughout the manuscript, our results and quantifications show that ANXA1 and LGN maintain a co-distribution at the cortex during both prometaphase and metaphase. We have addressed this important point, already, in our previous revised discussion (pages 16-17) in comparison with similar mechanisms mediated by the polarity proteins ABL1 (Matsumura et al., 2012 Nat Comms) and SAPCD2 (Chiu et al., 2016 Dev Cell). We would like to point out that SAPCD2 has also been shown to bind to LGN using similar proteomic and biochemical approaches to ours, but SAPCD2 controls the abundance and distribution of LGN between the lateral and apical cortex to regulate the balance between planar and perpendicular divisions (Chiu et al., 2016). We concluded our previous revised discussion with the following statement "Structure-function studies will be key to dissect the mechanisms underlying the assembly and spatiotemporal dynamics of the

ANXA1-LGN-NuMA complex at the cell cortex and understand how ANXA1 acts with NuMA and synergizes with G_{ai} to restrict the cortical accumulation of LGN.” (page 17). We have now discussed and addressed this point further in the new revised manuscript (pages 16-17). As also discussed above, future work developing new approaches for higher resolution live imaging will be key to accurately assess the spatiotemporal dynamic distribution of ANXA1 along with LGN during mitosis.

2. Depletion of ANXA1 significantly impact cortical actin (Supplementary Fig. 5E). It is known that altering actin dynamics significantly affects cortical NuMA. Thus it may well be that impact on LGN or NuMA in mitosis upon ANXA1 depletion is simply because of its effects on the actin cytoskeleton rather than directly related to its binding with LGN. Also, do the authors know if the interaction between LGN and ANXA1 is actin-independent?

Reviewer #1 is right; regulation of cortical F-actin organisation affects the localisation of LGN-NuMA at the cell cortex during mitosis. We would like to remind Reviewer #1 that in our discussion we have cited several studies to address this important point, for example we cited work from Marina Mapelli’s lab (Carminati et al., 2016, Nat SMB) demonstrating the importance of the interplay between F-actin and NuMA-LGN in the regulation of oriented cell division in HeLa cells. However, in sharp contrast to these studies showing that F-actin-mediated regulation affects the recruitment of LGN and NuMA to the cell cortex, our findings show that while ANXA1 knockdown impairs the integrity of F-actin this affects specifically the polarised cortical distribution of LGN-NuMA during mitosis, but not their cortical recruitment. This is an important difference that we want to highlight for Reviewer #1. In the previous revision of our manuscript, we discussed that ANXA1 could also act through the regulation of actin dynamics (page 17). ANXA1 has been shown to regulate the dynamics of actin through direct binding or interaction with the actin polymerising profilin (see references 54, 55, 74, 77 in the first revised manuscript). ANXA1 also interacts with vimentin which has recently been shown by Ewa Paluch lab to regulate, through F-actin, the mechanics of cell division in HeLa cells (Serres et al., 2020 Dev Cell). As we also discussed in the previous revision, whether ANXA1 binds concomitantly to LGN and F-actin or competes LGN from F-actin remain outstanding important questions that we aim to investigate in the future using structure-function and competition assays, combined with high resolution imaging.

In response to Reviewer #1’s question, we have added new experiments, in the new revision, where we treated MCF-10A with a low dose of latrunculin A (1 μ M for 30 min) and show that latrunculin A-mediated depolymerisation of cortical F-actin impairs the recruitment of LGN-NuMA to the cell cortex, similarly to what has been shown by Marina Mapelli’s lab in HeLa cells (Carminati et al., 2016, Nat SMB), but has no effect on the accumulation of ANXA1 at the plasma membrane (revised Supplementary Figure S4g-i). These new results, together with our findings demonstrating that ANXA1 acts specifically on the restricted cortical distribution of LGN-NuMA but not their recruitment, further reinforce a model where ANXA1 acts as an upstream cue to regulate LGN-NuMA independently of F-actin. In the light of these new results, we have improved the discussion in the new revised manuscript (pages 17-18).

3. In cells depleted for ANXA1, the authors noted an abnormal spindle (Fig. 3). Thus it is more likely that ANXA1 siRNA cells have a widespread impact on astral microtubule dynamics, leading to spindle orientation defects. This point was also raised earlier.

We would like to point out that in their previous Major Point 2 Reviewer #1 suggested that the effect of ANXA1 on spindle assembly and dynamics was rather due to defects in chromosome alignment. We are delighted that Reviewer #1 has now agreed that ANXA1 can exert its function through the regulation of the cell cortex. As also discussed below in response to Reviewer #1’s Major Point 4, a tight regulation of the interplay between the microtubule and F-actin cytoskeleton and cortical force generators is essential for correct spindle orientation. While we cannot fully rule out a global effect of ANXA1 knockdown on astral microtubules, our

new results in cells treated with latrunculin A showing that F-actin depolymerisation does not affect ANXA1 localisation to the plasma membrane and highlighting different effects between ANXA1 (Figure 2; new Figure 3) and F-actin (revised Supplementary Figure S4g-i) on LGN and NuMA, with ANXA1 directing their polarised cortical accumulation and F-actin regulating their cortical recruitment, further reinforce our model where ANXA1 acts as an upstream cue to control the restricted cortical distribution of LGN-NuMA, which in turn ensures a focalised crosstalk between F-actin and astral microtubules to generate balanced cortical forces for proper mitotic spindle orientation. Consistent with this, recent studies have shown that correct targeting of NuMA and its binding to LGN and astral microtubules are required for the dynamic crosstalk between microtubules and the cortex and for the stabilisation of dynein on astral microtubules to generate balanced forces that orient the mitotic spindle (Carminati et al., 2016 Nat SMB; Okumura et al., 2018 Elife; Pirovano et al., 2019 Nat Comms). Thus, our results indicate that the observed defects in astral microtubules and spindle orientation in ANXA1-depleted cells are likely to be due to a specific effect of the mislocalisation of LGN-NuMA at the cell cortex upon ANXA1 knockdown.

4. In general, cells depleted for ANXA1 are generally compromised for actin and microtubules. The phenotype that the authors are reporting could simply be because of its broad impact on the cytoskeleton rather than a specific impact on the LGN-NuMA-dynein-dynactin complex. In fact, Figure 2a shows that cells depleted for ANXA1 take longer for cell rounding, suggesting that cortical actin is significantly perturbed in these cells.

This point brings together Reviewer #1's Major Points 2 and 3. Increasing evidence shows the importance of the crosstalk between the plasma membrane, cortical force generators, and F-actin and microtubule cytoskeleton in the regulation of the mechanics of mitosis, this is discussed in our response above and beautifully reviewed by Marina Mapelli's lab (Rizzelli et al., 2020 Open Biology). Yet, the mechanisms regulating this crosstalk remain unclear in mammalian epithelial cells. Our new results (see response to Major Point 2), together with those included in the previous revision indicate that ANXA1 acts at the plasma membrane to specifically control the cortical distribution of LGN-NuMA and thereby cortical recruitment of Dynein to generate balanced forces on astral microtubules that ensure correct mitotic spindle assembly and orientation. However, as also discussed above, the fact that ANXA1 has been shown to regulate actin dynamics, this could suggest that ANXA1 action on mitotic spindle orientation could also be mediated through a direct effect of ANXA1 on cortical actin integrity. An interesting idea that emerges from our findings is that ANXA1 could act at the plasma membrane as a molecular link ensuring a proper crosstalk between the mitotic spindle, astral microtubules, cortical F-actin, cortical force generators and plasma membrane to regulate cell mechanics, which is important for mitotic spindle orientation (see Rizzelli et al., 2020 Open Biology). As we have also discussed in our previous revised manuscript, it is our aim to investigate the molecular mechanisms linking plasma membrane-bound ANXA1 to LGN and F-actin to understand how ANXA1 regulates astral microtubule-cortical actin crosstalk to ensure balanced pulling forces that orient the mitotic spindle to its correct position.

Minor points:

1. It remained unclear to me if GFP-LGN is functional? The author should deplete endogenous LGN and test if the spindle positional defect seen upon endogenous depletion is rescued in cells expressing GFP-LGN.

We would like to emphasise that our proteomic experiments analysing the interactome of GFP-LGN in mitotic MCF-10A cells reveal that LGN's major functional partner, namely NuMA, and another key binder INSC, in addition to key components of the spindle orientation machinery (e.g., Dynein and Dynactin) co-purify with GFP-LGN, validating that GFP-LGN is functional. It is also important to note that, as detailed in the manuscript's Methods, we use the same pTK14-GFP-LGN construct that was previously characterised and shown to

functionally complement depletion of endogenous LGN for NuMA and p150^{Glued} localisation (Kiyomitsu and Cheeseman, 2012 Nat Cell Biol). Additionally, our live imaging and immunofluorescence experiments throughout the manuscript show that GFP-LGN behaves similarly to endogenous LGN. Finally, we have also included important controls that show that expression of GFP-LGN do not affect the dynamics or progression of mitosis.

2. PLA data shown in Figure 1e is not convincing. Authors must add more positive and negative controls in the experiment. Loss of LGN and ANXA1 colocalization could be because of the low level of LGN in interphase as shown previously (it is a cell cycle-regulated protein, see Du and Macara, 2004).

In response to Reviewer #1's point, we have included below the negative and positive controls that we have used to validate our PLA data presented in the manuscript's Figure 1e (Reviewer Figure 1). We would like to clarify that our PLA data clearly shows that proximity between ANXA1 and LGN is mostly in the cytoplasm of interphase MCF-10A cells, whereas this proximity is evident at the cell cortex in metaphase cells. As rightly pointed out by Reviewer #1 we are also aware that LGN is a cell-cycle dependent protein, explaining why we performed our proteomics in both interphase and metaphase MCF-10A cells. It is important to remind Reviewer #1 that throughout our manuscript we show that ANXA1 is more abundant at the cell cortex even during interphase whereas LGN has a cytoplasmic distribution. Therefore, increased proximity between LGN and ANXA1 at the cell cortex during metaphase is likely due to the mitotic-dependent translocation of LGN to the cortex. This is supported by the positive control included in Reviewer Figure 1, where proximity between endogenous and GFP-LGN is in the cytoplasm during interphase, and specific to the cell cortex in metaphase. Of note, our western blot analysis (Supplementary Figure 1d) shows no noticeable differences in the protein levels of LGN between MCF-10A cells synchronised in G2 and metaphase, which corroborates previous work performed in HeLa and MDCK cells about the cell-cycle dependent expression of LGN (Du and Macara, 2004 Cell).

Reviewer Figure 1. Representative confocal images of control conditions of the proximity ligation assay (PLA) in interphase and metaphase MCF-10A cells expressing GFP-LGN. (a) Positive control, GFP-LGN and LGN PLA. Anti-GFP and anti-LGN antibodies were used for the assay. (b) Negative control. Only anti-ANXA1 or anti-GFP antibody was used. DNA (in blue) was visualised with DAPI. LGN expression is shown in green and PLA signals are shown in magenta. Insets on the right show magnified images of the framed region. Scale bars, 10 μ m.

3. Still, I found issues related to proper citations- for instance, the authors mentioned that GFP-LGN progressively decreases in anaphase and cite reference 6-which is incorrect. These errors exist throughout the manuscript.

In the paragraph in page 8 of the previous revised manuscript where we cite reference 6, we say the following "In cells transfected with a control siRNA (si-Control), we observed that GFP-LGN is recruited to the cell cortex during prometaphase to accumulate bilaterally at the cortex opposite to the spindle poles during metaphase (Fig. 2a, Supplementary Movie 1), consistent with previous studies in HeLa cells ⁶. The amounts of cortical GFP-LGN decrease progressively during anaphase, until telophase and cytokinesis where the protein redistributes to the cytosol (Fig. 2a)." We cite this paper (reference 6) to support our observation that GFP-LGN localises at the lateral cortex during metaphase. As clearly shown in our statement we do not cite reference 6 to describe our observation showing a translocation of GFP-LGN to the cytoplasm from anaphase to telophase. We hope that after this clarification, Reviewer #1 now appreciates that we cite reference 6 (Kiyomitsu and Cheeseman, 2012 Nat Cell Biol) at the appropriate place.

4. Authors could mutate ANXA1 to its phosphodead form at serine 5 at its N-terminus and could test if its localization/and thus LGN localization is affected in cells depleted of endogenous ANXA1-rather than purely relying on the drug.

We agree with Reviewer #1 that mutating Ser5 on the N-terminal domain of ANXA1 would be interesting to further ascertain how ANXA1 localisation at the plasma membrane regulates the localisation of LGN-NuMA. We aim to conduct future detailed studies, using inducible plasmid systems to examine the importance of ANXA1 post-translational modifications for the protein's function in the regulation of LGN-NuMA. In response to Reviewer #1 previous Major Point 4, we have used successfully the TG100-115 inhibitor (2h-treatment) to demonstrate that inhibition of the translocation of ANXA1 to the plasma membrane is sufficient to impair the polarised localisation of LGN-NuMA at the lateral cortex during mitosis. Additionally, our rescue experiments show that ectopic expression of ANXA1-mCherry in ANXA-depleted cells is sufficient to restore normal polarised cortical accumulation of LGN-NuMA and planar mitotic spindle orientation (see Figure 2d-f; new Figure 3a, b)

5. The cortical localization of ANXA1 in MCF10A cells is not convincing (Supplementary Figure 6D), it appears to me that it localizes to the actin cloud next to the cell cortex rather than at the cell cortex.

We respectfully disagree with Reviewer #1 that ANXA1 does not localise at the cell cortex in the 3D acini in Supplementary Figure 6d (now Supplementary Figure 5d). Our labelling clearly shows an accumulation of ANXA1 at the cell cortex. As we have already described throughout our manuscript, ANXA1 also localises in the cytoplasm, and around the mitotic spindle during mitosis. Therefore, as pointed out by Reviewer #1, ANXA1 could also localise to the actin cloud underneath the cell cortex.

Reviewer #2 (Remarks to the Author):

The authors have addressed my concerns with the original manuscript. I have no additional comments on the proteomics experiments conducted.

We are delighted that we have addressed all Reviewer #2's concerns.

Reviewer #3 (Remarks to the Author):

The authors are submitting a revised version of a manuscript that, initially, was not given further consideration by this journal. Comments from previous reviewers are addressed in details to support findings that Annexin A1 participates in planar mitotic spindle orientation by influencing the corticolateral distribution of LGN.

The previous version was mainly lacking convincing data to support the claims of the authors; in other words, some of the statements regarding Annexin A1's role were beyond what the data were showing. Moreover, it was not clear that their findings was bringing significantly novel information regarding the mechanisms that control mitotic spindle orientation. The claims regarding Annexin A1 have been toned down to better reflect the results shown, and additional results have been included; however, the manuscript still lacks some rigor in the text to clearly represent the situation with Annexin A1, as described in the major comments below.

We are delighted that Reviewer #3 appreciates our detailed response to Reviewers' comments and acknowledges that the revisions have improved the manuscript and strengthened the conclusions about how ANXA1 regulates LGN to direct planar mitotic spindle orientation. As detailed below we have addressed Reviewer #3 comments to improve the rigor in the text.

Comment 1: The title states 'mammalian epithelial morphogenesis', yet such morphogenesis does not necessarily mean lumen expansion; it can be simply lumen formation. In the resting gland, the lumen might be very tiny as encountered in many cases representing real human tissues. It should be clearly stated from the start (as the authors do in the responses to comments) that it is about lumen expansion and not just any acinar morphogenesis, and a definition of lumen expansion should be given at that time. Indeed, in order to create this situation, the authors need to use mammary luminal cells from pregnant mice since the human models of mammary epithelial morphogenesis do not provide this particular type of morphogenesis. Lumen formation and lumen expansion should not be used interchangeably.

We thank Reviewer #3 for raising this important point; we agree that epithelial morphogenesis involves several processes so it cannot be defined by lumen expansion only. We would like to clarify key findings in our manuscript, which together led us to choose the general term of epithelial morphogenesis in the title and text to describe the phenotypes induced by ANXA1 knockdown in 3D culture. First, our data show that ANXA1 knockdown not only results in collapsed lumen indicating impaired 'lumen expansion', but leads to the formation of multiple small lumens, which reflects defects in 'lumen formation'; we agree with Reviewer #3 that lumen formation and lumen expansion should not be used interchangeably, but our data also indicate that these two processes cannot be uncoupled either, i.e., lumen forms and expands. Hence, for the sake of simplicity, we elected to use lumen formation throughout the initial manuscript. To address the Reviewer's point, we have replaced this by 'luminogenesis', which refers to the entire process of lumen formation initiation and expansion. Of note, the only place where we introduced the term expansion in the previous revised manuscript, we used it along with formation: "lumen formation and expansion is driven by planar cell divisions, independently of apical polarity." (page 19). Second, ANXA1 knockdown results in epithelial cell multi-layering indicating defects in epithelial architecture; this results in multi-lumen formation. It is likely that the generation of inner cells in multi-layered acini leads to physical constraints that prevent lumen expansion. Finally, as shown in our manuscript and previous response to Reviewers, ANXA1 results in defects in cell polarity as evidenced by aberrant E-cadherin localisation and cytoarchitecture. Together our findings show that ANXA1 knockdown affects several processes that underpin epithelial morphogenesis in 3D culture.

Comment 2: As explained by the authors in their responses to the reviewers' comments, the MCF10A model is not adequate because of the irrelevant apoptosis leading to a large central hole; hence there is no 'lumen expansion' process. In fact, according to the accepted definition of a lumen, there is no lumen in MCF10A structures in 3D culture because there are no tight junctions at the apicolateral membranes. The term lumen should be replaced with a better term in the text when referring to the MCF10A structures. Overall, the lack of correct use of the term lumen and the confusion between any morphogenesis and specifically lumen expansion is misleading to the readers.

Reviewer #3 is pointing out to the fact that MCF-10A are not a suitable culture model for the study of lumen expansion in 3D. As highlighted by the Reviewer, we have addressed this issue extensively in the previous revised manuscript. In MCF-10A grown in 3D, centrally located cells are cleared through apoptosis (also referred to as anoikis). This process is often referred to as 'lumen formation' or 'lumen clearing' in the literature, but the correct term to define this process is 'cavitation'. Therefore, as requested by the Reviewer we have replaced lumen formation by cavitation and lumen by cavity when we addressed 3D culture of MCF-10A cells and amended the text to focus on the process of cavitation instead of morphogenesis (pages 13; 19).

Comment 3: The story as presented is still a little convoluted. Many proteins display a different level of expression in cancerous tissues compared to normal tissues; it does not mean that they are essential to the cancerous behavior. It is not clear why it seems so important to the authors to mention that ANXA1 is differentially expressed in breast cancers. Most likely, there is no need to mention this information until the discussion since the manuscript should be focused on the control of mitotic spindle orientation in normal tissue. Without further experiments related to cancer onset, it seems distracting to present the rationale for looking at ANXA1 in the introduction as partly related to cancer onset, especially since key results are mainly obtained with a murine lumen expansion model.

We agree with Reviewer #3 that expression/repression of a given protein in cancer does not mean that it plays a role in carcinogenesis. ANXA1 is established as a biomarker in many tumours including breast cancer, but only a few studies have addressed the underlying mechanisms, which remain poorly characterised. Therefore, in our manuscript we have been careful when we addressed this aspect of ANXA1 biology. Nonetheless, Reviewer #3 is right with their suggestion to wait until the discussion section to talk about ANXA1 in cancer within the context of our findings. We thank the Reviewer for their suggestion, and we have amended the text accordingly in the new revised manuscript.

Comment 4: The authors should be mindful of the fact that it is not because a majority of people do something a certain way (like indicating dilutions vs. concentrations of antibodies when this information is available), that it makes this way the correct or appropriate one. Using as an argument the fact that they have always used dilutions and not concentrations and published this way is not a convincing one. The purpose of the materials and methods section is to allow the reader to be able to use the techniques or approaches with the most pertinent information.

We would like to emphasise that we have provided a very detailed Methods section in our manuscript and shared the scripts of new macros developed for our quantifications, as well as all the generated raw and processed proteomics data. We are in line with Reviewer #3's approach to sharing of data and knowledge. To improve the Methods section, we have included the requested information about antibody concentrations, when this is available on the supplier's antibody specification sheet (pages 26; 28).

Minor comments

Comment 1: Throughout the manuscript, few sentences are missing a word necessary to comprehend the sentence.

We thank Reviewer #3 for pointing out these errors, which we have fixed in the new revised manuscript.

Comment 2: There should be a space between numbers and units: e.g., 10 ng/ml instead of 10ng/ml

These errors have also been fixed in the new revised manuscript.

Comment 3: The main new experiment to ascertain the role of Annexin A1 on LGN is the rescue of the ANXA1 normal phenotype shown figure 2. It seems that all the other new data are placed in supplementary figures. The rule is that data in supplements are not necessary to bring convincing results to support the main story. It is not sure that new supplementary figure 3 should be considered supplementary, as it is an important added control.

We thank Reviewer #3 for pointing this out. Since the data presented in Supplementary Figure S3 are important to provide more mechanistic insights onto how ANXA1 regulates LGN-NuMA, we have now moved this figure to the main manuscript and amended the text accordingly (new Figure 3).

Reviewers' comments:

Reviewer #4 (Remarks to the Author):

Fankhaenel and colleagues identify Annexin A1 (ANXA1) as a novel interactor with LGN in mammary epithelial cells (MEC). Through loss of function (LOF) experiments, using siRNA or shRNA to deplete endogenous ANXA1 in MECs, the authors present data and conclude that ANXA1 is required to restrict LGN distribution to the lateral cortex thereby influencing the planar orientation of the mitotic spindle.

The authors performed LOF experiments via transient transfection in immortal human MCF-10A cells or via lentiviral transduction in primary mouse MECs. Western blot analysis of cell lysates treated by either LOF protocol indicates that the population of treated cells retained expression of ANXA1, and this level of ANXA1 expression varied between experiments (see siANXA1#1 in Figure 2b vs Figure 2e). But, the authors did not measure the residual levels of ANXA1 expressed in the mitotic cells that they examined. Unfortunately, the authors do not clone any of these LOF cells or use genome editing to introduce stable LOF mutation(s). Thus, the experimental protocol is prone to variable levels of ANXA1 depletion both within mitotic cells in each experiment and between experiments.

Given the variable levels of ANXA1 LOF, it is difficult to interpret the multiple LOF phenotypes the authors observed, which include alteration to LGN-NuMA localization but also changes to mitotic kinetics, changes to the mitotic spindle, changes to actin, and changes to astral microtubules. And, as Reviewer 1 indicates, the authors provide no clear mechanism for how loss of ANXA1 alters the phenotypes they observe, including for how ANXA1 influences LGN localization. Indeed, some of the data provided by the authors seems to indicate the localization patterns for GFP-LGN and ANXA1 are mutually exclusive in prometaphase (current Figure 1G). Moreover, the data identifies multiple LOF phenotypes in siANXA1-treated mitotic cells. Each of these phenotypes could explain the changes to LGN localization but none of these ANXA LOF phenotypes are explained mechanistically.

In addition to the criticisms mentioned above, a major concern is the incomplete and likely variable level of ANXA1 depletion obtained in the experiments. Incomplete depletion in a population of cells is likely to give variation both within each experiment and between experiments. The authors do not include ANXA2 staining in siANXA2-treated cells to measure the levels of depletion but rather assume equivalency in a likely heterogeneous population of cells.

In summary, I believe the measurements presented by Fankhaenel and colleagues to be of high quality and the potential new role of ANXA1 to be of interest. But, the paper provides little mechanistic insight and uses LOF experimental models that are nonclonal and are likely to give variable results within and between experiments. Finally, I find many of the conclusions drawn by the authors to be insufficiently supported by the presented data, as discussed below.

Figure 1:

Fankhaenel and colleagues express GFP-LGN in a clonal population of MCF-10A cells. ANXA1 is then identified as a potential novel interactor through LC-MS/MS analysis of GFP immunoprecipitation. The putative interaction was validated by reciprocal co-purification experiments. In addition, the authors used PLA to validate proximity. In their rebuttal, the authors provide negative control experiments (GFP only or ANXA1 only) and positive control experiments (GFP and LGN detection of GFP-LGN), as requested by the reviewer. The authors also provide a line profile for ANXA1 intensity and GFP-LGN intensity that implies anticorrelated cortical intensity during prometaphase (Fig. 1G). The line intensity suggests mutually exclusive cortical distribution, as commented on by Reviewer1 point 1.

Figure 2:

Fankhaenel and colleagues transiently depleted ANXA1 in MCF10A cells with siRNA resulting in incomplete depletion measured 72 hours after transfection (Fig. 2b). This result is not unexpected, as complete depletion may be incompatible with growth, but it does complicate the interpretation of the

LOF phenotypes. That is, it is not completely clear what level of depletion is present in the individual mitotic cells that are being examined. At a minimum, for the analyses performed in fixed samples, the authors should include immunofluorescence analysis of ANXA1 in siANXA1-treated mitotic cells to measure the level of ANXA1 expression, which hopefully will be low or null. However, a more robust method would be to establish edited clones that are ANXA1 $-/-$ or, if not compatible with growth, ANXA1 $+/-$. In the current manuscript, the many LOF phenotypes may be the result of a varying level of ANXA1 depletion both within and between experiments. To appreciate this issue, one can compare the level of depletion with siANXA1 #1 shown in Figure 2b with the level of depletion shown in Figure 2e, which appears to this reader to be different.

siANXA1-treated mitotic cells show changes in cell shape prior to division and in the duration of prometaphase in the movies presented with Figure 2. However, the authors focus their initial analysis on the dynamics or patterns of cortical GFP-LGN, which they refer to as bilateral, unilateral, central and circumferential. In their discussion of the results, they provide % values with 2 significant digits (e.g. 50.37 ± 5.19 %) but I could not find the experimental n for these measurements nor the number of mitotic cells analysed per experiment; the 2 significant digits suggest a level of precision in their measurements that may not be possible given, as mentioned above, they have not confirmed the levels of ANXA1 depletion for the mitotic cells that have been examined. Nonetheless, the authors conclude "ANXA1 acts on LGN restricted accumulation at the lateral cortex". This is one of many examples where the data presented cannot (yet) support the author's conclusions because it implies a direct action by ANXA1 on LGN. As Reviewer 1 points out, the variable changes to LGN localization could be secondary to alterations to cell shape, or changes to cortical actin, or changes to astral microtubule density, etc.

The authors provide a second imprecise conclusion at the end of the next paragraph: "ANXA1 is required to localize the LGN-NuMA-Dynein-Dynactin complex at the lateral cortex during mitosis, independently of Gai". But, a 'requirement for an action' implies that the action does not occur in its absence, and it is apparent in Figure 2D that LGN-NuMA complexes form at the lateral cortex in siANXA1-treated mitotic cells although the localization is different in relation to siControl-treated cells.

So, a key question to answer is whether LGN localization is altered in siANXA1-treated mitotic cells that possess "normal segregation", as indicated in Figure 4e. That is, in a phenotypically normal cell division, in which the authors also confirm ANXA1 is depleted within that specific mitotic cell, is the localization of LGN-NuMA affected? This would be most easily approached, as suggested above, through the generation of clonal cells with ANXA1 LOF mutations rather than transient depletion followed by analysis in a heterogeneous population of cells.

Figure 3:

The authors propose that ANXA1 recruitment to the membrane requires phosphorylation on the N-terminal Ser 5 via TRPM7 channel-kinase. It is difficult to interpret their subsequent experiment wherein MCF10A cells were treated with 20 μ M TG100-115 prior to analysis. This small molecule is advertised as a selective PI3K γ /PI3K δ inhibitor with IC50s of 83 and 235 nM. So, it is hard to interpret the author's findings without first knowing why the drug and dosing strategy was selected. The authors state that the selected dose does not affect cell proliferation but they also argue that TG100-115 phenocopies the effects of siANXA1-treatment, which has dramatic consequences on cell division (Figure 4). To test their hypothesis that N-terminal modification of Ser 5 is an important cue for ANXA1 recruitment, and LGN localization or exclusion, the authors should create and study mutants of this locus.

Figure 4

Figure 4 demonstrate multiple ANXA1 LOF phenotypes. But, the authors do not address whether the mislocalization of LGN is a cause or consequence of, for example, spindle oscillation; similarly, they do not address whether LGN mislocalization is a cause or consequence of changes to cortical actin. Addressing these questions is essential to support the author's conclusion that "ANXA1 acts upstream

of LGN to control mitotic spindle orientation". Similarly, I do not find their analysis of astral microtubule content to be convincing. It would be much more convincing if the authors included EB1 to identify each astral microtubule prior to their measurements.

In summary, the data contained in the manuscript illustrates many consequences for siANXA1 treatment but no direct mechanism of action is provided to explain those effects. The authors do not confirm the efficacy of their ANXA1 depletion in the analysed individual mitotic cells. They do not provide a potential mechanism for how ANXA1 may directly alter LGN localization and they do not exclude the likelihood that these changes are secondary to alterations in the oscillation of the mitotic spindle or changes to the actin cytoskeleton, etc.

Point-by-point response to Reviewer #4:

Thanks very much for sending along the reviews of our paper. We are grateful for the Reviewer's comments, which we have addressed to improve the previous revised manuscript. Please find below our point-by-point response to these comments. We have performed additional experiments and analyses, revised the figures, and modified the text as appropriate to address all the Reviewer's concerns. We have strengthened the text to highlight the significance and novelty of our model of ANXA1-mediated regulation of LGN and planar mitotic spindle orientation in mammalian epithelial cells.

Reviewer #4 (Remarks to the Author):

Fankhaenel and colleagues identify Annexin A1 (ANXA1) as a novel interactor with LGN in mammary epithelial cells (MEC). Through loss of function (LOF) experiments, using siRNA or shRNA to deplete endogenous ANXA1 in MECs, the authors present data and conclude that ANXA1 is required to restrict LGN distribution to the lateral cortex thereby influencing the planar orientation of the mitotic spindle.

The authors performed LOF experiments via transient transfection in immortal human MCF-10A cells or via lentiviral transduction in primary mouse MECs. Western blot analysis of cell lysates treated by either LOF protocol indicates that the population of treated cells retained expression of ANXA1, and this level of ANXA1 expression varied between experiments (see siANXA1#1 in Figure 2b vs Figure 2e). But, the authors did not measure the residual levels of ANXA1 expressed in the mitotic cells that they examined. Unfortunately, the authors do not clone any of these LOF cells or use genome editing to introduce stable LOF mutation(s). Thus, the experimental protocol is prone to variable levels of ANXA1 depletion both within mitotic cells in each experiment and between experiments.

Reviewer #4 is right, there is still residual ANXA1 after siRNA-mediated knockdown. To control for potential variation in the levels of ANXA1 in our knockdown experiments, we performed the following control measurements:

- We quantified the knockdown efficiency shown in the western blots in all our siRNA experiments and confirm that all experiments reach a very high knockdown efficiency of ~72-75% (Reviewer Figure 1a). We would like to bring to the Reviewer's attention that the brightness and contrast in the ANXA1 western blots in Fig 2b and 2e are different, because they are from two different experiments. However, the overall knockdown efficiencies are very similar, as we demonstrate with the quantifications provided below in Reviewer Figure 1a (each dot represents an individual knockdown experiment). We are happy to include these quantifications in the main Figure 2 if the Reviewer thinks it is appropriate. We can also replace the ANXA1 blot in Fig 2e with a brighter alternative from another experiment (see Reviewer Figure 1b). Please note that for transparency, all the uncropped blots are included.
- In addition, we performed immunofluorescence experiments to validate our siRNA strategy at the individual cell level (Reviewer Figure 2). We confirm very high knockdown efficiency of ANXA1 in both interphase and mitotic cells and show that there are no noticeable variations of ANXA1 depletion between cells.
- Furthermore, we performed rescue experiments using an siANXA1-resistant ANXA1-mCherry that allowed restoration of normal mitotic spindle orientation and lateral accumulation of LGN-NuMA in ANXA1-depleted cells (Fig 2 d-f; Supplementary Fig 3 a,b of the manuscript). This further strengthens the conclusions that are drawn from this approach.

With the combination of these approaches, we are confident that our siRNA-mediated knockdown experiments are well controlled and support our conclusions.

Reviewer Figure 1. Western blotting experiments showing highly efficient and consistent knockdown of ANXA1 in MCF-10A using si-RNAs. (a) Histograms showing quantification of ANXA1 relative protein expression obtained from western blotting extracts in si-Control, si-ANXA1#1 or si-ANXA1#2 treated MCF-10A cells (α -tubulin was used as a loading control). All data are presented as means \pm s.e.m. from three independent experiments. *** $P \leq 0.001$ (one-way ANOVA). (b) Western blotting of extracts from si-Control, si-ANXA1#1 or si-ANXA1#2 treated cells expressing or not ANXA1-mCherry.

We would like to emphasise that siRNA-mediated knockdown remains a widely used technology for gene function studies. While we agree that the CRISPR/Cas9 is a powerful technology for gene knockout, results from the labs of David Pellman (Harvard University, USA) and Rudolf Jaenisch (MIT, USA) have shown deleterious on-target effects of CRISPR-based editing on genome stability during mitosis, generating abnormal structures of the nucleus that arise as a consequence of errors in mitosis, including formation of micro-nuclei and chromosome bridges (Leibowitz et al., 2021 Nature Genetics DOI: 10.1038/s41588-021-00838-7 ; Papathanasiou et al., 2021 Nature Communications DOI: 10.1038/s41467-021-26097-y pre-printed in BiorXiv in 2020). These defects have subsequently been confirmed in cell lines widely used to study mitosis (Rodriguez-Munoz et al., 2021 DOI: 10.3389/fcell.2021.745195). These data indicate that CRISPR/Cas9 has its own limitations, just like any other methodology.

Given the variable levels of ANXA1 LOF, it is difficult to interpret the multiple LOF phenotypes the authors observed, which include alteration to LGN-NuMA localization but also changes to mitotic kinetics, changes to the mitotic spindle, changes to actin, and changes to astral microtubules. And, as Reviewer 1 indicates, the authors provide no clear mechanism for how loss of ANXA1 alters the phenotypes they observe, including for how ANXA1 influences LGN localization. Indeed, some of the data provided by the authors seems to indicate the localization patterns for GFP-LGN and ANXA1 are mutually exclusive in prometaphase (current Figure 1G). Moreover, the data identifies multiple LOF phenotypes in siANXA1-treated mitotic cells. Each of these phenotypes could explain the changes to LGN localization but none of these ANXA LOF phenotypes are explained mechanistically.

As detailed throughout our manuscript we have performed 3 independent experiments that allowed us to obtain similar phenotype categories at similar proportions upon ANXA1 depletion using two siRNAs. To address the Reviewer's comment suggesting variable levels of ANXA1

knockdown, we included representative confocal images and quantifications from 3 independent experiments showing that while there is a residual level of ANXA1 upon siRNA treatment, ANXA1 knockdown is homogeneous and not variable between cells (Reviewer Figure 2).

Reviewer Figure 2. Immunofluorescence experiments showing highly efficient and consistent knockdown of ANXA1 in MCF-10A cells using si-RNAs. (a) Representative confocal images from three independent experiments of MCF-10A treated with si-Control, siANXA1#1 or si-ANXA1#2, and immunostained for ANXA1 (grey) and counterstained with DAPI (blue, DNA). Scale bar, 50 µm. (b) Line scans obtained using Fiji software, showing ANXA1 fluorescence intensities in interphase and metaphase MCF-10A cells treated with si-Control, siANXA1#1 or si-ANXA1#2 (a.u. = arbitrary unit). (c) Histograms showing ANXA1 average fluorescence intensities in si-Control, siANXA1#1 or si-ANXA1#2 treated MCF-10A cells from three independent experiments. In each experiment, three max-projection confocal images of (184.61 x 184.61 microns, each) were used to measure ANXA1 fluorescence intensities with Fiji. All data are presented as means ± s.e.m. from three independent experiments. *** P ≤ 0.001 (one-way ANOVA).

Using two siRNAs we show that ANXA1 knockdown impairs the lateral cortical accumulation of LGN and NuMA. These two proteins remain at the cortex upon ANXA1 knockdown, but display random distributions (*i.e.*, central, circumferential, unilateral). These phenotypes are at similar proportions across three independent experiments. Importantly, expression of an siANXA1-resistent ANXA1-mCherry successfully rescues the cortical localisation defects of LGN, NuMA, induced upon depletion of endogenous ANXA1 (see Fig 2 d-f). These rescue experiments not only show a specific effect of ANXA1 knockdown on LGN-NuMA, but also demonstrate that the categories of cortical localisations of LGN-NuMA upon ANXA1 knockdown are not the result of variable levels of ANXA1 knockdown. It is also worth highlighting our live imaging results in Fig 2a, where all the aberrant cortical distributions of LGN (unilateral, central, circumferential) upon ANXA1 knockdown lead to an asymmetric segregation of LGN during telophase. This is another strong evidence that ANXA1 knockdown has a specific and consistent effect of LGN dynamics during mitosis.

This Reviewer mentioned Reviewer #1 comments about the effect of ANXA1 knockdown on actin and microtubules. In our previous response and revision, we have extensively addressed these issues and added new data in Supplementary Fig 4 to complement those in Supplementary Fig 3, where we further demonstrate that ANXA1 acts as an upstream cue to regulate LGN at the lateral cortex. A key finding from these experiments is that ANXA1 knockdown affects cortical F-actin integrity, whereas actin depolymerisation does not affect ANXA1 plasma membrane translocation (Supplementary Fig 4 h-j). Of note, LGN knockdown also does not affect ANXA1 plasma membrane localisation (Supplementary Fig 3 c-f). In our previous revision (pages 16-17) and point-by-point response we have also discussed extensively, supported by our data and the literature, that ANXA1 is likely to act directly on LGN-NuMA, which in turn can influence the actin-microtubule crosstalk. To strengthen this point, previously, we have cited several studies to address this important point, for example we cited work from Marina Mapelli's lab (Carminati et al., 2016, Nat SMB DOI: 10.1038/nsmb.3152) demonstrating the importance of the interplay between F-actin and NuMA-LGN in the regulation of oriented cell division in HeLa cells. However, in sharp contrast to these studies showing that F-actin-mediated regulation affects the recruitment of LGN and NuMA to the cell cortex, our findings show that while ANXA1 knockdown impairs the integrity of F-actin, this affects specifically the polarised cortical distribution of LGN-NuMA during mitosis, but not their cortical recruitment. Our data are also corroborated by important studies, which were cited as well, showing that correct targeting of NuMA and its binding to LGN and astral microtubules are required for the dynamic crosstalk between microtubules and the cortex and for the stabilisation of dynein on astral microtubules to generate balanced forces that orient the mitotic spindle (Carminati et al., 2016 Nat SMB; Okumura et al., 2018 Elife DOI: 10.7554/eLife.36559; Pirovano et al., 2019 Nat Comms DOI: 10.1038/s41467-019-09999-w). These data, together with those showing that ANXA1 localisation is not affected upon LGN knockdown and the successful rescue experiment, reinforce our conclusion that all the observed defects in F-actin and astral microtubules and spindle orientation in ANXA1-depleted cells are likely to be caused by the mislocalisation of LGN-NuMA at the cell cortex upon ANXA1 RNAi-based knockdown.

In our previous point-by-point responses to Reviewer #1 we have made it clear that our results did not allow us to conclude that ANXA1 and LGN were mutually exclusive during prometaphase. We have also discussed this point further in our previously revised manuscript (pages 16-17). Now that this Reviewer is bringing this point again, we would like to say the followings: **1)** our proteomics and interactome uses metaphase cells where we identify ANXA1 as an interactor of LGN; **2)** as this is the first time ANXA1 is assigned a function in mitosis, we looked at the co-distribution of LGN and ANXA1 throughout all phases of mitosis. For the sake of transparency and good research practice we needed to show all our observations, even those that we cannot explain, in this first paper; **3)** our manuscript is focussed on mitotic spindle orientation, which as the Reviewer will agree, happens after the spindle is formed in metaphase. In metaphase when the spindle is centred, we observe a homogeneous plasma membrane distribution of ANXA1 and confirm its co-localisation with LGN; and **4)** it is not uncommon that important proteins in the spindle orientation machinery, such as Dynein, oscillates between the two lateral cortexes during metaphase depending on PLK1 regulation (Kiyomitsu and Cheeseman, 2012 Nat Cell Biol DOI: 10.1038/ncb2440), while NuMA, a key Dynein interactor and regulator, displays a symmetric bilateral distribution. Does this Dynein lateral localisation contradict the finding demonstrating its interaction with NuMA? In our manuscript's discussion (and previous point-by-point response), we also cite another study where SAPCD2 negatively regulates the abundance of LGN at the cortex, while the study shows the two proteins to interact using similar approaches to those described in our manuscript (Chiu et al., 2016 Dev Cell DOI: 10.1016/j.devcel.2015.12.016). We anticipate that a similar mechanism may be at play between ANXA1 and LGN during prometaphase. While it is key to further dissect the dynamic localisation of ANXA1 and LGN during prometaphase in the future, our present study focusses on mitotic spindle orientation in metaphase. As discussed in our revised manuscript (pages 16-17), we aim to further investigate the

spatiotemporal regulation of ANXA1 and LGN during mitosis using advanced live imaging approaches and competition assays that we are optimising in my laboratory, and this will form a basis for future studies/publications.

In addition to the criticisms mentioned above, a major concern is the incomplete and likely variable level of ANXA1 depletion obtained in the experiments. Incomplete depletion in a population of cells is likely to give variation both within each experiment and between experiments. The authors do not include ANXA2 staining in siANXA2-treated cells to measure the levels of depletion but rather assume equivalency in a likely heterogeneous population of cells.

As discussed in our response to the Reviewer's previous points above, and as shown in Reviewer Figures 1 and 2, we obtain similar results across our 3 independent experiments. We have not observed noticeable variations of the levels of ANXA1 knockdown using siRNAs. Moreover, we obtain similar results with two different siANXA1. Importantly, our successful rescue experiments using an siANXA1-resistant ANXA1-mCherry, which allows to restore control phenotypes, further demonstrates the robustness of our knockdown approach and controls used.

Knockdown of key proteins involved in mitosis often result in multiple effects and defects on chromosome, spindle and LGN-NuMA dynamics, in 2D and 3D cell and *in vivo*. We cited some of the studies describing this, in our manuscript, but here are a few examples (Yu et al., 2019 Cell Research DOI: 10.1038/s41422-019-0189-9; Knouse et al., 2018 Cell DOI: 10.1016/j.cell.2018.07.042; di Pietro et al., 2017 Current Biology DOI: 10.1016/j.cub.2017.06.055; Kschonsak and Hoffmann, 2018 J Cell Sci DOI: 10.1242/jcs.214544). Even in normal cells we find some defects in chromosome and spindle dynamics, but at very low frequencies. Knockdown studies of important proteins like ANXA1 will only exacerbate these phenotypes.

In summary, I believe the measurements presented by Fankhaenel and colleagues to be of high quality and the potential new role of ANXA1 to be of interest. But, the paper provides little mechanistic insight and uses LOF experimental models that are nonclonal and are likely to give variable results within and between experiments. Finally, I find many of the conclusions drawn by the authors to be insufficiently supported by the presented data, as discussed below.

We thank the reviewer for remarking that our work is of high quality and for highlighting that the potential new role of ANXA1 is interesting. However, we respectfully disagree with the Reviewer that our conclusions are not supported by the results. Throughout the revision rounds, not only have we toned them down but included substantive new data (which have almost doubled the size of the initial submission) that strengthened our conclusions. We strongly believe that the conclusions are well supported by the provided evidence. Please, see below more detailed explanations.

Figure 1:

Fankhaenel and colleagues express GFP-LGN in a clonal population of MCF-10A cells. ANXA1 is then identified as a potential novel interactor through LC-MS/MS analysis of GFP immunoprecipitation. The putative interaction was validated by reciprocal co-purification experiments. In addition, the authors used PLA to validate proximity. In their rebuttal, the authors provide negative control experiments (GFP only or ANXA1 only) and positive control experiments (GFP and LGN detection of GFP-LGN), as requested by the reviewer. The authors also provide a line profile for ANXA1 intensity and GFP-LGN intensity that implies anticorrelated cortical intensity during prometaphase (Fig. 1G). The line intensity suggests mutually exclusive cortical distribution, as commented on by Reviewer1 point 1.

We discussed this point in detail, above (page 4; paragraph 2). In our revised manuscript (pages 16-17) and previous point-by-point response we clearly say that our results do not suggest that LGN and ANXA1 are mutually exclusive. In prometaphase, ANXA1 and LGN retain a level of co-localisation at the lateral cortex. During metaphase, ANXA1 distribution at the plasma membrane becomes homogeneous where the protein co-localises more with LGN, which accumulates at the lateral cortex. We do not suggest anywhere in the text that ANXA1 and LGN are mutually exclusive. Again, we would like to kindly remind Reviewer #4 that the focus of our study is on metaphase where we investigate the role ANXA1 plays in mitotic spindle orientation, and how the protein regulates LGN-NuMA dynamics at the cell cortex. Therefore, while we characterise the co-distribution of LGN and ANXA1 throughout mitosis – because this is the first study describing a role for ANXA1 in mitosis – and that this brings exciting new observations, we cannot answer all these questions in one single paper. As discussed in the revised manuscript (pages 16-17) and above (page 4, paragraph 2), we aim to investigate the mechanisms that define the spatiotemporal co-distribution of LGN and ANXA1 and how these proteins assemble during mitosis, by combining live imaging and biochemical assays.

Figure 2:

Fankhaenel and colleagues transiently depleted ANXA1 in MCF10A cells with siRNA resulting in incomplete depletion measured 72 hours after transfection (Fig. 2b). This result is not unexpected, as complete depletion may be incompatible with growth, but it does complicate the interpretation of the LOF phenotypes. That is, it is not completely clear what level of depletion is present in the individual mitotic cells that are being examined. At a minimum, for the analyses performed in fixed samples, the authors should include immunofluorescence analysis of ANXA1 in siANXA1-treated mitotic cells to measure the level of ANXA1 expression, which hopefully will be low or null. However, a more robust method would be to establish edited clones that are ANXA1 $-/-$ or, if not compatible with growth, ANXA1 $+/-$. In the current manuscript, the many LOF phenotypes may be the result of a varying level of ANXA1 depletion both within and between experiments. To appreciate this issue, one can compare the level of depletion with siANXA1 #1 shown in Figure 2b with the level of depletion shown in Figure 2e, which appears to this reader to be different.

In our experiments, we performed several controls to assess the knockdown efficiency of the siANXA1 and found that these are more efficient after 72h than 48h. We hypothesise this is likely due to the protein's turnover. Moreover, unlike HeLa cells (the most widely used cells to study mitosis), MCF10A cells need time to polarise and establish cell-cell adhesions after transfection which takes ~48h post-transfection. These are important aspects that we considered carefully to design our experiments investigating the role of ANXA1 in the regulation of polarised epithelial cell divisions, the main point of this study. As discussed above we achieve a 72-75% knockdown using our siRNAs, which is very good with this strategy. We obtain a similar efficiency across 3 independent experiments as evidenced in the data provided above (Reviewer Figures 1 and 2). If needed, we can provide the source Excel files of all our quantifications. As we also suggest above, we can replace the western blot in Fig 2e with an alternative provided in Reviewer Figure 1. We have also done control immunofluorescence experiments, which show efficient knockdown – we have not observed the ANXA1 expression heterogeneity suggested by the Reviewer (see Reviewer Figure 2).

Even within the CRISPR editing era, siRNAs remain a widely used and robust assay of choice to knockdown genes and study their function in a large variety of systems, in culture and *in vivo*. To repeat, our rescue experiments using an siANXA1-resistant ANXA1-mCherry allow restoration of the normal mitotic spindle orientation and lateral accumulation of LGN-NuMA. These experiments demonstrate the specificity and consistency of our siANXA1-mediated knockdown approach. As also discussed in detail above, increasing evidence showing that

CRISPR-based gene editing provokes genome instability and mitosis defects in several systems. If the CRISPR/Cas9 approach was to cause lethal or extremely strong phenotype, we would have not been able to conclude anything from this way of protein depletion. We chose a different, valid, still very frequently used method to achieve this.

siANXA1-treated mitotic cells show changes in cell shape prior to division and in the duration of prometaphase in the movies presented with Figure 2. However, the authors focus their initial analysis on the dynamics or patterns of cortical GFP-LGN, which they refer to as bilateral, unilateral, central and circumferential. In their discussion of the results, they provide % values with 2 significant digits (e.g. 50.37 ± 5.19 %) but I could not find the experimental n for these measurements nor the number of mitotic cells analysed per experiment; the 2 significant digits suggest a level of precision in their measurements that may not be possible given, as mentioned above, they have not confirmed the levels of ANXA1 depletion for the mitotic cells that have been examined. Nonetheless, the authors conclude “ANXA1 acts on LGN restricted accumulation at the lateral cortex”. This is one of many examples where the data presented cannot (yet) support the author’s conclusions because it implies a direct action by ANXA1 on LGN. As Reviewer 1 points out, the variable changes to LGN localization could be secondary to alterations to cell shape, or changes to cortical actin, or changes to astral microtubule density, etc.

Reviewer 4 is right the shape of siANXA1-depleted cells is affected in prometaphase. However, these cells round up well during metaphase and there are no noticeable differences as compared to controls (Fig 2a). The focus of this figure is to address how ANXA1 influences the dynamics of LGN, which accumulates at the lateral cortex during metaphase in control cells, where LGN plays its key function in the regulation of the orientation of the mitotic spindle in metaphase. We would like to point the Reviewer to the fact that while cell mechanics and density regulate mitotic spindle orientation and LGN cortical recruitment (see Rizzelli et al., 2020 Open Biology DOI: 10.1098/rsob.190314), recent work from Martijn Gloerich lab in MDCK polarised epithelial cells showed that that E-cadherin and cortical LGN align epithelial cell divisions with tissue tension independently of cell shape, and that the localisation of E-cadherin at the plasma membrane is key to pattern LGN at the cell cortex (Gloerich et al., 2017 Nat Comms DOI: 10.1038/ncomms13996 ; Hart et al., 2017 PNAS DOI: 10.1073/pnas.1701703114).

We do not understand why Reviewer #4 states that they could not find the information about the number of experiments or number of mitotic cells used in our quantifications. We have provided all the details about experiments in the Methods section and figure legends where we clearly show the N and number of mitotic cells analysed, the number of independent experiments and statistical methods used. We would also like to point the Reviewer to our figures where we use Super-plots to represent our data, where the graphs include the number of independent experiments – we used this representation for transparency about any variability in our experiments (Lord et al., 2020 J Cell Biol DOI: 10.1083/jcb.202001064). As now discussed above and shown in Reviewer Figures 1 and 2, we have provided evidence that our siRNA-mediated knockdown strategy and rescue experiments allow a consistent quantitative examination the function of ANXA1 in mitosis.

Reviewer #4 has used this statement from our manuscript to say our results do not support the conclusion “ANXA1 acts on LGN restricted accumulation at the lateral cortex”. Here is what we say in the manuscript: “We did not find cells negative for cortical GFP-LGN in the absence of ANXA1, allowing us to conclude that ANXA1 acts on LGN restricted accumulation at the lateral cortex, rather than on its recruitment.” This statement aimed to say that because ANXA1 knockdown does not lead to loss of LGN from the cortex, but rather to an impaired accumulation of the protein to the lateral cortex (*i.e.* LGN remains at the cortex but not at the right location as compared to controls), our results suggest that ANXA1 regulates LGN

polarised accumulation to the lateral cortex, upon its $G\alpha i$ -mediated recruitment. To address the Reviewer's point, we have amended the statement in the newly revised manuscript to improve its clarity (page 9).

As we discussed above (page 4, paragraph 1), in our manuscript and previous point-by-point response to Reviewers, together our results and evidence from recent studies allow to conclude that the effect of ANXA1 knockdown on spindle orientation, astral microtubule and F-actin dynamics is likely to be through the action of ANXA1 on the cortical lateral patterning of LGN and NuMA. Increasing evidence shows the importance of the crosstalk between the plasma membrane, cortical force generators, and F-actin and microtubule cytoskeleton in the regulation of the mechanics of mitosis, this is discussed in our response above and beautifully reviewed by Marina Mapelli's lab (Rizzelli et al., 2020 Open Biology DOI: 10.1098/rsob.190314). Yet, the mechanisms regulating this crosstalk remain unclear in mammalian epithelial cells. Our study may contribute significantly to this story and may help to explain how exactly this complicated network works, specifically in the context of epithelial biology. Our results indicate that ANXA1 acts at the plasma membrane as a molecular landmark to specifically control the cortical distribution of LGN-NuMA and thereby cortical recruitment of Dynein to generate balanced forces on astral microtubules that ensure correct mitotic spindle assembly and orientation. This conclusion is also supported by our successful rescue experiments using ANXA1-mCherry, biochemical perturbation ANXA1 membrane translocation using TG100-115, experiments showing that LGN knockdown does not affect ANXA1 plasma membrane localisation. Finally, given that ANXA1 has been shown to interact and regulate actin (discussed in our previous point-by-point response and revised manuscript (pages 17-18)), we cannot rule out the possibility that ANXA1 may also have a direct effect on actin. However, addressing all these intricacies requires a whole new future project.

The authors provide a second imprecise conclusion at the end of the next paragraph: "ANXA1 is required to localize the LGN-NuMA-Dynein-Dynactin complex at the lateral cortex during mitosis, independently of Gai". But, a 'requirement for an action' implies that the action does not occur in its absence, and it is apparent in Figure 2D that LGN-NuMA complexes form at the lateral cortex in siANXA1-treated mitotic cells although the localization is different in relation to siControl-treated cells.

Reviewer 4 is right when they say "a 'requirement for an action' implies that the action does not occur in its absence". We would like to clarify our conclusion and the underlying results which have been discussed in the previous revised manuscript (e.g., page 9; paragraph 2). Indeed, we demonstrate that ANXA1 depletion and mislocalisation from the plasma membrane impairs the restricted patterning of LGN-NuMA to the lateral cortex. This is a significant impairment in the localisation of LGN-NuMA because it causes a loss in the focalised cortical forces on astral microtubules that ensure planar mitotic spindle orientation. Upon loss of ANXA1, LGN and NuMA are randomly distributed at the cortex leading to unbalanced forces. Thus, ANXA1 is required for the accurate polarised cortical patterning of LGN-NuMA; we don't need to lose LGN-NuMA from the cortex to show a requirement for ANXA1. In fact, the entire manuscript is dedicated to highlight the specific role of ANXA1 on the polarised accumulation of LGN-NuMA to the lateral cortex, but not on their recruitment. Therefore, what we show in our revised manuscript, is that ANXA1 acts as a membrane-associated molecular landmark that ensures polarised cortical distribution of LGN-NuMA upon their $G\alpha i$ -mediated recruitment. This is a novel mechanism that we explain in detail in the manuscript, throughout the description of the study. To address the Reviewer point we have now improved clarity of our conclusion in the newly revised manuscript (page 9).

So, a key question to answer is whether LGN localization is altered in siANXA1-treated mitotic cells that possess "normal segregation", as indicated in Figure 4e. That is, in a phenotypically normal cell division, in which the authors also confirm ANXA1 is depleted within that specific

mitotic cell, is the localization of LGN-NuMA affected? This would be most easily approached, as suggested above, through the generation of clonal cells with ANXA1 LOF mutations rather than transient depletion followed by analysis in a heterogeneous population of cells.

Our live imaging data of LGN-GFP in Fig 2a show impaired LGN localisation during mitosis, but also chromosome segregation defects and delays in their alignment in metaphase in the siANXA1-treated cells. As also shown in Reviewer Figure 2 above, we have now provided samples of representative confocal images from three independent experiments showing the high efficiency of our siRNA-mediated knockdown approach and ruling out variable levels of ANXA1 upon siRNA treatments. We ran independent experiments and analysed multiple cells in each experiment. We provide robust statistical analyses here and they show a consistent, reproducible biological effect. Importantly, our successful rescue experiments, and the fact we obtain similar effects on LGN-NuMA using TG100-115, show that our approach of siRNA-mediated knockdown of ANXA1 is appropriate.

Figure 3:

The authors propose that ANXA1 recruitment to the membrane requires phosphorylation on the N-terminal Ser 5 via TRPM7 channel-kinase. It is difficult to interpret their subsequent experiment wherein MCF10A cells were treated with 20 μ M TG100-115 prior to analysis. This small molecule is advertised as a selective PI3K γ /PI3K δ inhibitor with IC50s of 83 and 235 nM. So, it is hard to interpret the author's findings without first knowing why the drug and dosing strategy was selected. The authors state that the selected dose does not affect cell proliferation but they also argue that TG100-115 phenocopies the effects of siANXA1-treatment, which has dramatic consequences on cell division (Figure 4). To test their hypothesis that N-terminal modification of Ser 5 is an important cue for ANXA1 recruitment, and LGN localization or exclusion, the authors should create and study mutants of this locus.

Reviewer #4 is right, TG100-115 is a PI3K inhibitor. However, as we discussed in our manuscript, TG100-115 has been characterised as a potent inhibitor of TRPM7 (e.g. by Song et al. 2017 DOI: 10.1016/j.bbagen.2017.01.034), a channel-kinase that phosphorylates ANXA1 to influence its localisation to the plasma membrane. We have used a 2hr-treatment that has been described in the literature to not affect proliferation, but which we found to be sufficient to impair the localisation of ANXA1 at the plasma membrane during metaphase. This acute treatment, results in similar effects on the localisation LGN-NuMA at the cell cortex as those obtained upon ANXA1 knockdown. Reviewer 4 is right, ANXA1 knockdown results in a reduced number of cells that complete mitosis, but this can be explained by the longer treatment with siRNAs (72h) and the other non-mitotic functions of cytoplasmic ANXA1 (such as vesicular trafficking) that may be affected upon depletion. We also agree with the Reviewer that mutating Ser5 would be important, but as discussed in our previous response to Reviewer #1's minor point 4, we plan to do this in the future as part of a project where we will assess the overall role of post-translational modifications of ANXA1 in its mitotic function. Indeed, we have identified several additional phosphorylation sites, additional to Ser 5, which a new PhD student in my group will investigate in detail. Our experiments with TG100-115 are sufficient to demonstrate that the localisation of ANXA1 to the plasma membrane is essential for its function in the regulation of LGN-NuMA, an experiment that we did in response to Reviewer #1 who requested to demonstrate that the association of ANXA1 to the plasma membrane is sufficient to maintain LGN-NuMA at the lateral cortex during metaphase. It is very likely that the whole pathway, which we are describing in this manuscript for the first time, is regulated by posttranslational modifications at a molecular level, but this is not the focus of the present work.

Figure 4:

Figure 4 demonstrate multiple ANXA1 LOF phenotypes. But, the authors do not address whether the mislocalization of LGN is a cause or consequence of, for example, spindle oscillation; similarly, they do not address whether LGN mislocalization is a cause or consequence of changes to cortical actin. Addressing these questions is essential to support the author's conclusion that "ANXA1 acts upstream of LGN to control mitotic spindle orientation". Similarly, I do not find their analysis of astral microtubule content to be convincing. It would be much more convincing if the authors included EB1 to identify each astral microtubule prior to their measurements.

As discussed above and shown in Reviewer Figure 2 the multiple phenotypes obtained upon ANXA1 knockdown are unlikely to be due to varying levels of residual ANXA1. Additionally, in our previous revised manuscript's Fig 3. Supplementary Fig 3 and 4, we have included substantive data that reinforce our model that ANXA1 is an upstream polarity cue that acts on LGN-NuMA, which cortical localisation has been demonstrated to ensure astral microtubule integrity and thereby microtubule-actin crosstalk (discussed above: page 8 – paragraph 2; revised manuscript: page 17): **1)** we show that while si-LGN affects mitotic spindle orientation and NuMA cortical accumulation, LGN knockdown does not affect ANXA1 localisation to the plasma membrane (Supplementary Fig 3 c-f); **2)** we show that ANXA1 knockdown affects actin and astral microtubule organisation (Supplementary Fig 4 a-g), whereas F-actin depolymerisation using latrunculin A does not affect ANXA1 plasma membrane localisation while abrogating cortical recruitment of LGN-NuMA (Supplementary Fig 4 h-j); **3)** we demonstrate that our rescue experiments using ANXA1-mCherry restores the normal phenotypes including proper spindle orientation (Supplementary Fig 3 a-b) and LGN-NuMA lateral accumulation at the cell cortex (Fig 2 d-f). Together, these different experiments indicate that ANXA1 acts as a plasma membrane-bound upstream polarity cue that regulates mitotic spindle orientation. Building upon these results we have improved our discussion in the previous revised manuscript (pages 17-18) and provided a detailed response in the accompanying point-by-point response to Reviewers. It is an important point, as the Reviewer states, but we have already addressed this matter, as described above.

As we also discussed in our previous revised manuscript and in response to Reviewers, given studies showing that ANXA1 can regulate F-actin, either through direct binding or actin-regulating proteins such as vimentin and profilin (see Discussion, page 18), we cannot rule out another effect of ANXA1 on F-actin organisation during mitosis. However, addressing this important question requires a whole new project that will aim to investigate the molecular mechanisms linking ANXA1 to LGN and F-actin to understand how ANXA1 regulates the astral microtubule-cortical actin crosstalk to ensure balanced pulling forces that orient the mitotic spindle to its correct position. Reviewer 4 is right that mitotic spindle oscillation may affect LGN cortical localisation. While this is an important question, it remains unclear, in the literature, how spindle oscillation could act on LGN-NuMA. Instead, it has been well documented that cortical LGN-NuMA-Dynein define the dynamics and oscillation of the mitotic spindle in several systems (e.g., Peyre et al., 2011 DOI: 10.1083/jcb.201101039; Saadaoui et al., 2014 DOI: 10.1083/jcb.201405060; Kotak et al., 2012 DOI: 10.1083/jcb.201203166; Matsumura et al., 2012 DOI: 10.1038/ncomms163). These studies corroborate our findings indicating that ANXA1-loss-mediated impaired cortical distribution of LGN-NuMA results in unbalanced cortical forces on astral microtubules and impaired mitotic spindle dynamics.

In our previous revised manuscript, we show that ANXA1 depletion results in astral microtubule buckling and elongation, without affecting the overall intensity (Supplementary Fig 4 a,b), using similar quantification methods as in many important papers in the field. Our high-resolution images allowed us to perform robust quantifications and conclude that ANXA1 is important for astral microtubule dynamics. To achieve this, we have optimised an

immunofluorescence protocol that led to a high-quality labelling of astral microtubules (see Methods), which are often overshadowed by the spindle saturating signal when this is labelled using regular/classic microtubule immunofluorescence. To address Reviewer #4, we have now provided new data using clonal stable MCF-10A cells expressing EB3-GFP further indicating that ANXA1 knockdown increases the stability of astral microtubules (new Supplementary Fig 4 b,c), affecting the positioning of the mitotic spindle. All data are consistent with previous observations.

In summary, the data contained in the manuscript illustrates many consequences for siANXA1 treatment but no direct mechanism of action is provided to explain those effects. The authors do not confirm the efficacy of their ANXA1 depletion in the analysed individual mitotic cells. They do not provide a potential mechanism for how ANXA1 may directly alter LGN localization and they do not exclude the likelihood that these changes are secondary to alterations in the oscillation of the mitotic spindle or changes to the actin cytoskeleton, etc.

We do hope that altogether, our detailed response to Reviewer #4 above, the new data and amendments to the new revised manuscript, indicate that **1)** we have used appropriate approaches to knockdown ANXA1 resulting in specific and consistent effects on LGN-NuMA cortical accumulation and mitotic spindle orientation which we have rescued successfully using an siANXA1-resistant ANXA1-mCherry; **2)** we combined multiple experiments to show that ANXA1 is an upstream cue that regulates the LGN-mediated spindle orientation machinery; **3)** we used a unique mammary 3D culture system to validate our results and further demonstrate that ANXA1-mediated regulation of planar spindle orientation is required for proper luminogenesis and epithelial morphogenesis; **4)** while our results indicate that ANXA1 has a specific effect on LGN-NuMA and that this is likely to affect the F-actin-astral microtubule crosstalk; we have also conducted a candid discussion in our manuscript and in our response above of the possibility that ANXA1 may also act on the regulation of cortical actin organisation; these are important questions that we aim to investigate in my group in the future. We are very careful and modest with the statements that we make and conclusions that we draw throughout the text. The study describes a new mechanism, which most likely will take years to fully understand at a molecular level. But we feel that our results are robust and important to be conveyed to the researchers in the cell and developmental biology community.

REVIEWER COMMENTS

Reviewer #5 (Remarks to the Author):

The manuscript by Dr Elias and colleagues describes Annexin-1 (Anxa1) as a new player of the spindle orientation pathway in mammary epithelial cells, which is a novel finding.

To address the role of Anxa1 in the authors used siRNA based methods which induce down-regulation with possible differential penetrance in the cell population. To overcome this problem, the authors repeat the experiments several times and also show that misorientation and NuMA-LGN mislocalization defects can be rescued by ectopic expression of mCherry-Anxa1 in a silenced background, this way corroborating the idea that the phenotypes reported can be ascribed to lack of Anxa1. This considered, I would replace the blot in figure 2e with the one in figure 1b of the rebuttal, which seems more consistent with what shown in panel 2b. In addition, for the sake of completeness, I would also specify in the figure legends the number of replicates used for the plots quantifying "percentage of cells" for a given phenotypes (such as 2c, 2f, etc..).

My most substantial comment is on the mechanistic role of Anxa1 in determining LGN-NuMA cortical distribution and orientation. The authors show that Anxa1 knock-down induces defects in actomyosin cortex organization in metaphase (figure S4f). Several studies, included the ones cited in the manuscript and in the rebuttal, indicated that in the absence of properly assembled actomyosin cortex cells undergo misoriented divisions. Thus, if Anxa1 maintains actomyosin integrity and this is its main mitotic role, the altered distribution of LGN-NuMA and the misorientation will follow as a consequence of cortical disruption. The evidence that in metaphase Anxa1 is found among the interactors of LGN seems to indicate that additional molecular links exist between the two proteins, although this remains to be explored.

In the rebuttal, the authors clarify that they analyse MCF10A polarised cells with cell-cell contacts that are formed after 48h transfection (rebuttal to Reviewer-4). This is an aspect that is worth specifying in the main text because in analogous spindle orientation studies mitotic HeLa cells are analysed in isolation where cell-cell adhesion forces are not in place and cannot impact on spindle orientation dynamics.

Finally, regarding the use of TG100-115, which is also a PI3K inhibitor, in the absence of additional mutational analysis on Anxa1 or TRPM7, it could be useful to rule out the possibility that PI3K inhibition impacts on Anxa1 localization.

Point-by-point response to Reviewer #5:

Thanks very much for sending along the reviews of our paper. We are grateful for the Reviewer's comments, which we have addressed to improve the manuscript. We are also delighted that the Reviewer remarked the novelty of our findings establishing ANXA1 as a new key player regulating the spindle orientation machinery in mammary epithelial cells. Please find below our point-by-point response to all Reviewer's comments. We have revised the figures and modified and strengthened the text to address all the Reviewer's concerns.

Reviewer #5 (Remarks to the Author):

The manuscript by Dr Elias and colleagues describes Annexin-1 (Anxa1) as a new player of the spindle orientation pathway in mammary epithelial cells, which is a novel finding.

To address the role of Anxa1 in the authors used siRNA based methods which induce down-regulation with possible differential penetrance in the cell population. To overcome this problem, the authors repeat the experiments several times and also show that misorientation and NuMA-LGN mislocalization defects can be rescued by ectopic expression of mCherry-Anxa1 in a silenced background, this way corroborating the idea that the phenotypes reported can be ascribed to lack of Anxa1. This considered, I would replace the blot in figure 2e with the one in figure 1b of the rebuttal, which seems more consistent with what shown in panel 2b. In addition, for the sake of completeness, I would also specify in the figure legends the number of replicates used for the plots quantifying "percentage of cells" for a given phenotypes (such as 2c, 2f, etc..).

We thank The Reviewer for their comment that our siRNA-based methods and rescues allow us to conclude that spindle misorientation and LGN-NuMA mislocalisation defects are the result of ANXA1 knockdown. As requested by the Reviewer, we have now revised Figure 2 of the manuscript and replaced the blot in Figure 2e with the one included in the previous response to reviewer's Figure 1b. We have also added the number of replicates used for our quantifications in Figure 2c and 2f, as suggested by the Reviewer, and included this information in the other figure legends of the manuscript as appropriate.

My most substantial comment is on the mechanistic role of Anxa1 in determining LGN-NuMA cortical distribution and orientation. The authors show that Anxa1 knock-down induces defects in actomyosin cortex organization in metaphase (figure S4f). Several studies, included the ones cited in the manuscript and in the rebuttal, indicated that in the absence of properly assembled actomyosin cortex cells undergo misoriented divisions. Thus, if Anxa1 maintains actomyosin integrity and this is its main mitotic role, the altered distribution of LGN-NuMA and the misorientation will follow as a consequence of cortical disruption. The evidence that in metaphase Anxa1 is found among the interactors of LGN seems to indicate that additional molecular links exist between the two proteins, although this remains to be explored.

As remarked by the Reviewer, we discussed in our previous response to reviewers and revised manuscript the potential additional molecular mechanisms that may mediate the role of ANXA1 in the regulation of the crosstalk between the actomyosin cortex and the spindle orientation machinery. Indeed, given that ANXA1 has been shown to regulate F-actin dynamics, either through direct binding or actin-regulating proteins such as vimentin and profilin, ANXA1 effect on LGN-NuMA distribution at the cortex may be the result of F-actin disruption upon ANXA1 knockdown. An important finding in our study is that ANXA1 knockdown disrupts cortical F-actin organisation but does not prevent LGN-NuMA recruitment to the cell cortex (only affecting LGN-NuMA polarised cortical accumulation). By contrast F-actin depolymerisation using latrunculin A does not affect ANXA1 plasma membrane localisation while abrogating cortical recruitment of LGN-NuMA. As rightly pointed out by the

Reviewer, our findings throughout the manuscript suggest the existence of additional molecular mechanisms that may link between ANXA1 and LGN to regulate the dynamics of LGN-NuMA at the cortex. This is an important question that we are exploring in our group in a project where we are characterising the interactome of ANXA1 in mitotic mammary epithelial cells, to dissect the molecular mechanisms linking ANXA1 to LGN and F-actin to orient the mitotic spindle to its correct position. We have now amended the discussion of the revised manuscript to clarify this point further, as suggested by the Reviewer (page 18).

In the rebuttal, the authors clarify that they analyse MCF10A polarised cells with cell-cell contacts that are formed after 48h transfection (rebuttal to Reviewer-4). This is an aspect that is worth specifying in the main text because in analogous spindle orientation studies mitotic HeLa cells are analysed in isolation where cell-cell adhesion forces are not in place and cannot impact on spindle orientation dynamics.

We thank the Reviewer for their suggestion to revise the main text to clarify our siRNA-based experiments in polarised MCF-10A cells. Indeed, we transfect cells with siRNAs for 24h hours using the RNAiMAX procedure (Invitrogen), then cells are left in culture for additional 48h to ensure efficient knockdown of ANXA1 and the formation of cell-cell adhesions. We have amended the Results and Methods sections to explain why it is important to have established cell-cell adhesion in place for proper mitotic spindle orientation (pages 8, 22).

Finally, regarding the use of TG100-115, which is also a PI3K inhibitor, in the absence of additional mutational analysis on Anxa1 or TRPM7, it could be useful to rule out the possibility that PI3K inhibition impacts on Anxa1 localization.

TG100-115 is established as a potent inhibitor of the TRPM7 channel kinase activity (Song et al., 2017 BBA; Kollwe et al., 2021 eLife). TRPM7 phosphorylates ANXA1 thereby regulating its binding to S100A11 and recruitment to the plasma membrane. As the Reviewer pointed out, TG100-115 is also an inhibitor of the PI3K activity. Rarely inhibitor compounds are specific, thus we agree with the Reviewer this constitutes a limitation in the context of our experiments using TG100-115. Nonetheless, these experiments demonstrate that ANXA1 localisation to the plasma membrane is essential for its function as a regulator of LGN-NuMA. There are several PI3K inhibitors used in cell culture to inhibit the PI3K/AKT signalling pathway that is key for several essential cellular processes such as cell proliferation and survival. Most of these inhibitors (including the commonly used Y294002 compound), need long treatments reaching 24h to effectively inhibit PI3K/AKT signalling. In our experiments, we used TG100-115 for 2hr only, a duration that is sufficient to inhibit the TRPM7 kinase activity (Kollwe et al., 2021 eLife). As we discussed in our previous revised manuscript (page 10), this short treatment was sufficient to impair ANXA1 plasma membrane translocation without affecting cell proliferation. Importantly while the underlying mechanisms remain to be determined, several studies have established ANXA1 as an upstream regulator of the PI3K/AKT signalling in several normal and cancer cell models (Zhu et al., 2018 Cell Death & Disease; Hagihara et al., 2019; Sci Rep; Wei et al., 2021 ASN Neuro). Thus, our experimental setup, together with the data from the literature suggest that it is unlikely that PI3K affects the localisation or function of ANXA1. We have now discussed this point in our new revised manuscript to explain the limitation of the approach (page 17).

REVIEWERS' COMMENTS

Reviewer #5 (Remarks to the Author):

In the revised manuscript, Dr. Elias and colleagues modified the text and the figure format as suggested.

They explained in the rebuttal letter the rationale for using TG100-115 to inhibit the TRPM7 kinase, without performing additional experiments to rule out a possible inhibition of PI3K under the conditions used.

They did not provide additional evidence for the molecular role of ANXA1 in spindle positioning and LGN cortical distribution, beside the fact that it controls actomyosin integrity (already shown in supplementary figure S4f in the pervious version). In the absence of other information, I suggest to move panel S4f in the main figures, as actomyosin disruption remains the only possible molecular mechanism accounting for the misorientation defects observed upon ANXA1 loss.

Point-by-point response to Reviewer #5:

Reviewer #5 (Remarks to the Author):

In the revised manuscript, Dr. Elias and colleagues modified the text and the figure format as suggested.

They explained in the rebuttal letter the rationale for using TG100-115 to inhibit the TRPM7 kinase, without performing additional experiments to rule out a possible inhibition of PI3K under the conditions used.

They did not provide additional evidence for the molecular role of ANXA1 in spindle positioning and LGN cortical distribution, beside the fact that it controls actomyosin integrity (already shown in supplementary figure S4f in the pervious version). In the absence of other information, I suggest to move panel S4f in the main figures, as actomyosin disruption remains the only possible molecular mechanism accounting for the misorientation defects observed upon ANXA1 loss.

In response to the Reviewer's suggestion, we have now moved Supplementary Figure S4 to the main Figures to become Figure 5. Additionally, to address the Reviewer's points, we have now toned down further our statements related to the function of ANXA1 in the regulation of LGN and cortical F-actin, throughout the manuscripts.